# EFFICIENT REASONING WITH BALANCED THINKING

**Yulin Li**[1*] **Tengyao Tu**[1,5*] **Li Ding**[1] **Junjie Wang**[1]
**Huiling Zhen**[2] **Yixin Chen**[4] **Yong Li**[3,5] **Zhuotao Tian**[1,6†]

[1]Harbin Institute of Technology (Shenzhen) [2]Huawei Noah's Ark Lab [3]Tsinghua University
[4]The Chinese University of Hong Kong [5]Zhongguancun Academy [6]Shenzhen Loop Area Institute

## ABSTRACT

Large Reasoning Models (LRMs) have shown remarkable reasoning capabilities, yet they often suffer from overthinking, expending redundant computational steps on simple problems, or underthinking, failing to explore sufficient reasoning paths despite inherent capabilities. These issues lead to inefficiencies and potential inaccuracies, limiting practical deployment in resource-constrained settings. Existing methods to mitigate overthinking, such as suppressing reflective keywords or adjusting reasoning length, may inadvertently induce underthinking, compromising accuracy. Therefore, we propose REBALANCE, a training-free framework that achieves efficient reasoning with balanced thinking. REBALANCE leverages confidence as a continuous indicator of reasoning dynamics, identifying overthinking through high confidence variance and underthinking via consistent overconfidence. By aggregating hidden states from a small-scale dataset into reasoning mode prototypes, we compute a steering vector to guide LRMs' reasoning trajectories. A dynamic control function modulates this vector's strength and direction based on real-time confidence, pruning redundancy during overthinking, and promoting exploration during underthinking. Extensive experiments conducted on four models ranging from 0.5B to 32B, and across nine benchmarks in math reasoning, general question answering, and coding tasks demonstrate that REBALANCE effectively reduces output redundancy while improving accuracy, offering a general, training-free, and plug-and-play strategy for efficient and robust LRM deployment. Code is available at https://github.com/yu-lin-li/ReBalance.

## 1 INTRODUCTION

Recent advances in Supervised Fine-Tuning (SFT) and Reinforcement Learning (RL) have substantially enhanced the reasoning capabilities of Large Reasoning Models (LRMs) (Jaech et al., 2024; Guo et al., 2025; Team, 2025). However, LRMs may exhibit *overthinking* (Chen et al., 2024b), allocating redundant reasoning steps to simple problems. This redundancy incurs substantial computational costs with marginal performance gains (Sui et al., 2025), and may introduce hallucinations (Sun et al., 2025). Thus, overthinking severely limits the practical deployment of LRMs in resource-constrained environments.

Recent efforts (Yue et al., 2025) have been made to mitigate overthinking by shortening reasoning chains. However, these approaches primarily target overthinking and may overlook the critical issue of *underthinking* (Wang et al., 2025c), where LRMs fail to sufficiently explore valid reasoning paths despite possessing the inherent capability to solve the problem, as shown in Fig. 1(a). Specifically, Wang et al. (2025a), Ma et al. (2025b), and Chen et al. (2025b) suppress keywords indicative of reflection and exploration, but indiscriminately affect both redundant and valuable reasoning, inevitably causing underthinking. Another direction (Zhang et al., 2025c; Lou et al., 2025; Huang et al., 2025a) adjusts reasoning length based on problem difficulty via SFT or RL, yet often penalizes lengthy reasoning (Su et al., 2025b) or dilutes rewards for control tokens (Fang et al., 2025). Such designs may cause decision boundary collapse (Lou et al., 2025), biasing models toward overly short reasoning chains and inducing underthinking. Hence, a key question arises: *How can we mitigate overthinking without inducing underthinking, achieving efficient reasoning with balanced thinking?*

---

*Equal Contribution    † Corresponding Author (tianzhuotao@hit.edu.cn)

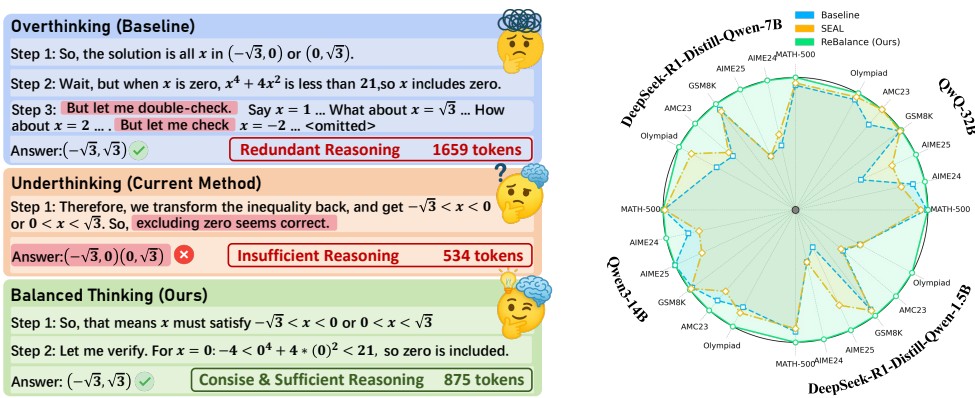

**(a)** Qualitative comparison.      **(b)** Quantitative comparison.

**Figure 1: Qualitative and quantitative comparisons with previous state-of-the-art methods for mitigating overthinking. (a)** Given the question "For what real values of $x$ is $-4 < x^4 + 4x^2 < 21$?", the model first obtains intervals $(-\sqrt{3}, 0)$ and $(0, \sqrt{3})$, and then verifies if $x = 0$ is included. However, the baseline (Guo et al., 2025) redundantly checks irrelevant values after correctly validating $x = 0$, causing overthinking. Current mitigation methods (Yang et al., 2025b) overly suppress necessary reflection, leading to underthinking. Our method dynamically controls the reasoning state, effectively balancing these two extremes. **(b)** REBALANCE outperforms previous state-of-the-art method (Chen et al., 2025b) across multiple mathematical reasoning datasets and model scales (0.5B–32B), reducing reasoning length while simultaneously improving accuracy.

**Key observations.** To address this issue, we need to develop a dynamic mechanism capable of explicitly modeling and controlling both overthinking and underthinking. Though recent works (Zhang et al., 2025a; Yang et al., 2025b; Lin et al., 2025a) have achieved dynamic control by adopting manually designed metrics to adaptively retain or discard entire reasoning paths, this rigid binary selection may sacrifice the potentially valuable intermediate reasoning steps, thus still risking underthinking. This motivates us to investigate a continuous and reliable indicator of reasoning states for providing dynamic fine-grained reasoning control.

As shown in Fig. 2, we can observe that the confidence values correlate with LRMs' reasoning behaviors. Specifically, high confidence variance may reflect frequent indecisive switching between different reasoning paths, causing redundant steps and delayed answer convergence, *i.e., overthinking*. Conversely, consistent overconfidence can lead to premature commitment to incorrect reasoning paths, *i.e., underthinking*. Thus, confidence can be leveraged as an indicator of reasoning dynamics. Given that LRMs' internal reasoning states are inherently represented by their hidden states (Su et al., 2025a), this observation prompts us to consider *whether the efficient reasoning can be achieved through balanced thinking, by dynamically adjusting hidden states according to confidence levels.*

**Our solution.** In this work, we propose **ReBalance**, a training-free method that achieves efficient **Re**asoning with **Balance**d thinking. To achieve dynamic control between overthinking and underthinking, we first identify reasoning steps indicating overthinking and underthinking from a small-scale seen dataset, aggregate their corresponding hidden states into reasoning mode prototypes, and compute a steering vector that encodes the transition between them, *i.e.,* from overthinking to underthinking. Since the steering vector captures the model's inherent reasoning dynamics, it exhibits strong generalization across diverse unseen data, as demonstrated in our experiments.

With this steering vector, we further introduce a dynamic control function that modulates the strength and direction of the vector based on the model's confidence at each step. When signs of overthinking emerge, the steering is amplified to prune redundancy. Conversely, when underthinking is inferred, steering is reversed to promote exploration of alternative reasoning paths. This adaptive mechanism effectively balances reasoning depth across various contexts, enhancing efficiency without compromising the core reasoning abilities.

Extensive experiments across four models ranging from 0.5B to 32B, and on nine benchmarks covering math reasoning, general question answering, and coding tasks, demonstrate the effectiveness and strong generalization capabilities of REBALANCE. Notably, REBALANCE not only reduces output length but also improves the accuracy. To summarize, our contributions are as follows:

- As the current methods struggle to balance between overthinking and underthinking, we identify that confidence can serve as a continuous and reliable signal for characterizing both overthinking and underthinking in LRMs, enabling fine-grained behavioral control.

- To achieve dynamic reasoning control, we propose REBALANCE, an efficient and training-free framework that dynamically steers the reasoning trajectory of LRMs by modulating their internal state based on confidence estimates.

- Extensive experiments across different models and tasks demonstrate that REBALANCE improves both inference efficiency and accuracy, offering a plug-and-play solution for boosting the efficiency of LRMs without compromising performance.

## 2 BACKGROUND AND MOTIVATION

### 2.1 PRELIMINARIES

In the following, to investigate the dynamics of the reasoning process of large reasoning models (LRMs), we introduce the computation of *stepwise confidence* and *confidence variance*. Stepwise confidence measures the degree to which the model consistently adheres to the same reasoning path, while confidence variance between different steps quantifies the frequency of switching between different reasoning paths. The discussion of related work is presented in Appendix G.

**Stepwise confidence.** For each token position $t \in \mathcal{T}_s$, we can define the tokenwise maximum predicted probability $p_t^{\max} = \max_{v \in V} \mathbf{p}_\theta(v \mid x_{<t})$. Then, we can obtain the confidence $c_s$ of the reasoning step $S_s$, which is the geometric average of these maxima across all tokens in the step:

$$c_s = \exp\left(\frac{1}{|\mathcal{T}_s|} \sum_{t \in \mathcal{T}_s} \ln p_t^{\max}\right) \tag{1}$$

**Confidence variance.** To capture short-term fluctuations in confidence, we compute the confidence variance $\mathrm{Var}(\cdot)$ over recent steps. Since long-term history is less relevant, we focus on local variability by calculating the variance within a sliding window of size $|W| \geq 1$, and we can define the window $\mathcal{W}_s$ for the $s$-th step as $\mathcal{W}_s = \{\max(1, s-W+1), \ldots, s\}$. Then, with the average step confidence $\bar{c}_s = \frac{1}{|\mathcal{W}_s|} \sum_{j \in \mathcal{W}_s} c_j$ within the window $\mathcal{W}_s$, we can obtain the confidence variance for the $s$-th step $\mathrm{Var}(c_s; \mathcal{W}_s)$ as:

$$\mathrm{Var}(c_s; \mathcal{W}_s) = \begin{cases} 0, & |\mathcal{W}_s| = 1, \\ \frac{1}{|\mathcal{W}_s|} \sum_{j \in \mathcal{W}_s} (c_j - \bar{c}_s)^2, & |\mathcal{W}_s| \geq 2. \end{cases} \tag{2}$$

To this end, regardless of the current stepwise confidence level, a high $\mathrm{Var}(c_s; \mathcal{W}_s)$ indicates frequent switching among different reasoning paths, which may force the model to continue generating redundant reasoning steps instead of concluding, leading to *overthinking*. Differently, consistently high $c_s$ with low $\mathrm{Var}(c_s; \mathcal{W}_s)$ implies premature commitment and potential *underthinking*. These statistics will guide the dynamic control mechanism that will be introduced later.

### 2.2 KEY OBSERVATIONS

As discussed above, existing approaches designed to mitigate overthinking effectively reduce the length of inference outputs, yet struggle to achieve satisfactory accuracy. To investigate the underlying reasons, we analyze how the length of reasoning sequences relates to the ground-truth reasoning length for both correctly and incorrectly answered samples, before and after applying methods intended to mitigate overthinking, as shown in Fig. 2(a). Specifically, we collect inference samples under three conditions: the original model, the model after applying existing methods, and the model after applying our proposed method. We utilize ground-truth as a proxy for ideal reasoning length.

**The trade-off between overthinking and underthinking.** Theoretically, if an overthinking mitigation approach effectively reduces redundant reasoning steps, the reasoning sequence lengths of

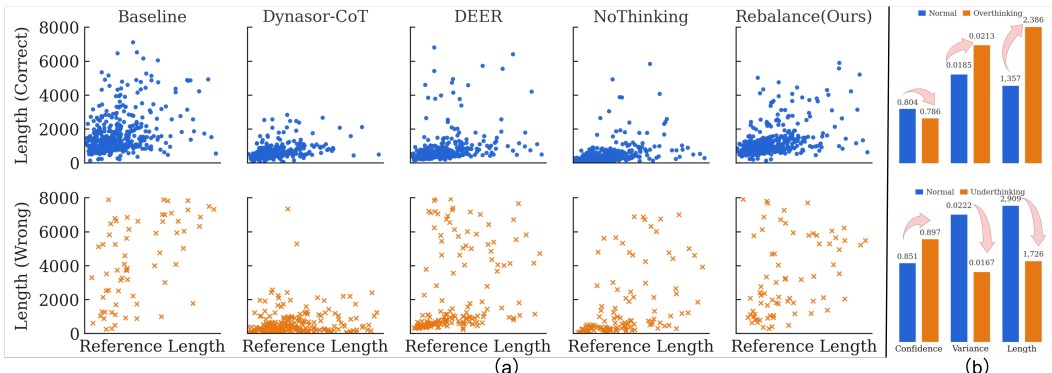

Figure 2: **(a) Effects of overthinking mitigation on reasoning modes.** We compare the distributions of reasoning lengths for correct and incorrect predictions before and after applying overthinking mitigation methods. The reduction in reasoning lengths for correct and incorrect predictions indicates the degree to which overthinking is alleviated and underthinking is introduced, respectively. Existing methods significantly introduce underthinking, whereas our method effectively achieves a balanced reduction of both. **(b) Correlation between confidence and reasoning modes.** We observe that the overthinking samples exhibit higher confidence variance compared to normal samples, while underthinking samples show persistently high confidence levels.

correctly answered samples should accordingly decrease. Conversely, if such methods introduce underthinking by prematurely truncating necessary reasoning, resulting in errors, the reasoning lengths for these incorrect samples should also decrease. As shown in Fig. 2(a), both existing methods and our proposed approach significantly mitigate overthinking. However, existing methods introduce notable underthinking, whereas our proposed approach maintains reasoning length distribution similar to the original model, demonstrating superior balanced thinking capacity.

Consequently, addressing the critical issue of simultaneously mitigating overthinking and preventing underthinking becomes essential. Achieving this requires explicit modeling of these two reasoning modes. Intuitively, questions correctly answered by the original model but incorrectly answered after applying overthinking mitigation methods are likely due to restricted exploration, indicating underthinking. Conversely, questions correctly answered by both the original and mitigated models with shortened reasoning sequences likely reflect the successful reduction of redundant steps, indicating overthinking. Based on these categorizations, we analyze changes in stepwise confidence and confidence variance relative to normal reasoning, as illustrated in Fig. 2(b).

**Confidence indicates reasoning states.** Our analysis reveals that overthinking typically coincides with higher confidence variance, indicative of hesitation across reasoning steps, while underthinking is characterized by persistently high confidence levels, reflecting premature commitment to incorrect reasoning paths without sufficient exploration. These findings support our proposal that confidence can serve as a continuous and reliable indicator of the model's reasoning state, enabling fine-grained behavioral control. A comprehensive analysis, including the correlation between confidence and reasoning length (Appendix A.2), inertia effects of confidence states (Appendix A.3), confidence variations across models (Appendix A.4), model keywords and confidence states (Appendix A.6), and the discriminability of confidence in latent space (Appendix A.5) are provided in the Appendix.

## 3 METHOD

### 3.1 OVERVIEW

In this section, we present REBALANCE, a training-free framework designed to dynamically balance overthinking and underthinking, thereby improving efficiency without compromising accuracy.

Specifically, REBALANCE first explicitly models reasoning states prone to overthinking or underthinking using stepwise confidence and confidence variance (Sec. 3.2). Next, it utilizes these identified states to extract distinct steering vectors from deep-layer hidden states, capturing key behavioral patterns of different reasoning modes between overthinking and underthinking (Sec. 3.3). Finally, the steering vectors will be controlled by a dynamic function that adaptively modulates steering

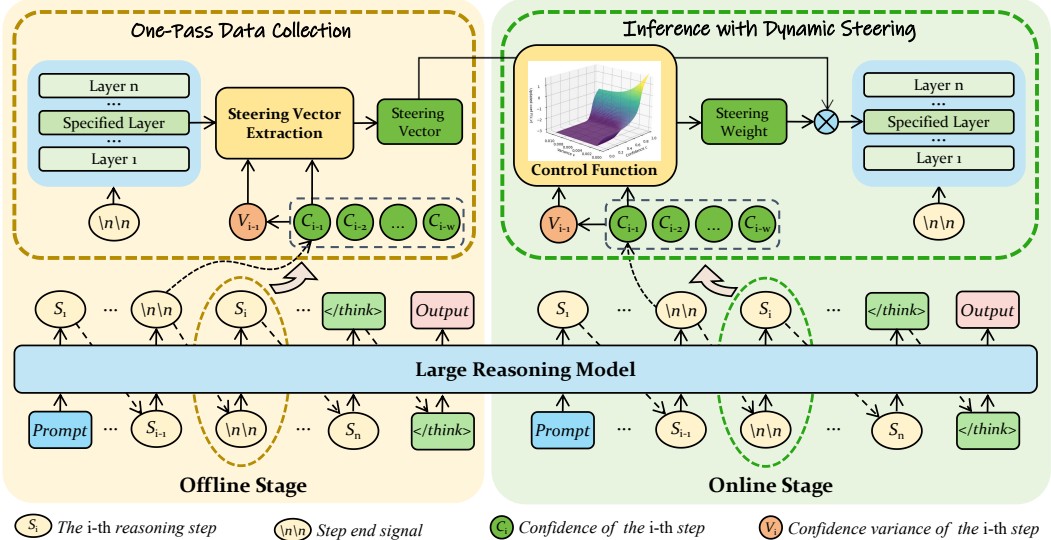

**Figure 3: Illustration of the REBALANCE framework.** We first perform offline one-pass data collection on a small-scale seen dataset. At each step, the steering vector is extracted at the first token of the specified layer based on confidence, and a dynamic function is fitted according to model behaviors. During deployment, the dynamic function outputs steering weights based on the model's real-time confidence online, thus balancing between overthinking and underthinking.

strength and direction, ensuring balanced thinking during the reasoning process (Sec. 3.4). Collectively, these complementary components provide precise, adaptive, and efficient control over the reasoning process. The overview is shown in Fig. 3.

## 3.2 EXPLICIT MODELING OF OVERTHINKING AND UNDERTHINKING

Building upon the insights that confidence serves as a reliable indicator of overthinking and underthinking, we first formally define these reasoning states and then explicitly model them using confidence metrics.

**Definitions of overthinking and underthinking.** Let the `<think>`...`</think>` trajectory be segmented into steps $S_1, \ldots, S_{s_{\max}}$ by the delimiter mentioned in Sec. 2.1. Denote the partial reasoning up to step $s$ by $r_{\leq s}$ and the induced answer distribution (if forced to stop at $s$) by $\pi_s$; let prediction $d_s = \arg\max \pi_s$ under a specified decoding rule, then we define the *stability index* as:

$$s^\star = \min\left\{ s : d_{s'} = d_s \text{ for all } s' \geq s \text{ and } d_s \text{ is correct} \right\}. \tag{3}$$

The stability index $s^\star$ serves as a signal to distinguish different reasoning modes. Specifically, A trajectory may exhibit *overthinking* if it continues after $s^\star$. Conversely, it exhibits *underthinking* if it stops at step $s$ with incorrect prediction $d_s$ while there exists $s' > s$ with correct $d_{s'}$. These definitions formalize the notions of redundant computation after convergence to the correct answer and premature termination before sufficient reasoning.

**Explicit modeling with confidence.** Then, the above definitions can be instantiated using the stepwise confidence $c_s$ and the confidence variance $v_s = \mathrm{Var}(c_s; \mathcal{W}_s)$ introduced in Sec. 2.1. With a small-scale seen dataset that has been used for training, we can obtain the empirical quantiles (Hyndman & Fan, 1996) $Q_c(\cdot)$ and $Q_v(\cdot)$ and thresholds as:

$$\tau_c^{\mathrm{L}} = Q_c(q_L), \quad \tau_c^{\mathrm{H}} = Q_c(q_H), \qquad \tau_v^{\mathrm{L}} = Q_v(q_L), \quad \tau_v^{\mathrm{H}} = Q_v(q_H), \tag{4}$$

where $0 < q_L < q_H < 1$ specify the lower and upper quantiles, respectively. Then, with these thresholds, we can classify the reasoning steps into two sets $\mathcal{O}$ and $\mathcal{U}$:

$$\mathcal{O} \leftarrow \left\{ s : c_s \leq \tau_c^{\mathrm{L}} \wedge v_s \geq \tau_v^{\mathrm{H}} \right\}, \qquad \mathcal{U} \leftarrow \left\{ s : c_s \geq \tau_c^{\mathrm{H}} \wedge v_s \leq \tau_v^{\mathrm{L}} \right\}. \tag{5}$$

Concretely, as illustrated in Fig. 2(b), the overthinking set $\mathcal{O}$ contains instances characterized by high reasoning variance and low confidence, reflecting unstable or oscillating reasoning trajectories.

On the other hand, the underthinking set $\mathcal{U}$ comprises cases with low variance and persistently high confidence, indicating premature convergence and a tendency toward underthinking. Instances not belonging to $\mathcal{O} \cup \mathcal{U}$ can be treated as *normal* and are excluded from further analysis.

### 3.3 Confidence-Based Steering Vector Extraction

In this section, based on the modeling of overthinking and underthinking introduced in Sec 3.2, we extract prototypical representations of both reasoning modes from the hidden states of LRMs via an offline, single forward pass. Then, the resulting prototypes enable the construction of a steering vector that delineates the trajectory from overthinking to underthinking, thereby facilitating fine-grained behavior control.

**One-pass prototype extraction.** To obtain prototypes, we perform a single offline inference pass over a small seen dataset $\mathcal{D}_{\text{seen}}$, segmenting reasoning steps by the delimiter \n\n. During this pass, we automatically select the optimal deep-layer based on LRMs' intrinsic separability of reasoning modes (see Appendix A.5), from which we collect hidden states $\mathbf{h}_{t_s^{(1)}}$ at the first token $t_s^{(1)}$ of each step. $\mathbf{h}_{t_s^{(1)}}$ serves as a compact encoding of step-level intent (Yang et al., 2025b) and, under causal masking, conditions the generation of all subsequent tokens within the step. We find that deeper layers exhibit stronger discriminability between reasoning modes and improved generalization across datasets, as analyzed in Appendix A.5.

Then, with the hidden stages and the tags $\mathcal{O}$ and $\mathcal{U}$ mentioned in Sec. 3.2 for each step, we can obtain the overthinking and underthinking prototypes, *i.e.,* $\boldsymbol{\mu}^{\text{O}}$ and $\boldsymbol{\mu}^{\text{U}}$, respectively:

$$\boldsymbol{\mu}^{\text{O}} = \frac{1}{|\mathcal{O}|} \sum_{s \in \mathcal{O}} \mathbf{h}_{t_s^{(1)}}, \qquad \boldsymbol{\mu}^{\text{U}} = \frac{1}{|\mathcal{U}|} \sum_{s \in \mathcal{U}} \mathbf{h}_{t_s^{(1)}}. \tag{6}$$

**Steering vector construction.** The prototypes $\boldsymbol{\mu}^{\text{O}}$ and $\boldsymbol{\mu}^{\text{U}}$ denote the representations leading to overthinking and underthinking respectively. The steering vector is then defined as the direction from underthinking $\boldsymbol{\mu}^{\text{U}}$ to overthinking $\boldsymbol{\mu}^{\text{O}}$:

$$\mathbf{v} = \frac{\boldsymbol{\mu}^{\text{O}} - \boldsymbol{\mu}^{\text{U}}}{\|\boldsymbol{\mu}^{\text{O}} - \boldsymbol{\mu}^{\text{U}}\|_2}. \tag{7}$$

With the steering vector $\mathbf{v}$, we can formalize the transition between two reasoning modes. To modulate the behavior during inference, we adjust the initial token $\mathbf{h}_{t_s^{(1)}}$ of each step as follows:

$$\tilde{\mathbf{h}}_{t_s^{(1)}} = \mathbf{h}_{t_s^{(1)}} + \alpha_s \mathbf{v}, \qquad \alpha_s = \lambda_s \delta_s, \ \lambda_s \geq 0, \ \delta_s \in \{+1, -1\}, \tag{8}$$

where $\alpha_s$ represents the signed steering weight at step $s$, combining the steering strength $\lambda_s$ and direction $\delta_s$. When $\delta_s = +1$, we can address underthinking by stimulating the exploration of alternative reasoning paths. Conversely, $\delta_s = -1$ mitigates overthinking by encouraging commitment. These adjustments conceptually establish the boundaries within which the model's reasoning process operates, aiming to maintain a balanced state that ensures efficient and effective reasoning.

### 3.4 Model Behavior–Based Dynamic Control Function

Considering the evolving nature of model states and contexts over time, we introduce a dynamic control function that adaptively adjusts steering strength and direction during inference. Motivated by Sec. 2.2, which shows that the confidence correlates with reasoning modes, the steering weight $\alpha_s$ can be deemed as the output of a continuous function $g(c_s, v_s)$ with respect to the current confidence $c_s$ and variance $v_s$. Therefore, the steering weight $\alpha_s$, strength $\lambda_s$ and direction $\delta_s$ are defined as:

$$\alpha_s = g(c_s, v_s) = \delta_s \cdot \lambda_s. \tag{9}$$

During inference, at each step $s$, we obtain the confidence $c_s$ and variance $v_s$, set $\alpha_s = g(c_s, v_s)$, and inject $\alpha_s \mathbf{v}$ at the first token $t_s^{(1)}$ for the selected layer as in Eq. (8). This keeps trajectories between the overthinking and underthinking boundaries while adding no extra forward passes beyond standard decoding. Concretely, the dynamic control function $g(c_s, v_s)$ formulates as:

$$g(c_s, v_s) = \underbrace{\text{sign}\left(c_s - \tau_c^{\text{H}}\right)}_{\text{Steering direction } \delta_s} \cdot \underbrace{B(c_s, v_s) \tanh\left(\left|c_s - \tau_c^{\text{H}}\right|\right)}_{\text{Steering strength } \lambda_s} \tag{10}$$

| Method | MATH-500 | | AIME24 | | AIME25 | | GSM8K | | AMC23 | | Olympiad | |
|---|---|---|---|---|---|---|---|---|---|---|---|---|
| | Pass@1 ↑ | #Tokens ↓ | Pass@1 ↑ | #Tokens ↓ | Pass@1 ↑ | #Tokens ↓ | Pass@1 ↑ | #Tokens ↓ | Pass@1 ↑ | #Tokens ↓ | Pass@1 ↑ | #Tokens ↓ |
| **DeepSeek-R1-Distill-Qwen-1.5B** | | | | | | | | | | | | |
| Baseline (Guo et al., 2025) | 79.6 | 4516 | 23.3 | 12739 | 20.0 | 12739 | 76.0 | 1018 | 62.5 | 8239 | 41.2 | 8785 |
| CoD (Xu et al., 2025a) | 80.2 | 3512 | 33.3 | 11894 | 13.3 | 10941 | 69.5 | 531 | 62.5 | 6812 | 35.2 | 7160 |
| DEER (Yang et al., 2025b) | 67.0 | 2251 | 20.0 | 8135 | 23.3 | 8719 | 69.2 | 684 | 57.5 | 5132 | 35.4 | 5982 |
| NoThinking (Ma et al., 2025b) | 75.0 | 1582 | 6.7 | 6998 | 16.6 | 7473 | 61.6 | 285 | 60.0 | 3267 | 37.3 | 3485 |
| NoWait (Wang et al., 2025a) | 78.0 | 2645 | 30.0 | 8225 | 16.6 | 7574 | 75.1 | 641 | 60.0 | 3302 | 40.6 | 4795 |
| Dynasor-CoT (Fu et al., 2025) | 77.2 | 3694 | 26.7 | 10564 | 26.7 | 12462 | 77.1 | 1035 | 72.5 | 6505 | 42.6 | 8859 |
| SEAL (Chen et al., 2025b) | 78.6 | 3259 | 23.3 | 10785 | 26.7 | 8544 | 76.4 | 754 | 75.0 | 5084 | 32.7 | 7117 |
| Manifold Steering (Huang et al., 2025b) | 78.6 | 3458 | 30.0 | 10134 | – | – | 77.2 | 593 | 72.5 | 6236 | – | – |
| **ReBalance (Ours)** | **83.0** | 3474 | **33.3** | 10668 | **26.7** | 10672 | **78.3** | 765 | **72.5** | 5744 | **43.9** | 7235 |
| Δ vs. Baseline | (+3.4) | (−23.1%) | (+10.0) | (−14.3%) | (+6.7) | (−16.2%) | (+2.3) | (−24.9%) | (+10.0) | (−30.2%) | (+2.7) | (−17.6%) |
| **DeepSeek-R1-Distill-Qwen-7B** | | | | | | | | | | | | |
| Baseline (Guo et al., 2025) | 89.8 | 3699 | 46.7 | 11923 | 30.0 | 11923 | 89.2 | 1098 | 85.0 | 6375 | 56.1 | 7590 |
| CoD (Xu et al., 2025a) | 90.0 | 3127 | 46.7 | 11663 | 36.7 | 10198 | 84.5 | 339 | 85.0 | 3654 | 47.5 | 5688 |
| DEER (Yang et al., 2025b) | 87.8 | 2367 | 50.0 | 8924 | 40.0 | 8919 | 90.4 | 676 | 80.0 | 5157 | 53.9 | 5804 |
| NoThinking (Ma et al., 2025b) | 80.6 | 834 | 26.7 | 4427 | 20.0 | 7850 | 87.1 | 284 | 65.0 | 1911 | 45.3 | 3331 |
| NoWait (Wang et al., 2025a) | 86.8 | 2479 | 50.0 | 6844 | 26.7 | 6979 | 90.2 | 806 | 85.0 | 3795 | 52.1 | 4760 |
| Dynasor-CoT (Fu et al., 2025) | 88.2 | 2723 | 46.7 | 9864 | 33.3 | 11069 | 88.6 | 732 | 85.0 | 5121 | 55.4 | 7427 |
| SEAL (Chen et al., 2025b) | 90.6 | 2843 | 43.3 | 10112 | 26.7 | 9835 | 88.4 | 811 | 77.5 | 5164 | 53.9 | 6261 |
| Manifold Steering (Huang et al., 2025b) | 88.4 | 2239 | 53.3 | 8457 | – | – | 87.6 | 440 | 87.5 | 4440 | – | – |
| **ReBalance (Ours)** | **92.6** | 2903 | **53.3** | 9948 | **40.0** | 11074 | **91.6** | 912 | **92.5** | 4667 | **57.0** | 6321 |
| Δ vs. Baseline | (+2.8) | (−21.5%) | (+6.6) | (−16.6%) | (+10.0) | (−7.1%) | (+2.4) | (−16.9%) | (+7.5) | (−26.8%) | (+0.9) | (−16.7%) |
| **Qwen3-14B** | | | | | | | | | | | | |
| Baseline (Yang et al., 2025a) | 93.8 | 4470 | 66.7 | 10888 | 56.7 | 13125 | 95.1 | 2231 | 95.0 | 7240 | 60.6 | 7450 |
| CoD (Xu et al., 2025a) | 93.8 | 2950 | 66.7 | 10212 | 53.3 | 11828 | 95.6 | 627 | 95.0 | 5360 | 62.2 | 6554 |
| DEER (Yang et al., 2025b) | 93.0 | 2825 | 66.7 | 9973 | 56.7 | 11806 | 95.8 | 934 | 95.0 | 5527 | 66.1 | 6849 |
| NoThinking (Ma et al., 2025b) | 93.8 | 2657 | 70.0 | 8898 | 53.3 | 9892 | 95.1 | 369 | 87.5 | 4503 | 64.0 | 5880 |
| NoWait (Wang et al., 2025a) | 92.8 | 3219 | 60.0 | 10507 | 56.7 | 10924 | 95.6 | 1129 | 95.0 | 5050 | 59.2 | 7332 |
| Dynasor-CoT (Fu et al., 2025) | 93.8 | 4063 | 73.3 | 10369 | 60.0 | 12159 | 95.6 | 1483 | 95.0 | 6582 | – | – |
| SEAL (Chen et al., 2025b) | 93.4 | 3727 | 63.3 | 10322 | 50.0 | 10901 | 95.7 | 1369 | 90.0 | 6126 | 62.3 | 7131 |
| **ReBalance (Ours)** | **94.0** | 3641 | **73.3** | 9464 | **56.7** | 11057 | **96.3** | 1441 | **100.0** | 5230 | **66.3** | 7257 |
| Δ vs. Baseline | (+0.2) | (−18.5%) | (+6.6) | (−13.1%) | (+0.0) | (−15.8%) | (+1.2) | (−35.4%) | (+5.0) | (−27.8%) | (+5.7) | (−2.6%) |
| **QwQ-32B** | | | | | | | | | | | | |
| Baseline (Team, 2025) | 94.8 | 4535 | 66.7 | 14342 | 53.3 | 12627 | 96.3 | 1506 | 87.5 | 7021 | 66.7 | 8219 |
| CoD (Xu et al., 2025a) | 93.8 | 3516 | 63.3 | 11438 | 46.7 | 12189 | 96.2 | 670 | 92.5 | 6217 | 67.7 | 7028 |
| DEER (Yang et al., 2025b) | 94.4 | 3179 | 70.0 | 8885 | 46.7 | 10972 | 96.2 | 944 | 95.0 | 6435 | 64.3 | 7085 |
| NoThinking (Ma et al., 2025b) | 94.8 | 3912 | 66.7 | 10507 | 66.7 | 11839 | 96.5 | 1326 | 90.0 | 7119 | 66.1 | 8132 |
| NoWait (Wang et al., 2025a) | 93.8 | 2879 | 66.7 | 8190 | 63.3 | 8970 | 96.3 | 942 | 92.5 | 4717 | 62.6 | 8223 |
| Dynasor-CoT (Fu et al., 2025) | 94.2 | 4176 | 63.3 | 11156 | – | – | 95.2 | 1095 | 90.0 | 6544 | – | – |
| SEAL (Chen et al., 2025b) | 92.6 | 3536 | 63.3 | 10344 | 56.7 | 11384 | 96.2 | 1221 | 95.0 | 6341 | 67.5 | 7371 |
| FlashThink (Jiang et al., 2025) | 93.2 | 3144 | 60.0 | 10034 | 40.0 | 11861 | 96.5 | 910 | 92.5 | 6702 | – | – |
| TrimR (Lin et al., 2025a) | 93.8 | 3830 | 56.7 | 8345 | 43.3 | 8827 | 93.7 | 1319 | 90.0 | 6055 | – | – |
| **ReBalance (Ours)** | **95.2** | 3662 | **70.0** | 10350 | **63.3** | 11575 | **96.8** | 1289 | **95.0** | 6064 | **68.6** | 7422 |
| Δ vs. Baseline | (+0.4) | (−19.3%) | (+3.3) | (−27.8%) | (+10.0) | (−8.3%) | (+0.5) | (−14.4%) | (+7.5) | (−13.6%) | (+1.9) | (−9.7%) |

**Table 1: Performance on math reasoning benchmarks.** Metrics include Pass@1 (↑) and #Tokens (↓) on six math reasoning benchmarks. Changes are shown in orange for Pass@1 and blue for #Tokens. FlashThink and TrimR are reproduced according to the paper.

**The steering direction** $\delta_s$ is determined by the sign function $\operatorname{sign}(c_s - \tau_c^{\mathrm{H}})$, where the confidence threshold $\tau_c^{\mathrm{H}}$ is obtained as Eq. (4). It takes a negative value when confidence is below the high-confidence threshold ($c_s < \tau_c^{\mathrm{H}}$) to mitigate overthinking, and a positive value when confidence is above this threshold ($c_s > \tau_c^{\mathrm{H}}$) to alleviate underthinking. This guarantees the steering consistently directs the state away from the nearer reasoning boundary.

**The steering strength** $\lambda_s$ is composed of two parts: (1) *soft saturation* $\tanh\big(|c_s - \tau_c^{\mathrm{H}}|\big)$ and (2) *variance-aware amplitude* $B(c_s, v_s)$. Specifically, regarding the soft saturation $\tanh\big(|c_s - \tau_c^{\mathrm{H}}|\big)$, a smooth, saturating growth in $|c_s - \tau_c^{\mathrm{H}}|$ avoids abrupt changes and keeps the mapping monotone in $c_s$ for any fixed $v_s$. The soft saturation function guarantees the steering strength grows gradually as the state approaches the reasoning boundary, ensuring numerical stability.

Differently, the variance-aware amplitude $B(c_s, v_s)$ is a model behavior-based scalar amplitude that adapts across models based on the step confidence $c_s$ and variance $v_s$. It is required to indicate the model's current thinking status, shifting between moderate and overthinking/underthinking reasoning modes. To this end, the amplitude function can be formulated as:

$$B(c_s, v_s) = \begin{cases} B_{\mathrm{m}} + \big(B_{\mathrm{o}} - B_{\mathrm{m}}\big)\psi(c_s, v_s) & \text{if } c_s \le \tau_c^{\mathrm{L}} \text{ and } v_s \ge \tau_v^{\mathrm{H}}, \\ B_{\mathrm{m}} + \big(B_{\mathrm{u}} - B_{\mathrm{m}}\big)\psi(c_s, v_s) & \text{if } c_s \ge \tau_c^{\mathrm{H}} \text{ and } v_s \le \tau_v^{\mathrm{L}}, \\ B_{\mathrm{m}} & \text{otherwise.} \end{cases} \tag{11}$$

In Eq. (11), $B_{\mathrm{m}}$, $B_{\mathrm{o}}$, and $B_{\mathrm{u}}$ are adaptive mode boundaries representing moderate, overthinking, and underthinking, respectively. $\psi(c_s, v_s)$ denotes a conditioned gating function whose output ranges from 0 to 1 to ensure smooth transitions. The thresholds $(\tau_c^{\mathrm{L}}, \tau_c^{\mathrm{H}}, \tau_v^{\mathrm{L}}, \tau_v^{\mathrm{L}})$ are obtained in Eq. (4). Following the reasoning mode definitions outlined in Eq. (5), when $c_s \le \tau_c^{\mathrm{L}}$ and $v_s \ge \tau_v^{\mathrm{H}}$, indicating a state of overthinking, the transition occurs between $B_{\mathrm{m}}$ and $B_{\mathrm{o}}$. Differently, when $c_s \ge \tau_c^{\mathrm{H}}$ and $v_s \le \tau_v^{\mathrm{L}}$, indicating a state of overthinking, the transition should be performed between $B_{\mathrm{m}}$ and $B_{\mathrm{u}}$. Notably, $B_{\mathrm{m}}$ and $B_{\mathrm{o}}$ are adaptively derived from models without manual tuning.

In this context, the amplitude $B(c_s, v_s)$ serves as an indicator of the current reasoning status, complemented by the saturation function, which ensures the numerical stability of the final steering

| | SCIENCE | | COMMONSENSE | | PROGRAMMING | |
| | *GPQA-D* | | *StrategyQA* | | *LiveCodeBench* | |
| Method | Pass@1 ↑ | #Tokens ↓ | Pass@1 ↑ | #Tokens ↓ | Pass@1 ↑ | #Tokens ↓ |
|---|---|---|---|---|---|---|
| **DeepSeek-R1-Distill-Qwen-1.5B** | | | | | | |
| Baseline | 17.1 | 8 727 | 63.2 | 435 | 19.5 | 12 509 |
| Ours | 21.7 (+4.6) | 6 902 (−20.9%) | 67.7 (+4.5) | 401 (−7.8%) | 22.5 (+3.0) | 11 622 (−7.1%) |
| **DeepSeek-R1-Distill-Qwen-7B** | | | | | | |
| Baseline | 33.8 | 7 392 | 88.1 | 350 | 44.0 | 9 851 |
| Ours | 39.4 (+5.6) | 5 180 (−29.9%) | 88.9 (+0.8) | 310 (−11.4%) | 46.5 (+2.5) | 8 651 (−12.2%) |
| **Qwen3-14B** | | | | | | |
| Baseline | 60.6 | 7 451 | 94.2 | 267 | 83.5 | 7 101 |
| Ours | 67.2 (+6.6) | 5 779 (−22.4%) | 94.3 (+0.1) | 260 (−2.6%) | 84.6 (+1.1) | 6 088 (−14.3%) |
| **QwQ-32B** | | | | | | |
| Baseline | 63.1 | 7 424 | 93.6 | 274 | 87.5 | 6 622 |
| Ours | 67.2 (+4.1) | 6 296 (−15.2%) | 95.7 (+2.1) | 265 (−3.3%) | 88.3 (+0.8) | 5 649 (−14.7%) |

**Table 2: Generalization capabilities on other non-math tasks.** Metrics include Pass@1 (↑) and #Tokens (↓). Changes are shown in orange for Pass@1 and blue for #Tokens.

| Method | Math500 | | GSM8K | | Olympiad | |
| | Pass@1 ↑ | #Tokens ↓ | Pass@1 ↑ | #Tokens ↓ | Pass@1 ↑ | #Tokens ↓ |
|---|---|---|---|---|---|---|
| ReBalance (Ours) | 83.0 | 3474 | 78.3 | 765 | 43.9 | 7235 |
| $\|\mathcal{W}_s\| = 5$ | 82.4 | 3686 | 78.2 | 910 | 42.2 | 7343 |
| $\|\mathcal{W}_s\| = 10$ | 81.0 | 4084 | 77.8 | 940 | 44.7 | 7679 |
| $B_u = 0$ | 82.0 | 3343 | 78.1 | 761 | 39.4 | 7147 |
| $B_u = 0.2$ | 83.6 | 3600 | 80.5 | 850 | 44.2 | 7923 |
| $B_u = 0.5$ | 81.4 | 3543 | 77.7 | 890 | 41.3 | 8773 |

**Table 3: Ablations on the R1-1.5B backbone across difficulty levels.** We analyze performance changes on three math benchmarks with varying difficulty: **Math500** (medium), **GSM8K** (easy), and **Olympiad** (hard). Metrics are Pass@1 accuracy (%) and generated token numbers. Arrows indicate change relative to our original REBALANCE: Acc. ↑ increase, ↓ decrease; Tokens ↓ decrease, ↑ increase.

strength. More details, theoretical derivations, and proofs regarding the mode boundaries and the gating function are provided in Appendix B due to the page limit.

# 4 EXPERIMENT

Evaluation is conducted on *mathematics reasoning* datasets: MATH-500 (Lightman et al., 2023b), AIME24 (AI-MO, 2024a), AIME25 (OpenCompass, 2025), AMC23 (AI-MO, 2024b), GSM8K (Cobbe et al., 2021), and OLYMPIADBENCH (He et al., 2024); *scientific reasoning* dataset, GPQA DIAMOND (Rein et al., 2024); *commonsense reasoning* dataset, STRATEGYQA (Geva et al., 2021); and *code reasoning* dataset, LIVECODEBENCH (Jain et al., 2024). Besides, the proposed steering extraction and dynamic control function fitting are performed for each backbone once and held fixed across all unseen benchmarks for evaluation. 500 randomly sampled MATH (Hendrycks et al., 2021) problems are utilized during these processes, and the sensitivity analysis is shown in Fig. 4(c). More comprehensive experimental details, including the baseline introductions, are provided in Appendix D.

## 4.1 MAIN RESULTS

**Math reasoning.** As shown in Tab. 1, REBALANCE outperforms all baselines on six math reasoning benchmarks spanning diverse difficulties and distributions. Without introducing any auxiliary models or additional inference stages, it simultaneously improves Pass@1 by up to 7.0 points and reduces the average generated token count by at most 52.3%. Beyond these overall gains, the results also reveal several consistent behavioral patterns across different model families.

In particular, prompt-based reasoning suppression methods such as NoThinking affect distilled models much more severely than non-distilled ones. For example, large accuracy drops are observed on the DeepSeek-R1-Distill models, while the impact on Qwen3-14B and QwQ-32B remains comparatively modest. This suggests that distilled reasoning models are more sensitive to prompt-level manipulations of the reasoning structure. We also observe that methods relying on external auxiliary

Figure 4: (a-b) Layerwise Performance of MATH-500 for (a) R1–7B and (b) QwQ–32B. (c) Sensitivity to sample size for steering vector extraction. (d) Performance with cross-domain vectors.

models for early-exit decisions, such as TrimR and FlashThink, generally underperform approaches based on intrinsic model signals, such as DEER. This suggests that external stopping criteria may not faithfully reflect the internal reasoning state of the target model, whereas intrinsic signals provide a more reliable and lightweight control mechanism.

**Other reasoning scenarios.** We evaluate REBALANCE in a cross-domain setting, fixing the steering vector and control surface across tasks. As shown in Tab. 2, without domain-specific tuning, REBALANCE maintains Pass@1 and reduces the reasoning length by at most 29.9% on challenging scientific reasoning, programming, and simpler commonsense QA tasks. These results demonstrate strong cross-domain generalization.

Furthermore, the confidence signals extracted from mathematical reasoning generalize well across different task domains. On more challenging tasks, such as GPQA and code generation, the proposed method significantly improves effective reasoning while reducing redundant computation. In contrast, on relatively saturated tasks like StrategyQA, the method behaves more conservatively, as the model exhibits fewer overthinking states to be captured.

**Compatibility with NPU devices.** To further validate the generality of REBALANCE, we evaluate its performance on NPU devices using the NPU-native LRM openPangu-Embedded-7B-V1.1 (Chen et al., 2025a). REBALANCE consistently demonstrates strong balanced thinking capabilities. The implementation details and results are provided in Appendix C.10 due to page limit.

## 4.2 ABLATION STUDY

**Impact of static $\alpha_s$ control.** We ablate the dynamic schedule by fixing the steering weight $\alpha_s$. As shown in Fig. 5, positive $\alpha_s$ ($\mathcal{U} \to \mathcal{O}$) improves accuracy but increases reasoning length, intensifying with larger $|\alpha_s|$ (e.g., $\alpha_s = +3$ on QwQ–32B yields a 147% token increase). Negative $\alpha_s$ reduces tokens at the expense of accuracy. These results motivate dynamic adapting $\alpha_s$ to instance difficulty.

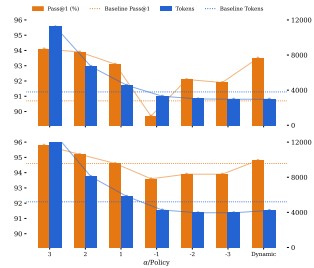

**Impact of the steering layer.** We test generalization by fitting *steering vectors* and *control surfaces* at various layers (selected by depth ratio). Fig. 4(a–b) shows that steering at any tested depth reduces token count without harming accuracy. The strongest trade-off appears in mid-to-late layers, consistent with our probing analysis (Appendix A.5), where representations from these layers exhibit the highest confidence separability and thus incur minimal noise.

**Figure 5: Static $\alpha_a$ Control on MATH-500**. Top: *R1-7B*; Bottom: *QwQ–32B*.

**Steering vector choice and generalization.** As shown in Fig. 4(c–d), we estimate steering vectors from math datasets of varying scale and a cross-domain science corpus (GPQA), fitting a *control surface* for each. Vectors generalize across datasets: REBALANCE improves efficiency while preserving accuracy. A clear trend emerges: vectors from harder datasets prioritize accuracy gains over token savings, aligning with the method's mechanism: harder data induces a *conservative* surface prioritizing correctness, while easier data yields a more *aggressive* one favoring token savings.

**Impact of window size $|\mathcal{W}_s|$.** Rows 4 and 5 of Tab. 10 show how the control-surface window size $|\mathcal{W}_s|$ affects reasoning. Empirically, larger windows substantially increase token usage, consistent

| Gating Functions | MATH-500 | | GSM8K | | Olympiad | | GPQA-D | |
|---|---|---|---|---|---|---|---|---|
| | Pass@1 ↑ | #Tokens ↓ | Pass@1 ↑ | #Tokens ↓ | Pass@1 ↑ | #Tokens ↓ | Pass@1 ↑ | #Tokens ↓ |
| **DeepSeek-R1-Distill-Qwen-1.5B** | | | | | | | | |
| Baseline | 79.6 | 4516 | 76.0 | 1018 | 41.2 | 8785 | 17.1 | 8727 |
| Sigmoid Gate | 83.0 | 3474 | 78.3 | **765** | 43.9 | 7235 | **21.7** | 6902 |
| Linear Gate | 83.6 | 3600 | 79.2 | 800 | 42.5 | **6794** | 17.7 | 7019 |
| Hard-Step Gate | 81.2 | 3830 | 79.7 | 820 | 41.8 | 7528 | 17.2 | 6537 |
| Polynomial-Fitted Gate | 82.8 | 3508 | 79.0 | 798 | **44.2** | 7059 | 20.7 | **6161** |
| ReLU-Shaped Gate | **84.0** | **3277** | **80.1** | 818 | 43.0 | 6875 | 19.7 | 6966 |
| **DeepSeek-R1-Distill-Qwen-7B** | | | | | | | | |
| Baseline | 89.8 | 3699 | 89.2 | 1098 | 56.1 | 7590 | 33.8 | 7392 |
| Sigmoid Gate | **92.6** | **2903** | 91.6 | 912 | 57.0 | 6321 | 39.4 | **5180** |
| Linear Gate | 91.2 | 3113 | 90.0 | 936 | 56.2 | 6303 | **43.0** | 5589 |
| Hard-Step Gate | 92.0 | 3115 | 90.0 | **909** | **57.8** | 6319 | 40.4 | 5448 |
| Polynomial-Fitted Gate | 91.6 | 2932 | 91.3 | 913 | 56.0 | **6206** | 38.9 | 5768 |
| ReLU-Shaped Gate | 91.4 | 3054 | **91.7** | 928 | 57.0 | 6362 | 39.9 | 5758 |

**Table 4: Performance comparison of different gating functions across benchmarks.** We evaluate both DeepSeek-R1-Distill-Qwen-1.5B and DeepSeek-R1-Distill-Qwen-7B on four reasoning datasets. Metrics are Pass@1 accuracy (%) and the number of generated tokens.

with the intuition that they smooth short-term fluctuations while reducing responsiveness to local anomalies. Theoretically, we show that the confidence trajectory during inference satisfies a Markovian continuity assumption (see Appendix A.3); a small window ($|\mathcal{W}_s| = 2$) is therefore sufficiently expressive and more sensitive to local reasoning patterns. However, with a larger window, accuracy increases on the hard *Olympiad* benchmark, in line with prior findings that extended deliberation improves performance at the expense of longer outputs (Jin et al., 2024; Muennighoff et al., 2025).

**Impact of underthinking mode boundary $B_u$.** Tab. 10 (Row 6) shows that removing the underthinking mode boundary slightly reduces tokens but significantly lowers accuracy, especially on tasks demanding extended reasoning. Moderate increases ($0.1 \to 0.2$; Rows 7–8) encourage deeper deliberation and boost accuracy at a modest token cost, while larger increases ($0.1 \to 0.5$) trigger overthinking and degrade performance. An overly strong boundary disrupts these normal reasoning paths. The other two mode boundaries $B_m$ and $B_o$, are adaptively determined based on model behavior, requiring no manual tuning (Appendix B).

**Impact of gating function.** To investigate the sensitivity of our method to the specific form of the gating function, we conduct an ablation study by replacing the default sigmoid gate with several alternative gating strategies. As shown in Tab. 4, we evaluate four variants: a linear gate, a hard-step gate, a polynomial-fitted gate, and a ReLU-shaped gate. Results across multiple models, difficulty levels, and domains indicate that all variants achieve effective reasoning performance, with only minor differences observed across datasets. This demonstrates that our approach is robust to the particular choice of gating function.

## 5 CONCLUSION

This paper analyzes the limitations of existing approaches to overthinking mitigation, and we observe that such attempts often introduce the countervailing problem of underthinking. Therefore, we propose REBALANCE, a training-free method that curbs overthinking while avoiding underthinking. Extensive experiments across diverse models and datasets show that REBALANCE reduces redundancy while preserving accuracy, achieving efficient reasoning with balanced thinking. A promising future direction is to apply REBALANCE to the multi-modal reasoning scenarios.

## ACKNOWLEDGEMENT

This work was supported by the Guangdong Basic and Applied Basic Research Foundation (2025A1515011546) and by the Shenzhen Science and Technology Program (JCYJ20240813105901003, KJZD20240903102901003, ZDCY20250901113000001).

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

CONTENTS

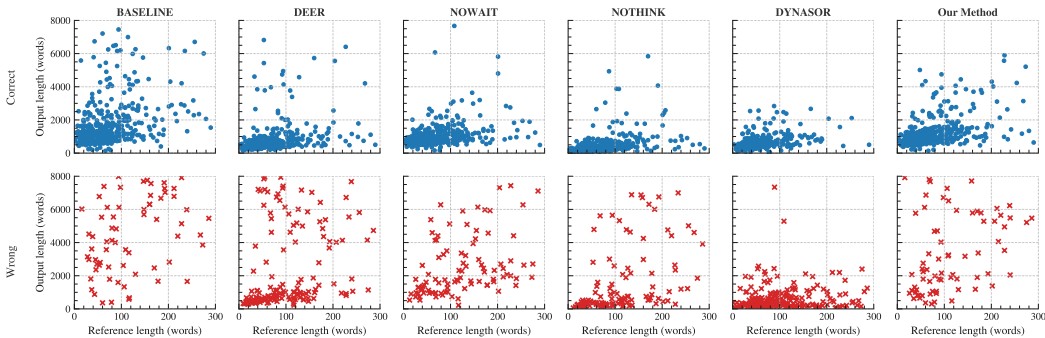

**Figure 6:** Mitigating Overthinking Without Inducing Underthinking. Evaluation on the *MATH* dataset with DeepSeek-R1-Distill-Qwen-1.5B. Each panel plots output length vs. reference length (words), restricted to reference $\leq 300$ and output $\leq 8000$. Top: correct examples. Bottom: incorrect examples. Competing methods reduce overthinking at the cost of underthinking, whereas **Our Method** mitigates overthinking without inflating underthinking.

## A SUPPLEMENTARY MOTIVATION AND EVIDENCE

### A.1 FROM OVERTHINKING TO UNDERTHINKING

*Why do many anti-overthinking techniques backfire as underthinking?* Empirically, the chain of thought (CoT) length and model performance are positively correlated, so aggressively truncating or penalizing long chains can excise necessary reasoning and degrade accuracy (Jin et al., 2024). Token-complexity theory (Lee et al., 2025) further posits an intrinsic minimum token budget for success. Enforcing uniformly short budgets or early termination pushes more instances below this threshold, yielding concise but wrong outputs. Moreover, the reasoning-boundary framework (Chen et al., 2024a) shows that optimal CoT length and reasoning path selection are task-dependent, thus global length controls disregard this heterogeneity and may steer reasoning trajectories outside the feasible region for a given task.

As shown in Fig. 6, on the *MATH* dataset with DEEPSEEK-R1-DISTILL-QWEN-1.5B, prior methods indeed curb *overthinking* but often induce *underthinking*, manifesting as a collapse of error distributions toward short outputs. In contrast, **Our Method** adaptively identifies and modulates the reasoning process, selectively shortening reasoning chains when appropriate while preserving longer explorations necessary for challenging instances. Consequently, our approach mitigates overthinking without inducing underthinking, as evidenced by error distributions that avoid collapsing into shorter outputs.

### A.2 CONFIDENCE-LENGTH ASSOCIATION

Although confidence is commonly used as a proxy for a model's certainty, its connection to actual reasoning behavior remains ambiguous. To effectively utilize confidence for identifying suboptimal reasoning patterns or enabling adaptive control, it is essential to first establish a clear and quantifiable relationship between confidence and specific aspects of model behavior. Building upon the quantitative patterns presented in Fig. 2(b), which reveal distinct confidence signatures associated with overthinking and underthinking, we focus here on another key dimension of efficient reasoning: reasoning length.

Specifically, we measure how response length relates to step-level confidence on **MATH-500**, using four models (DeepSeek-R1-Distill-Qwen-1.5B, DeepSeek-R1-Distill-Qwen-7B, Qwen3-14B, QwQ-32B; each with $n = 500$ samples). Here, the length of a response is defined as the total number of words in the reasoning text before the first `</think>`, and this count is aligned with the list of confidence values for each step. For every answer, we compute two key quantities: (i) the *minimum* step-level confidence, and (ii) the *variance* of step-level confidence within the answer.

Since confidence values produced by non-greedy decoding tend to be heavily skewed toward the upper end of the interval $[0, 1]$ rather than following a balanced and symmetric bell-shaped distribu-

tion, the usual Pearson correlation is not suitable. Therefore, we use the nonparametric Spearman rank correlation to capture the relationship more reliably.

Our correlation analysis results are shown in Tab. 5. Across all four models, word count shows a clear negative association with the minimum step-level confidence (Spearman $\rho$ values roughly between $-0.62$ and $-0.80$), and a clear positive association with the variance of step-level confidence within each answer (Spearman $\rho$ values roughly between $0.46$ and $0.70$). For all reported entries, the 95% bootstrap confidence intervals do not include zero. These results provide quantitative evidence that confidence trajectories encode meaningful signals about reasoning effort and stability, supporting their use as reliable indicators for dynamic reasoning control.

| Model | Words vs Min confidence | | | Words vs Confidence variance | | |
|---|---|---|---|---|---|---|
| | $\rho$ | CI_lo | CI_hi | $\rho$ | CI_lo | CI_hi |
| DeepSeek-R1-Distill-Qwen-1.5B | $-0.733$ | $-0.777$ | $-0.684$ | $0.602$ | $0.542$ | $0.655$ |
| DeepSeek-R1-Distill-Qwen-7B | $-0.624$ | $-0.681$ | $-0.560$ | $0.458$ | $0.385$ | $0.527$ |
| QwQ-32B | $-0.801$ | $-0.835$ | $-0.765$ | $0.698$ | $0.643$ | $0.746$ |
| Qwen3-14B | $-0.681$ | $-0.730$ | $-0.628$ | $0.623$ | $0.561$ | $0.682$ |

**Table 5:** Spearman correlations ($\rho$) between word count length and confidence statistics on **MATH-500**. The 95% bootstrap percentile confidence intervals (CI_lo and CI_hi, with $B = 2000$) are reported in separate columns. All correlation coefficients are significant at $p < 0.001$.

### A.3 MARKOV PERSISTENCE OF CONFIDENCE STATES

Effective dynamic control over a model's reasoning trajectory often relies on the ability to recognize its current reasoning state. Intuitively, this requires examining a contextual window of recently generated reasoning steps, i.e., a sliding window over the chain-of-thought trace. However, for complex problems such as those in AIME (AI-MO, 2024a), reasoning traces can span thousands of tokens. While a larger window might seem necessary to capture sufficient context, it introduces significant computational overhead and may obscure fine-grained shifts in reasoning behavior.

In this section, we demonstrate that the model's confidence trajectory exhibits strong first-order Markov persistence. This finding reveals a key insight: the current reasoning state can be accurately inferred from just the immediately preceding step. Consequently, a minimal window of size two is sufficient and often preferable for capturing the essential dynamics of the reasoning process.

To formalize this, for each answer, we split the model outputs by double newlines (\n\n) and align the resulting segments with the sentence-level confidences; we only consider adjacencies that occur within complete, untruncated reasoning trajectories that contain the final answer. We then collect all adjacent confidence pairs $(c_{t-1}, c_t)$ from the model's reasoning traces.

We convert each confidence value into either a high state or a low state using the model-wise median threshold $\tau$:
$$s_t = \mathbb{I}(c_t \geq \tau), \qquad s_t \in \{H, L\}, \qquad \tau = \text{median}\{c_t \text{ over all sentences of the model}\}.$$
Here $H$ (high) represents $c_t \geq \tau$ and $L$ (low) represents $c_t < \tau$. If a confidence value equals the threshold, that sentence is placed in the high state.

For each model, we form a two-by-two transition count matrix
$$\mathbf{N} = \begin{bmatrix} HH & HL \\ LH & LL \end{bmatrix},$$
where $HH$ is the number of transitions from high to high and $HL$ is the number of transitions from high to low.

By normalizing each row, we obtain the corresponding transition probabilities:
$$P(H{\to}H) = \frac{HH}{HH + HL}, \qquad\qquad P(H{\to}L) = \frac{HL}{HH + HL},$$
$$P(L{\to}H) = \frac{LH}{LH + LL}, \qquad\qquad P(L{\to}L) = \frac{LL}{LH + LL}.$$

To measure how strongly a state tends to be followed by the same state, we use the odds ratio

$$\text{OR} = \frac{HH \cdot LL}{HL \cdot LH}.$$

We evaluate statistical significance using a two-sided Fisher exact test applied to $\mathbf{N}$. When all cells of the matrix are positive, we also report the approximate ninety-five percent confidence interval for the odds ratio (Woolf method):

$$\log(\widehat{\text{OR}}) \pm 1.96 \sqrt{\tfrac{1}{HH} + \tfrac{1}{HL} + \tfrac{1}{LH} + \tfrac{1}{LL}}, \qquad \text{CI}_{95\%} = \exp(\cdot).$$

For easier interpretation, we additionally provide the same state rate

$$\text{SameRate} = \frac{HH + LL}{HH + HL + LH + LL}.$$

All values reported in our results, including transition probabilities, odds ratios, confidence intervals, and significance levels, are computed using this median-based thresholding procedure.

From Tab. 6, all four models show clear evidence of strong like-to-like persistence in confidence when using the model-wise median threshold. The transition probabilities for remaining in the same state, $P(H{\to}H)$ and $P(L{\to}L)$, are both larger than the probabilities of switching to the opposite state. The overall same state rate satisfies $\text{SameRate} > 0.5$, the Fisher exact tests give $p < 0.001$, and the diagonal odds ratios are consistently greater than one with ninety-five percent confidence intervals that do not include one. Taken together, these results provide strong support for the presence of first-order Markov persistence, which reflects a clear tendency for the confidence state to remain stable from one step to the next.

| Model | $P(H{\to}H)$ | $P(L{\to}L)$ | SameRate | OR | CI_lo | CI_hi | Sig. |
|---|---|---|---|---|---|---|---|
| DeepSeek-R1-Distill-Qwen-1.5B | 0.666 | 0.665 | 0.666 | 3.96 | 3.83 | 4.10 | *** |
| DeepSeek-R1-Distill-Qwen-7B | 0.653 | 0.651 | 0.652 | 3.51 | 3.38 | 3.65 | *** |
| QwQ-32B | 0.670 | 0.673 | 0.672 | 4.19 | 4.03 | 4.35 | *** |
| Qwen3-14B | 0.657 | 0.659 | 0.658 | 3.70 | 3.56 | 3.86 | *** |

**Table 6:** Adjacent state persistence in step-level confidence using a *median* threshold for binarization. Rows report the same state transition probabilities and diagonal odds ratios (OR) with ninety-five percent Woolf confidence intervals. Significance codes (Sig.): * $p < 0.05$, ** $p < 0.01$, *** $p < 0.001$ (two sided Fisher exact test).

This observation directly guides the design of our sliding window. Based on this insight, we set the window size to $w = 2$ instead of a larger value. A window of length two records each pair of adjacent states and therefore captures the transition patterns $P(H{\to}L)$ and $P(L{\to}H)$ without any loss of information. This choice keeps detection lag to a minimum and prevents brief reversals from being averaged away. When the model begins to drift away from its current reasoning direction, for example, when it moves into an overthinking regime, the adjacent transition window allows the intervention strength to increase immediately.

## A.4 Distributional Heterogeneity in Model Confidence

In this study, our goal is to design a dynamic control function that leverages stepwise confidence and its variance during the reasoning process to enable real-time, adaptive regulation of model behavior. We aim to make this control mechanism highly adaptable, i.e., capable of fully realizing our proposed concept of balanced thinking and delivering accurate, smooth, and responsive control. However, greater adaptability inherently involves introducing additional parameters, which raises concerns about the generalization ability of such a control function across diverse models. To address this, we analyze and compare confidence distributions across multiple models to investigate whether a universal hyperparameter configuration can be identified.

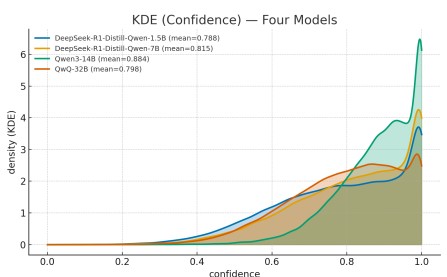

**Figure 7:** KDE (Confidence) for QwQ-32B, Qwen3-14B, and DeepSeek-R1-Distill-Qwen-7B and 1.5B on MATH-500.

As shown in Fig. 7, we visualize confidence distributions across reasoning steps for multiple models on the same dataset. The three models based on the QWEN2 family, namely QWQ-32B, DEEPSEEK-R1-DISTILL-QWEN-7B, and DEEPSEEK-R1-DISTILL-QWEN-1.5B, display broadly similar distributional patterns, although each model still exhibits its own characteristic shape. In contrast, the QWEN3-14B model, built upon the QWEN3 family, exhibits a clearly different confidence profile compared with the QWEN2 family. These observations are consistent with previous findings that the LLAMA-3.1-NEMOTRON-NANO-8B model tends to operate in a uniformly low confidence regime throughout its reasoning process (Yang et al., 2025b).

These distributional differences highlight the difficulty of designing a single set of hyperparameters capable of effectively generalizing across various models. Consequently, this motivates our proposed *model behavior-based dynamic control function fitting* approach in Sec. 3.4, which automatically derives parameters tailored to the unique confidence behaviors of each model. By leveraging this behavior-aware strategy, our method eliminates the need for manual hyperparameter tuning, thereby ensuring robust adaptability and broad applicability across diverse model architectures and confidence profiles.

## A.5 Encoding of Confidence Signals in Latent Representations

To enable dynamic control over a model's reasoning behavior through confidence-aware steering, it is crucial to understand how confidence manifests in the model's internal representations. Since hidden states directly encode the evolving behavioral dynamics of a transformer during reasoning, they provide a natural substrate for both analyzing and manipulating the model's certainty. In this section, we demonstrate that confidence is not merely correlated with, but systematically and predominantly linearly encoded in hidden states. This insight underpins an automated approach for identifying the most effective layers to target in steering interventions.

**Confidence is discernible in hidden layers.** From Eq. (1), we obtain stepwise confidence values. We also extract the hidden state $\mathbf{H}_s^{(i)}$ of the token following the delimiter \n\n. This gives rise to a direct mapping between hidden state and confidence:

$$\mathbf{H}_s^{(i)} \longmapsto c_s.$$

Thus, the layer-$i$ hidden state for sentence $s$ corresponds to its confidence $c_s$. As shown in Fig. 8, we apply t-SNE to project the hidden states $\mathbf{H}_s^{(i)}$ into a two-dimensional space and color each point according to its corresponding confidence $c_s$. Clear clusters emerge: high-confidence and low-confidence representations form visibly distinct regions, with this separation becoming even more pronounced in the mid-late layer embeddings. These observations indicate that confidence acts as a discernible signal in the hidden space, providing direct empirical support for our subsequent linear probing analysis.

**Linear probing of the confidence signal.** Building on the above observations, we employ a linear probing approach to examine how confidence is encoded in the hidden layers and to assess the

strength of this linear relationship. Formally, the mapping is expressed as:

$$c_s = \mathbf{w}^{(i)} \mathbf{H}_s^{(i)} + b^{(i)}.$$

Here, $\mathbf{w}^{(i)}$ and $b^{(i)}$ denote the parameters of the linear model.

Because hidden layer representations typically have several thousand dimensions, using a linear head directly may lead to overfitting and unstable estimation. To address this, we apply a standard dimensionality reduction method, PCA, to project the hidden states into a low-dimensional subspace, and then study the relationship between the reduced representations and the corresponding confidence values. The detailed statistics before and after PCA are summarized in Tab. 7.

$$c_s = \mathbf{w}^{(i)} \mathbf{Z}_s^{(i)} + b^{(i)}, \qquad \mathbf{H}_s^{(i)} \longmapsto \mathbf{Z}_s^{(i)}.$$

Here, $\mathbf{Z}_s^{(i)}$ denotes the low-dimensional representation obtained from the original hidden state $\mathbf{H}_s^{(i)}$.

The overall probe analysis pipeline is illustrated in Fig. 10. We employ a standard ridge regression approach to estimate the parameters $\mathbf{w}^{(i)}$ and $b^{(i)}$.

$$\min_{\mathbf{w}^{(i)}, b^{(i)}} \sum_s \left( c_s - \mathbf{w}^{(i)} \mathbf{Z}_s^{(i)} - b^{(i)} \right)^2 + \lambda \left\| \mathbf{w}^{(i)} \right\|_2^2.$$

After fitting the ridge-based linear probe, we obtain the predicted confidence values as follows.

$$\hat{c}_s = \mathbf{w}^{(i)} \mathbf{Z}_s^{(i)} + b^{(i)}.$$

We assess how accurately confidence can be predicted from the hidden representations by computing the coefficient of determination $R^2$. The formulation is given below. A higher $R^2$ value, approaching 1, indicates that confidence is more readily linearly decodable from the representations of the corresponding layer.

$$R^2 = 1 - \frac{\sum_s \left( c_s - \hat{c}_s \right)^2}{\sum_s \left( c_s - \bar{c} \right)^2}.$$

**Automated steering layer selection using $R^2$.** We evaluate the relationship between the confidence values $c_s$ and the hidden state $\mathbf{H}_s^{(i)}$ across all layers of each model, as shown in Fig. 9. A consistent pattern emerges: the coefficient of determination $R^2$ typically reaches its maximum in the middle to late layers, indicating that $c_s$ is more easily linearly decodable from $\mathbf{H}_s^{(i)}$ in these layers. Since steering fundamentally operates through linear shifts in the hidden space, we select the layer with the highest $R^2$ as the steering layer. The entire procedure is fully automated, allowing the system to identify the optimal steering layer without manual intervention. In principle, choosing this layer minimizes the additional noise introduced by the steering operation.

| Model | Original Dim | PCA Dim | Retained Var. |
|---|---|---|---|
| DeepSeek-R1-Distill-Qwen-1.5B | 1 536 | 64 | 0.889 3 |
| DeepSeek-R1-Distill-Qwen-7B | 3 584 | 64 | 0.857 3 |
| QwQ-32B | 5 120 | 64 | 0.833 5 |
| Qwen3-14B | 5 120 | 64 | 0.900 0 |

**Table 7:** Cumulative explained variance retained by PCA ($k = 64$) across models. Higher retained variance indicates a stronger low-dimensional linear structure amenable to probing.

## A.6 CONFIDENCE AS EVIDENCE UNDER VOCABULARY COVERAGE GAPS

A growing number of efficient reasoning methods mitigate overthinking by relying on predefined keyword vocabularies. Representative strategies include *targeted suppression* of specific tokens (e.g., NOWAIT (WANG ET AL., 2025A)), *latent-space guidance* that steers the model away from

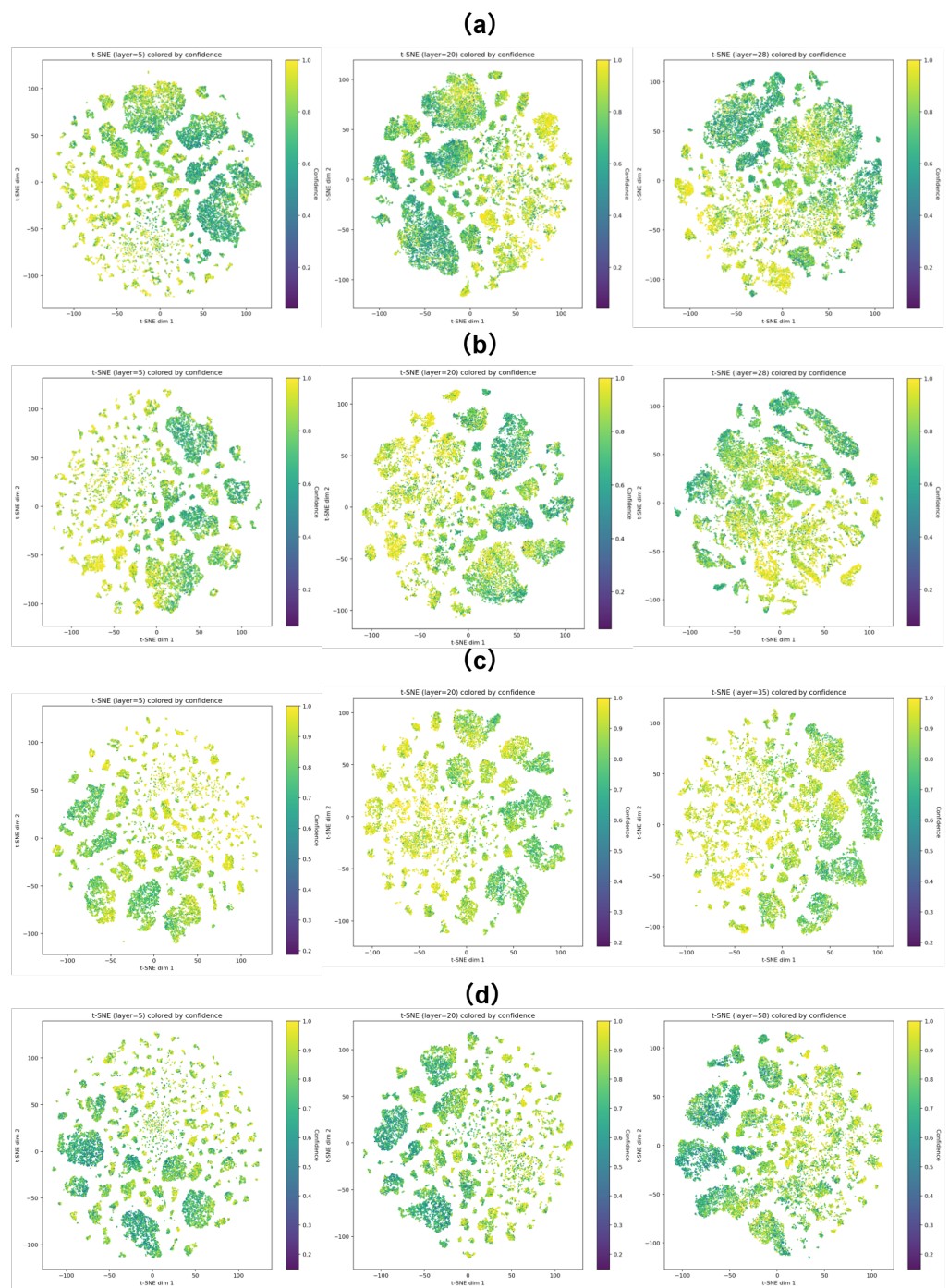

**Figure 8:** t-SNE projections of hidden states sampled immediately after \n\n; colors encode sentence-level confidence (0–1). (a) DeepSeek-R1-Distill-Qwen-1.5B — layers 5, 20, 28. (b) DeepSeek-R1-Distill-Qwen-7B — layers 5, 20, 28. (c) Qwen3-14B — layers 5, 20, 35. (d) QwQ-32B — layers 5, 20, 58.

states prone to emitting certain tokens (e.g., SEAL (Chen et al., 2025b), MANIFOLD STEER-ING (Huang et al., 2025b)), and *cue-word–driven early stopping*, which treats designated trigger terms as checkpoints to terminate the reasoning process when appropriate (e.g., FLASHTHINK-ING (JIANG ET AL., 2025), TRIMR (LIN ET AL., 2025A), DEER (YANG ET AL., 2025B),

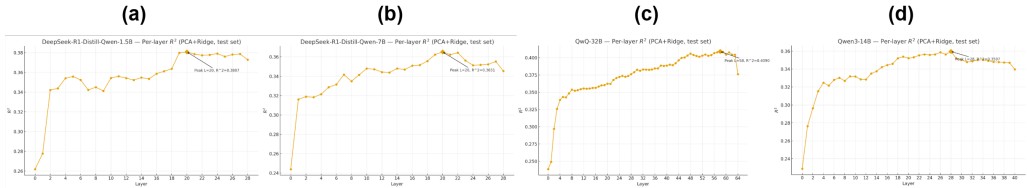

**Figure 9:** Layer-wise linear probe decodability ($R^2$) of confidence. Mid-to-late layers achieve the highest scores, motivating the steering-layer choice.

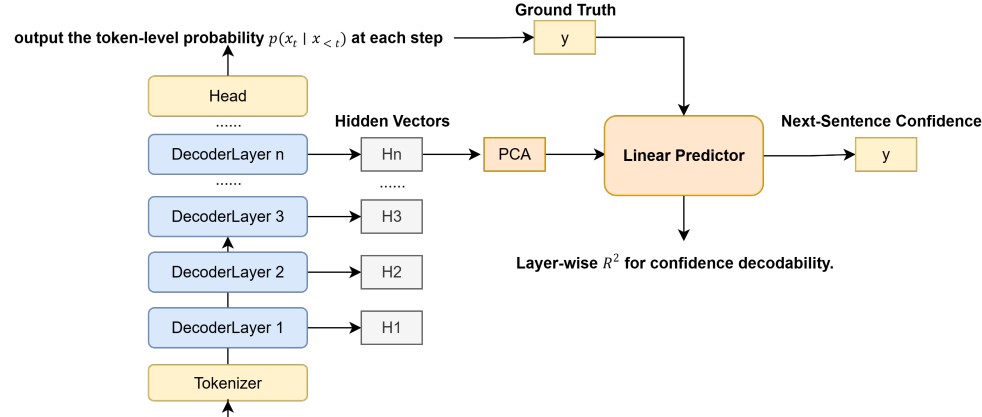

**Figure 10: Linear-confidence probe with PCA.** Given an input sequence, we freeze the language model (LM) and extract layer-wise hidden states. The representations are projected by PCA and passed to a linear predictor to estimate step-level confidence. We repeat this procedure across layers to obtain layer-wise test-set $R^2$ scores.

DYNASOR-COT (FU ET AL., 2025)). However, the fundamental reason why such lexical interventions improve reasoning behavior remains poorly understood. In this section, we aim to uncover the mechanism behind their effectiveness by aggregating keyword vocabularies from representative methods and analyzing their relationship with model confidence. Our key finding is that vocabulary-based strategies are, in essence, incomplete approximations of confidence-based control: they capture only the most frequent lexical manifestations of low-confidence reasoning, while missing a broader spectrum of uncertainty signals.

| Category | Vocabulary |
|---|---|
| **NoWait (suppress)** | `wait`, `alternatively`, `hmm`, `but`, `however`, `alternative`, `another`, `check`, `double-check`, `oh`, `maybe`, `verify`, `other`, `again`, `now`, `ah`, `any` |
| **SEAL—Transition** | `alternatively`, `think differently`, `another way`, `another approach`, `another method`, `another solution`, `another strategy`, `another technique` |
| **SEAL—Reflection** | `wait`, `verify`, `make sure`, `hold on`, `think again`, `'s correct`, `'s incorrect`, `let me check`, `seems right` |

**Table 8:** Unified vocabularies for NoWait (keyword suppression) and SEAL transition/reflection cues.

To investigate this, we first compile the keyword sets used by NOWAIT and SEAL as illustrative examples (see Tab. 8). Notably, these lexical items function as surface markers of the model's epistemic uncertainty. Using DEEPSEEK-R1-DISTILL-QWEN-7B on MATH-500, we compute a sentence-level confidence at each reasoning step and project it to all words appearing in that

sentence. As summarized in Tab. 9, the vast majority of these words are associated with confidences below the model's overall mean ($\bar{c} = 0.8162$). This pattern suggests a straightforward interpretation: vocabulary-based interventions such as NOWAIT and SEAL primarily suppress *low-confidence modes* of the model's reasoning, rather than targeting particular semantics per se.

As illustrated in Fig. 11, both SEAL and NoWait reliably elevate the model's confidence along the reasoning trajectory. This empirical pattern corroborates our analysis: NOWAIT achieves the effect by suppressing the emission of high-frequency lexical markers associated with low confidence, whereas SEAL steers the hidden representations away from states that tend to produce low-confidence, high-frequency sentences. In essence, both methods act by attenuating the model's low-confidence modes.

However, vocabulary-driven heuristics do not, by themselves, capture the model's *low-confidence modes*. In practice, such methods identify only a subset of *high-frequency lexical correlates* of low confidence, leaving a substantial long-tail of equally informative cues outside the predefined lists and thus unmeasured. As illustrated in Tab. 9, we enumerate several representative omissions that most existing approaches fail to account for. Consequently, confidence-based approaches systematically surface the model's low-confidence modes—irrespective of their lexical realization.

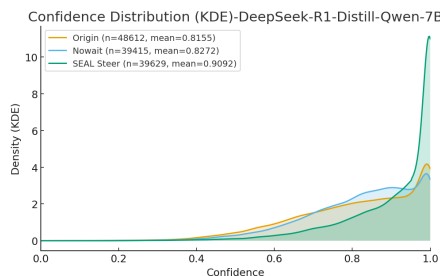

**Figure 11:** KDE (Confidence) - Origin vs NoWait vs SEAL.

This comprehensive extraction provides a principled basis for subsequent research to diagnose and mitigate overthinking, enabling more complete coverage than vocabulary-driven heuristics. Looking forward, a fruitful research agenda is to pursue a confidence-based line of work. One direction is to treat low-confidence states as actionable checkpoints for early exit, developing calibrated criteria and adaptive halting policies to further improve the accuracy–efficiency trade-off of early-exit models. Another is to analyze the relationship between semantic (meaning-level) uncertainty and model-internal confidence estimates, thereby deepening our understanding of—and ultimately mitigating—both overthinking and underthinking behaviors.

## B METHOD DETAILS

In this section, we provide a more detailed introduction to the technical details of REBALANCE presented in Sec. 3. First, in Appendix B.1, we formally define these reasoning modes and propose an explicit, confidence-based modeling paradigm to quantitatively distinguish between redundant reasoning and premature conclusion. Building upon this foundation, Appendix B.2 presents the *Confidence-Based Steering Vector Extraction*, where we leverage hidden-state representations to derive directional steering vectors that guide LRMs' reasoning trajectory towards optimal decision-making boundaries. Finally, Appendix B.3 details our *Model Behavior-Based Dynamic Control Function*, which dynamically modulates steering strength according to real-time confidence and variance metrics. By integrating these three sequential components, our framework achieves a robust and adaptive control mechanism, effectively balancing exploration and commitment in the reasoning processes of LRMs.

### B.1 EXPLICIT MODELING OF OVERTHINKING AND UNDERTHINKING

**Formal Definition.** Let the reasoning trajectory inside `<think>...</think>` be split into steps $S_1, \ldots, S_{s_{\max}}$ by the double newline delimiter $\backslash n \backslash n$ introduced in Sec. 2.1. Let $r_{\leq s}$ denote the partial reasoning up to step $s$. If the model is forced to stop after step $s$ and produce a conclusion from $r_{\leq s}$, it induces a distribution over answers which we denote by $\pi_s$. Let $d_s = \arg\max \pi_s$ be the predicted conclusion under a fixed decoding rule. Define the stability index

$$s^\star = \min\big\{ s : d_{s'} = d_s \text{ for all } s' \geq s \text{ and } d_s \text{ is correct}\big\}.$$

**Table 9:** Low-confidence lexicon (items from the curated vocabulary are marked with * and colored red).

| Block A | | | Block B | | | Block C | | | Block D | | |
| --- | --- | --- | --- | --- | --- | --- | --- | --- | --- | --- | --- |
| Word/Phrase | Confidence | Count | Word/Phrase | Confidence | Count | Word/Phrase | Confidence | Count | Word/Phrase | Confidence | Count |
| think differently* | 0.6067 | 49 | alternative* | 0.6375 | 13 | think again* | 0.6537 | 115 | another approach* | 0.6549 | 209 |
| sorry | 0.5990 | 1 | confusing | 0.6620 | 71 | confused | 0.6690 | 101 | maybe* | 0.6740 | 3,012 |
| another method* | 0.6781 | 78 | another way* | 0.6821 | 417 | alternatively* | 0.6844 | 1,857 | hold on* | 0.6881 | 491 |
| however* | 0.6914 | 68 | i think | 0.6920 | 866 | in summary | 0.6940 | 10 | i suspect | 0.6950 | 1 |
| verify (variants)* | 0.6971 | 354 | verify* | 0.6988 | 339 | any* | 0.7115 | 729 | i'm not sure | 0.7040 | 60 |
| let me think | 0.7050 | 647 | i guess | 0.7120 | 43 | it seems | 0.7120 | 101 | probably | 0.7120 | 77 |
| but* | 0.7092 | 6,866 | double-check* | 0.7161 | 339 | not sure | 0.7180 | 154 | Let me check* | 0.7223 | 384 |
| check* | 0.7228 | 1,093 | other* | 0.7290 | 776 | again* | 0.7347 | 734 | wait* | 0.7325 | 2,223 |
| anyway | 0.7270 | 6 | another solution* | 0.7357 | 10 | possibly | 0.7490 | 2 | hmm* | 0.7710 | 1,728 |
| i believe | 0.7700 | 11 | 's correct* | 0.7821 | 288 | wow | 0.7820 | 13 | 's incorrect* | 0.7821 | 288 |
| it looks like | 0.7830 | 17 | seems right* | 0.7919 | 92 | phew | 0.7900 | 1 | now* | 0.8182 | 1,734 |
| oh* | 0.8187 | 40 | mandatory | 0.5948 | 5 | easiest | 0.5948 | 6 | perspective | 0.5977 | 8 |
| summarizing | 0.4997 | 14 | rely | 0.5066 | 10 | overlooked | 0.5093 | 6 | systematically | 0.5342 | 5 |
| uses | 0.5359 | 5 | partially | 0.5389 | 11 | layer | 0.5447 | 5 | reason | 0.5461 | 5 |
| uniquely | 0.5468 | 7 | summed | 0.5470 | 7 | separate | 0.6460 | 75 | annual | 0.5602 | 5 |
| trick | 0.5612 | 5 | misinterpret | 0.5649 | 5 | generating | 0.5702 | 9 | crucial | 0.5759 | 6 |
| substitutions | 0.5763 | 8 | consist | 0.5797 | 7 | systems | 0.5809 | 10 | neat | 0.5809 | 5 |
| despite | 0.5811 | 9 | careful | 0.5817 | 21 | shake | 0.5818 | 10 | redo | 0.5819 | 8 |
| clear | 0.5859 | 29 | accept | 0.5866 | 15 | discriminants | 0.5869 | 6 | haven | 0.5871 | 18 |
| quickly | 0.5895 | 5 | diametrically | 0.5899 | 5 | misapplied | 0.5914 | 8 | homogeneous | 0.5937 | 6 |
| extends | 0.5979 | 5 | consuming | 0.6028 | 7 | obvious | 0.6033 | 7 | altered | 0.6038 | 6 |
| worried | 0.6046 | 7 | periodicity | 0.6055 | 11 | theory | 0.6067 | 10 | verification | 0.6072 | 11 |
| interpreting | 0.6073 | 11 | absolutely | 0.6076 | 12 | translation | 0.6082 | 5 | schedule | 0.6098 | 5 |
| elsewhere | 0.6100 | 9 | designed | 0.6123 | 8 | shapes | 0.6148 | 10 | may | 0.6150 | 15 |
| interpolation | 0.6159 | 7 | hard | 0.6162 | 8 | necessary | 0.6162 | 42 | unfolding | 0.6164 | 27 |
| concerning | 0.4954 | 2 | uncertain | 0.5537 | 1 | tangled | 0.6371 | 6 | mentally | 0.6378 | 10 |
| surprised | 0.6059 | 1 | relief | 0.6126 | 1 | allows | 0.6379 | 9 | mix | 0.6380 | 31 |
| doubt | 0.6559 | 1 | unsure | 0.6699 | 1 | solidify | 0.6394 | 5 | arrive | 0.6401 | 10 |
| concerned | 0.7315 | 3 | surprising | 0.7318 | 5 | precisely | 0.6407 | 13 | effectively | 0.6413 | 30 |
| nervous | 0.8295 | 1 | verifying | 0.6416 | 13 | mixing | 0.6416 | 11 | offset | 0.6416 | 12 |
| triplets | 0.6420 | 7 | complicate | 0.6422 | 38 | extensions | 0.6426 | 10 | handshake | 0.6427 | 8 |
| version | 0.6430 | 8 | eighteen | 0.6432 | 16 | abstract | 0.6433 | 9 | paper | 0.6433 | 5 |
| nature | 0.6435 | 7 | clarify | 0.6442 | 44 | shaking | 0.6442 | 7 | overcomplicating | 0.6442 | 45 |
| exploit | 0.6443 | 5 | cell | 0.6444 | 5 | safe | 0.6444 | 13 | interpreted | 0.6445 | 9 |
| pyramid | 0.6446 | 6 | assumptions | 0.6448 | 6 | crossing | 0.6451 | 7 | ... | ... | ... |

A trajectory exhibits *overthinking* if it continues generating steps after $s^\star$. Conversely, A trajectory exhibits *underthinking* if it stops at step $s$ with an incorrect $d_s$ while there exists $s' > s$ such that $d_{s'}$ would be correct. These definitions capture redundant reasoning beyond the earliest stable correct decision and premature commitment before sufficient exploration.

**Explicit Modeling with Confidence.** We instantiate the above definition using the sequence of stepwise confidence $\{c_s\}$ and confidence variance $\{v_s\}$ as defined in Sec. 2.1, where $v_s = \text{Var}(c_s; \mathcal{W}_s)$ and $\mathcal{W}_s$ is a sliding window. We can determine two-sided quantile thresholds (Hyndman & Fan, 1996) from a small-scale seen dataset. Let $Q_c(q)$ and $Q_v(q)$ be the empirical $q$-quantile of $\{c_s\}$ and $\{v_s\}$. Choose $0 < q_L < q_H < 1$ and define

$$\tau_c^{\text{L}} = Q_c(q_L), \quad \tau_c^{\text{H}} = Q_c(q_H), \qquad \tau_v^{\text{L}} = Q_v(q_L), \quad \tau_v^{\text{H}} = Q_v(q_H).$$

A step is tagged as *low-confidence* if $c_s \leq \tau_c^{\text{L}}$ and *high-confidence* if $c_s \geq \tau_c^{\text{H}}$. Similarly, A step is tagged as *high-variance* if $v_s \geq \tau_v^{\text{H}}$ and *low-variance* if $v_s \leq \tau_v^{\text{L}}$. We then define the sets

$$\mathcal{O} \leftarrow \{ s : c_s \leq \tau_c^{\text{L}} \text{ and } v_s \geq \tau_v^{\text{H}} \}, \qquad \mathcal{U} \leftarrow \{ s : c_s \geq \tau_c^{\text{H}} \text{ and } v_s \leq \tau_v^{\text{L}} \}.$$

As observed in Fig. 2(b), high variance reflects frequent switching across reasoning paths and often co-occurs with low confidence; thus, we treat $\mathcal{O}$ as a proxy for overthinking. Persistently high confidence with low variance indicates stable yet potentially premature commitment, making $\mathcal{U}$ a proxy for underthinking. Steps that fall outside both sets are considered to reflect a normal state and are excluded from subsequent analyses.

## B.2    CONFIDENCE-BASED STEERING VECTOR EXTRACTION

Building upon the explicit modeling paradigm, we propose deriving steering vectors from deep-layer hidden representations to guide LRMs away from undesirable reasoning modes. These vectors are efficiently obtained via a one-pass collection performed only once per model prior to deployment, eliminating additional computation during actual use.

**One-pass prototype extraction.** We prepare a small-scale seen dataset $\mathcal{D}_{\text{seen}}$ and run the model once per prompt. When the model generates a delimiter $\backslash n \backslash n$, the next token marks the first token of a new step. At this token, we save deep-layer hidden states $\mathbf{h}_{t_s^{(1)}}$ for step index $s$ and select layers $\ell$, chosen via a probing method maximizing confidence separability on a single dataset but shared across all datasets (see Appendix A.5). The first token of a step serves as a compact representation of the step mode for two reasons. First, it typically encodes the intent that sets the direction of the step (e.g., *wait* or *alternatively*)(Yang et al., 2025b), and due to the causal mask, all later tokens in the step condition on it. Second, deep layers show stronger distinguishability between the two reasoning modes in our empirical study, consistent with Gekhman et al. (2025); Skean et al. (2025).

Using the tags from sets $\mathcal{O}$ and $\mathcal{U}$, we form latent distributions for the overthinking and underthinking modes. Then, we can obtain mode prototypes by averaging the extracted hidden states

$$\boldsymbol{\mu}_\ell^{\text{O}} = \frac{1}{|\mathcal{O}|} \sum_{s \in \mathcal{O}} \mathbf{h}_{t_s^{(1)}}, \qquad \boldsymbol{\mu}_\ell^{\text{U}} = \frac{1}{|\mathcal{U}|} \sum_{s \in \mathcal{U}} \mathbf{h}_{t_s^{(1)}}.$$

**Steering vector construction.** The difference between the two prototypes defines a steering vector

$$\mathbf{v} = \frac{\boldsymbol{\mu}^{\text{O}} - \boldsymbol{\mu}^{\text{U}}}{\|\boldsymbol{\mu}^{\text{O}} - \boldsymbol{\mu}^{\text{U}}\|_2}.$$

This vector encodes the transition direction in latent space from underthinking toward overthinking, with its negation representing the reverse.

During inference, we inject the steering vector solely at each step's first token. Specifically, we modify the deep hidden state by

$$\tilde{\mathbf{h}}_{t_s^{(1)}} = \mathbf{h}_{t_s^{(1)}} + \alpha_s \mathbf{v}.$$

Let $t_s^{(1)}$ be its position and let $\alpha_s \in \mathbb{R}$ be a scalar steering weight that controls strength $\lambda_s$ and direction $\delta_s$ at step $s$.

$$\alpha_s = \lambda_s \delta_s, \qquad \lambda_s \geq 0, \ \delta_s \in \{+1, -1\}.$$

Setting $\delta_s = +1$ pushes the state away from underthinking and increases exploration. Setting $\delta_s = -1$ pushes the state away from overthinking and facilitates the model to coverage to a reasonable reasoning path. The values of $\lambda_s$ and $\delta_s$ will be determined by the dynamic control function introduced in the later section. Conceptually, the two prototypes act as boundaries of the model's reasoning process. Our goal is to keep the stepwise state between these boundaries so that the model reasons efficiently with balanced thinking.

### B.3 MODEL BEHAVIOR-BASED DYNAMIC CONTROL FUNCTION

Inputs differ in difficulty, and the model's reasoning state evolves over time. To keep the trajectory between the overthinking and underthinking boundaries, we set the steering weight $\alpha$ online as a continuous function of the current state. The function takes the stepwise confidence $c_s$ and the confidence variance $v_s$ as inputs, and outputs a steering weight $\alpha$ that determines both direction $\delta$ and magnitude $\lambda$. The weight pushes the state away from the closer boundary and grows as the state approaches that boundary.

**From a confidence curve to a control surface.** To derive this three-dimensional surface, we first construct a simplified two-dimensional curve $f(c_s)$ based solely on stepwise confidence $c_s$. From the previous analysis, the steering weight $\alpha_s$ needs to transition smoothly from a minimum negative value (away from overthinking) to a maximum positive value (away from underthinking) as confidence $c_s$ increases. Many functions satisfy this requirement. Here, we adopt the widely used sigmoid as an illustration.

For ease of spatial transformation, we express the sigmoid function in terms of the hyperbolic tangent:

$$\sigma(c_s) = \frac{1}{1 + e^{-c_s}} = \frac{1}{2} + \frac{1}{2} \tanh\left(\frac{c_s}{2}\right).$$

Our goal is to spatially transform this sigmoid to align precisely with the overthinking and underthinking boundaries. After transformation, the function becomes:

$$f(c_s) = a + b \tanh(k(c_s + m)),$$

where $a$, $b$, $k$, and $m$ represent spatial transformation parameters, which can be obtained by fitting.

However, as detailed in Appendix A.4, confidence distributions vary significantly across models, making it difficult to find universally applicable parameters. Thus, we propose a *model behavior-based* fitting method. This method adaptively determines these parameters based on model-specific behavior, using the previously collected stepwise confidence $c_s$ and confidence variance $v_s$ from the one-pass prototype extraction without additional computational cost.

Specifically, after the one-pass extraction, we obtain hidden-state distributions and corresponding prototypes for overthinking and underthinking, from which we derive a steering vector $\mathbf{v}$. Since the steering vector connects prototypes that represent their respective hidden-state distributions, the steering strength can be interpreted as the displacement of these distributions along the vector direction. Therefore, adjusting the magnitude of this displacement enables us to capture specific behavioral characteristics of the model, allowing tailored data point generation.

To illustrate this, consider first the alleviation of overthinking. Suppose the hidden-state distribution of overthinking is bounded. The aggressive displacement is defined as the minimal distance required

to shift all points within this distribution beyond its boundary. A more moderate displacement, however, moves only the overthinking prototype outside the boundary, defining the moderate distance. We explicitly anchor these through specific behavioral criteria:

- Anchor A1: At $c_s = \tau_c^{\mathrm{L}}$, the steering should yield a negative moderate displacement, effectively guiding the state away from overthinking.
- Anchor A2: At $c_s = \tau_c^{\mathrm{H}}$, the steering weight is set to zero, as the state is considered to lie within the normal confidence region, and thus no additional steering is applied.

In contrast, mitigating underthinking poses unique challenges. Experimental observations indicate that LRMs may also consistently exhibit high confidence during normal reasoning, making direct numerical quantification of overconfidence, a defining characteristic of underthinking tendency, difficult. Therefore, we adopt a conservative mitigation approach. Recognizing that greater dispersion in confidence distributions corresponds to larger distances between prototypes measured by the norm of the steering vector, we select aggressive and moderate displacements for underthinking as small proportional fractions of this norm. This proportional strategy reduces underthinking without adversely affecting normal reasoning and ensures scalability across diverse LRMs' confidence distributions. This approach is anchored by:

- Anchor A3: At $c_s = 1$, the maximum normalized confidence, the steering provides a positive moderate displacement to mitigate excessive confidence and counteract underthinking.

We obtain these moderate targets from model behavior in latent space. Let prototype distance $d^{\mathrm{prot}} = \|\boldsymbol{\mu}^{\mathrm{O}} - \boldsymbol{\mu}^{\mathrm{U}}\|_2$ and let $s_s = \mathbf{v}^{\top} \mathbf{h}_{t_s^{(1)}}$, where $\mathbf{h}_{t_s^{(1)}}$ denotes the hidden state of the first token generated at reasoning step $s$, encoding the reasoning state of that step. Thus, $s_s$ represents the coordinate of the step's reasoning state projected onto the steering vector $\mathbf{v}$. Define a separating threshold along $\mathbf{v}$ by $t = \frac{1}{2} \mathbf{v}^{\top} (\boldsymbol{\mu}^{\mathrm{O}} + \boldsymbol{\mu}^{\mathrm{U}})$. The *moderate* distance for overthinking is

$$d^{\mathrm{O,m}} = \mathbf{v}^{\top} \boldsymbol{\mu}^{\mathrm{O}} - t,$$

which moves the overthinking prototype to the boundary. The *aggressive* distance for overthinking is

$$d^{\mathrm{O,a}} = \max_{s \in \mathcal{O}} (s_s) - t,$$

which moves all overthinking steps past the boundary. For underthinking, we adopt a conservative rule since normal reasoning can also show sustained high confidence. We set

$$d^{\mathrm{U,m}} = \rho_{\mathrm{U}}^{\mathrm{m}} \, d^{\mathrm{prot}}, \qquad d^{\mathrm{U,a}} = \rho_{\mathrm{U}}^{\mathrm{a}} \, d^{\mathrm{prot}},$$

with constants $0 < \rho_{\mathrm{U}}^{\mathrm{m}} < \rho_{\mathrm{U}}^{\mathrm{a}}$ that are fixed and identical across all models. These distances scale with the separation between prototypes and thus naturally adapt to different models. Because $\mathbf{v}$ has unit norm, a displacement of size $d$ along $\mathbf{v}$ corresponds to a steering magnitude $d$ in the hidden space. We therefore fit $f$ so that

$$f(\tau_c^{\mathrm{L}}) = -d^{\mathrm{O,m}}, \qquad f(\tau_c^{\mathrm{H}}) = 0, \qquad f(1) = +d^{\mathrm{U,m}}.$$

We solve for $b$ and $k$ by least squares on these anchors. This yields a smooth curve that produces negative weights at low confidence and positive weights at high confidence, with zero at the center.

**Lifting to a variance aware surface.** We denote by $B$ the mode boundary that determines the maximal steering amplitude. Three distinct mode boundaries are employed. $B_m$ is the moderate boundary used outside the high-risk regions and is determined by the moderate fitting targets of $f$. $B_o$ and $B_u$ are the boundaries for overthinking and underthinking regions. Because the aggressive distances already encode how far the risky states extend from the boundary along $\mathbf{v}$, we set

$$B_o = d^{\mathrm{O,a}}, \qquad B_u = d^{\mathrm{U,a}}.$$

Using aggressive distances as boundaries provides a worst-case correction inside each high-risk region. For overthinking, $d^{\mathrm{O,a}}$ is defined from the most extreme detected overthinking step, so steering with this magnitude is sufficient to push all detected overthinking states back to the boundary. For underthinking, we use the conservative construction of $d^{\mathrm{U,a}}$ to reduce premature commitment while limiting disruption to normal high-confidence reasoning.

We now incorporate variance to obtain a two-input control $g(c_s, v_s)$. The idea is to keep the sign and basic shape from $f(c_s)$ while increasing the magnitude near the two high-risk regions. The overthinking region is $c_s \leq \tau_c^{\mathrm{L}}$ with $v_s \geq \tau_v^{\mathrm{H}}$. The underthinking region is $c_s \geq \tau_c^{\mathrm{H}}$ with $v_s \leq \tau_v^{\mathrm{L}}$. However, abrupt steering strength changes at region boundaries are undesirable. To mitigate this, we define smooth gates that approach one inside each region and decay to zero outside

$$\psi_{\mathrm{O}}(c_s, v_s) = \sigma\left(\frac{\tau_c^{\mathrm{L}} - c_s}{\eta_c}\right) \sigma\left(\frac{v_s - \tau_v^{\mathrm{H}}}{\eta_v}\right), \qquad \psi_{\mathrm{U}}(c_s, v_s) = \sigma\left(\frac{c_s - \tau_c^{\mathrm{H}}}{\eta_c}\right) \sigma\left(\frac{\tau_v^{\mathrm{L}} - v_s}{\eta_v}\right),$$

where $\sigma(x) = 1/(1 + e^{-x})$. The parameters $\eta_c > 0$ and $\eta_v > 0$ control the width of the transition and smooth the change near region boundaries.

We then interpolate between the moderate boundary and the mode-specific boundaries using the smooth gates:

$$B(c_s, v_s) = B_m + (B_o - B_m)\, \psi_{\mathrm{O}}(c_s, v_s) + (B_u - B_m)\, \psi_{\mathrm{U}}(c_s, v_s).$$

The final control surface is

$$g(c_s, v_s) = \mathrm{sign}(c_s - \tau_c^{\mathrm{H}})\, B(c_s, v_s)\, \tanh\left(\left|c_s - \tau_c^{\mathrm{H}}\right|\right).$$

For fixed $v_s$, the map is monotone in $c_s$. For fixed $c_s$, the magnitude increases smoothly as $v_s$ enters either high-risk region. The sign follows the confidence side so that the steering pushes away from the nearer mode boundary.

**Online steering.** At step $s$ we set

$$\alpha_s = g(c_s, v_s), \qquad \lambda_s = |\alpha_s|, \qquad \delta_s = \mathrm{sign}(\alpha_s).$$

These values plug into the injection rule and separate direction and magnitude as defined earlier. The procedure is training-free and uses only statistics that are already computed online. It adapts across models through the behavior-based distances and across inputs through the gates on $(c_s, v_s)$. The result is a continuous controller that keeps the trajectory between the two mode boundaries and allocates more steering when the state drifts toward either boundary.

**Function Visualization.** We visualize the fitted mapping $g(c_s, v_s)$ for DEEPSEEK–R1–DISTILL–QWEN–1.5B. Darker regions—characterized by *high variance* and *low confidence*—indicate where the *overthinking penalty* is strongest. Conversely, the lighter, positive region at *low variance* and *high confidence* marks where the *underthinking penalty* is strongest.

## C  ADDITIONAL EXPERIMENTAL RESULTS AND ABLATIONS

### C.1  ABLATION ON INDIVIDUAL AXES

We study univariate effects via axis-wise ablations of $g(c_s, v_s)$: define $g_c(c_s) := g(c_s, \bar{v})$ and $g_v(v_s) := g(\bar{c}, v_s)$, where $\bar{v}$ and $\bar{c}$ denote the means of confidence variance $v$ and stepwise confidence $c$ estimated on the extraction set. As shown in the first three rows of Tab. 10, both univariate variants degrade REBALANCE performance, indicating that the bivariate form $g(c, v)$ provides finer-grained and more effective control than single-variable schemes.

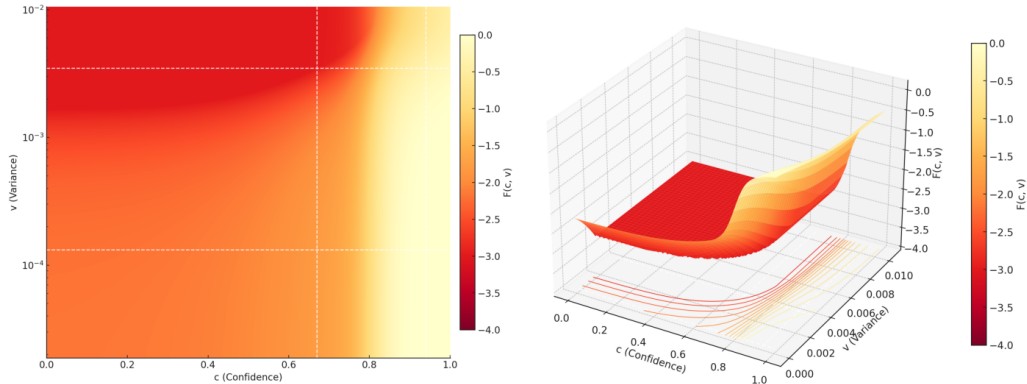

**Figure 12:** Left: heatmap of $g(c_s, v_s)$ with $v_s$ on a log scale; Right: 3D surface of $g(c_s, v_s)$. Dashed lines mark $q_{25}^c, q_{75}^c$ and $q_{25}^v, q_{75}^v$.

| Method | Math500 | | GSM8K | | Olympiad | |
|---|---|---|---|---|---|---|
| | **Pass@1** ↑ | **#Tokens** ↓ | **Pass@1** ↑ | **#Tokens** ↓ | **Pass@1** ↑ | **#Tokens** ↓ |
| **Full** $[g(c, v)]$ | 83.0 | 3474 | 78.3 | 765 | 43.9 | 7235 |
| $g(c)$ | 82.6↓ | 3387↓ | 78.1↓ | 890↑ | 41.7↓ | 6612↓ |
| $g(v)$ | 76.6↓ | 3596↑ | 78.0↓ | 658↓ | 39.6↓ | 6876↓ |

**Table 10: Axis-wise ablations on the R1-1.5B backbone across difficulty levels.** We analyze performance changes on three math benchmarks with varying difficulty: **Math500** (medium), **GSM8K** (easy), and **Olympiad** (hard). Metrics are Pass@1 accuracy (%) and generated token numbers. Arrows indicate change relative to **Full** [ $g(c, v)$ ]: Acc. ↑ increase, ↓ decrease; Tokens ↓ decrease, ↑ increase.

## C.2 CROSS-DOMAIN AND CROSS-DIFFICULTY TRANSFERABILITY

**Cross-domain transferability.** To validate the cross-domain transferability of ReBalance, we conduct experiments on DeepSeek-R1-Distill-Qwen-1.5B. Specifically, we extract steering vectors and confidence statistics from coding (LiveCodeBench) and commonsense (StrategyQA) tasks and transfer them to the mathematics domain (MATH-500). We also report the quartiles of the confidence distribution (q25c, q75c) and variance distribution (q25v, q75v) derived from each extraction dataset. For readability, confidence quartiles are scaled by 100 and variance quartiles by 1000.

As observed in Tab. 11, although the confidence statistics differ to some extent across domains, ReBalance achieves strong performance in all cases. Notably, extractions from StrategyQA can even achieve higher Pass@1 scores compared to those from MATH within the same domain.

**Cross-difficulty transferability.** We categorize commonly used mathematical reasoning datasets by difficulty level: easy (GSM8K, AMC23), medium (MATH/MATH-500), and hard (AIME24, AIME25, Olympiad). Using the medium-difficulty datasets as an intermediary, we examine two transfer directions, easy-to-medium and hard-to-medium, to explore how changes in task difficulty affect the transfer performance of ReBalance. The experimental results on DeepSeek-R1-Distill-Qwen-1.5B are shown in Tab. 11.

However, data distributions vary across different datasets, which may introduce confounding factors beyond just difficulty. Moreover, it is challenging to precisely quantify the difficulty gaps between datasets, posing significant challenges to the analysis. Therefore, we leverage the ground-truth difficulty grading within the MATH dataset and conduct an analysis based on QwQ-32B (Tab. 11), performing an in-distribution, fine-grained difficulty classification from Level 1 to Level 5 within the same dataset.

Based on the experimental results, we have the following observations. Firstly, the higher the difficulty of the extraction dataset, the lower the confidence and the higher the variance. This reflects

| Dataset | Confidence Distribution | | | | MATH-500 | |
|---|---|---|---|---|---|---|
| | q25c | q75c | q25v | q75v | Pass@1↑ | #Tokens↓ |
| **QwQ-32B** | | | | | | |
| MATH | 70.6 | 92.0 | 0.4 | 7.3 | 95.2 | 3662 |
| Level-1 | 74.8 | 92.8 | 0.3 | 5.2 | 94.8 | **3447** |
| Level-2 | 72.4 | 92.7 | 0.3 | 6.9 | 94.8 | 3623 |
| Level-3 | 71.0 | 92.1 | 0.4 | 6.9 | 95.2 | 3678 |
| Level-4 | 69.6 | 92.0 | 0.4 | 7.4 | 95.0 | 3696 |
| Level-5 | 69.2 | 91.1 | 0.4 | 7.7 | **95.4** | 3720 |
| **DeepSeek-R1-Distill-Qwen-1.5B** | | | | | | |
| MATH | 66.8 | 93.9 | 0.5 | 11.0 | 83.0 | 3474 |
| GSM8K | 76.9 | 94.4 | 0.3 | 6.7 | 80.6 | **3221** |
| AMC23 | 61.8 | 94.2 | 0.4 | 10.4 | 82.8 | 3277 |
| AIME24 | 63.2 | 92.1 | 0.5 | 9.8 | 83.2 | 3568 |
| AIME25 | 58.9 | 90.2 | 0.4 | 9.3 | **84.0** | 3796 |
| Olympiad | 58.1 | 84.2 | 0.5 | 9.5 | 83.6 | 3862 |
| GPQA | 56.5 | 82.8 | 0.5 | 9.5 | 81.4 | 4260 |
| LiveCodeBench | 62.4 | 91.2 | 0.5 | 6.2 | 82.0 | 3482 |
| StrategyQA | 61.2 | 85.8 | 0.4 | 8.3 | 83.4 | 3667 |

Table 11: **Performance Variation under Difficulty-Conditioned Control Surfaces.** We investigate performance shifts induced by control surfaces derived from datasets of varying difficulty. For each difficulty tier, we extract the steering vectors and fit the associated control surface, and subsequently evaluate them on DeepSeek-R1-Distill-Qwen-1.5B and QwQ-32B. The results indicate systematic differences in Rebalance behavior, suggesting that dataset difficulty plays a non-trivial role in shaping the resulting control dynamics.

the model's broader and more frequent exploration of reasoning paths, aligning with the decision rule we proposed in Eq. (5), where confidence serves as an indicator. Secondly, Extraction datasets of easy difficulty prioritize token reduction, while those of hard difficulty prioritize accuracy improvement. We believe this occurs because easy extraction datasets exhibit higher confidence and lower variance; thus, according to Eq. (5), the range classified as overthinking is wider, and the underthinking range is narrower. This results in more frequent triggering of the overthinking criterion, causing the dynamic control function to preferentially suppress redundant reasoning. The opposite occurs for harder datasets.

Therefore, we suggest using extraction datasets of medium difficulty (e.g., MATH) in practical applications to achieve an optimal accuracy-efficiency trade-off.

### C.3 PASS@K AND AVG@K PERFORMANCE ANALYSIS

The core issue addressed by ReBalance is balanced thinking, i.e., mitigating overthinking while simultaneously preventing underthinking. Here, "underthinking" refers to cases where the model is inherently capable of solving a problem but produces an incorrect answer due to insufficient reasoning. According to the definition of Pass@k, this metric effectively grants the model multiple reasoning attempts and expanded exploration space, counting a question as correct if any one of the k sampled solutions is successful. Therefore, theoretically, for problems that meet the underthinking criterion, which are those that the model is truly capable of solving, the likelihood of obtaining a correct answer approaches certainty as the number of samples increases, eventually converging toward the model's capability ceiling (Karan & Du, 2025). At this point, the impact of underthinking on accuracy becomes negligible, and ReBalance primarily serves to alleviate overthinking.

To validate the above analysis, we evaluate Pass@20 over 20 samples and measure the average token length of generated sequences on the few-example datasets AMC23, AIME24, and AIME25. The results are shown in Tab. 12.

As can be seen, consistent with our analysis, ReBalance significantly reduces reasoning length without any degradation in model accuracy. This demonstrates that our proposed confidence indicator accurately characterizes overthinking and underthinking. Thanks to this precise characterization, ReBalance prunes only genuinely redundant reasoning steps rather than essential or effective ones,

| Method | AMC23 | | AIME2024 | | AIME2025 | |
|---|---|---|---|---|---|---|
| | Pass@20↑ | #Tokens↓ | Pass@20↑ | #Tokens↓ | Pass@20↑ | #Tokens↓ |
| **DeepSeek-R1-Distill-Qwen-1.5B** | | | | | | |
| Baseline | 97.5 | 7430 | 63.3 | 10645 | 43.3 | 10447 |
| ReBalance | **97.5** | **4832** | **63.3** | **9243** | **46.7** | **8652** |
| **DeepSeek-R1-Distill-Qwen-7B** | | | | | | |
| Baseline | 100.0 | 6021 | 76.7 | 11249 | 66.7 | 11379 |
| ReBalance | **100.0** | **5264** | **76.7** | **8563** | **66.7** | **9019** |
| **Qwen3-14B** | | | | | | |
| Baseline | 100.0 | 7331 | 90.0 | 11367 | 80.0 | 12717 |
| ReBalance | **100.0** | **5124** | **90.0** | **9247** | **80.0** | **10381** |
| **QwQ-32B** | | | | | | |
| Baseline | 100.0 | 7034 | 86.7 | 14121 | 83.3 | 13386 |
| ReBalance | **100.0** | **5651** | **86.7** | **9853** | **83.3** | **11586** |

**Table 12: Pass@20 Performance and Token Efficiency.** ReBalance preserves accuracy while consistently reducing the token usage of baseline models.

| Model | Avg@1 | Tok@1 | Avg@4 $\pm$ Std | Tok@4 $\pm$ Std |
|---|---|---|---|---|
| DeepSeek-R1-Distill-Qwen-1.5B | 79.6 | 4516 | 79.6 $\pm$ 0.004 | 4620 $\pm$ 100.9 |
| w/ ReBalance | 83.0 | 3474 | 83.3 $\pm$ 0.007 | 3553 $\pm$ 55.5 |
| DeepSeek-R1-Distill-Qwen-7B | 89.8 | 3699 | 89.4 $\pm$ 0.007 | 3703 $\pm$ 56.4 |
| w/ ReBalance | 92.6 | 2903 | 92.6 $\pm$ 0.001 | 2894 $\pm$ 38.1 |
| Qwen3-14B | 93.8 | 4470 | 93.8 $\pm$ 0.001 | 4550 $\pm$ 38.8 |
| w/ ReBalance | 94.0 | 3641 | 94.1 $\pm$ 0.001 | 3674 $\pm$ 44.6 |

**Table 13: Multi-sample evaluation on MATH-500.**

enabling lossless and efficient sequence compression even when the model operates near its capability ceiling.

Following prior work (Jaech et al., 2024; Guo et al., 2025), we report Avg@4 on the large-scale MATH-500 dataset. For the few-sample benchmarks AMC23, AIME24, and AIME25, we also include Avg@16 together with Avg@4 to evaluate the model's average case reasoning performance under both low-sampling settings and medium-sampling settings. As shown in Tab. 13 and 14, we have the following observations:

- **Inter-dataset variability in randomness.** As reflected by the standard deviations of Avg@k and Tok@k, datasets with very few samples exhibit substantially higher variance—up to an order of magnitude larger than that of large-sample datasets such as MATH-500. This affects both accuracy and generated sequence length.

- **Performance stability on large-sample datasets.** On datasets with sufficient sample size, multi-sample evaluation has minimal impact on the *relative* performance between ReBalance and the baseline. The performance gap remains largely consistent across different sampling counts.

- **Performance stability on few-sample datasets.** Although both accuracy and sequence length fluctuate more significantly with increased sampling on few-sample datasets, the relative improvement of ReBalance over the baseline remains stable. Notably, under 16-sample evaluation, ReBalance yields even larger performance gains compared to the original single-sample setting across multiple datasets and models, further demonstrating its effectiveness and robustness.

## C.4 PERFORMANCE VARIATION UNDER DIFFERENT CONFIDENCE DISTRIBUTIONS

| Model | Avg@1 | Tok@1 | Avg@4 ± Std | Tok@4 ± Std | Avg@16 ± Std | Tok@16 ± Std |
|---|---|---|---|---|---|---|
| **AIME24** | | | | | | |
| **DeepSeek-R1-Distill-Qwen-1.5B** | | | | | | |
| Baseline | 23.3 | 12596 | 21.7 ± 0.02 | 12897 ± 830.7 | 19.6 ± 0.03 | 12931 ± 830.7 |
| ReBalance | 36.7 | 9040 | 34.2 ± 0.04 | 9179 ± 718.4 | 35.6 ± 0.05 | 9179 ± 602.6 |
| **DeepSeek-R1-Distill-Qwen-7B** | | | | | | |
| Baseline | 40.0 | 13994 | 41.7 ± 0.03 | 13636 ± 434.2 | 41.3 ± 0.04 | 13764 ± 418.2 |
| ReBalance | 56.7 | 9012 | 55.0 ± 0.02 | 8664 ± 896.5 | 57.9 ± 0.05 | 9167 ± 620.9 |
| **Qwen3-14B** | | | | | | |
| Baseline | 66.7 | 10888 | 70.0 ± 0.02 | 11488 ± 188.1 | 67.1 ± 0.06 | 11174 ± 307.9 |
| ReBalance | 73.3 | 9464 | 71.7 ± 0.02 | 9627 ± 115.3 | 73.1 ± 0.02 | 9613 ± 112.1 |
| **AIME25** | | | | | | |
| **DeepSeek-R1-Distill-Qwen-1.5B** | | | | | | |
| Baseline | 16.7 | 14556 | 15.0 ± 0.02 | 14107 ± 354.3 | 15.4 ± 0.02 | 14589 ± 416.3 |
| ReBalance | 30.0 | 8140 | 27.5 ± 0.03 | 8447 ± 447.0 | 27.5 ± 0.03 | 8723 ± 626.9 |
| **DeepSeek-R1-Distill-Qwen-7B** | | | | | | |
| Baseline | 26.7 | 13778 | 28.3 ± 0.04 | 12340 ± 308.3 | 27.3 ± 0.04 | 12192 ± 308.3 |
| ReBalance | 40.0 | 9227 | 40.0 ± 0.02 | 8813 ± 609.3 | 40.0 ± 0.02 | 9241 ± 463.2 |
| **Qwen3-14B** | | | | | | |
| Baseline | 56.7 | 13125 | 55.0 ± 0.05 | 12900 ± 241.7 | 54.4 ± 0.06 | 12457 ± 298.7 |
| ReBalance | 56.7 | 11057 | 57.5 ± 0.06 | 11023 ± 213.1 | 57.3 ± 0.07 | 11013 ± 224.3 |
| **AMC23** | | | | | | |
| **DeepSeek-R1-Distill-Qwen-1.5B** | | | | | | |
| Baseline | 55.0 | 8990 | 53.8 ± 0.01 | 8616 ± 302.6 | 54.1 ± 0.02 | 8637 ± 538.8 |
| ReBalance | 80.0 | 5216 | 80.6 ± 0.04 | 4730 ± 255.3 | 80.0 ± 0.04 | 4729 ± 433.5 |
| **DeepSeek-R1-Distill-Qwen-7B** | | | | | | |
| Baseline | 75.0 | 6898 | 74.9 ± 0.02 | 6297 ± 520.2 | 74.8 ± 0.02 | 6297 ± 335.1 |
| ReBalance | 95.0 | 4767 | 93.8 ± 0.02 | 4115 ± 423.4 | 92.2 ± 0.04 | 4226 ± 394.4 |
| **Qwen3-14B** | | | | | | |
| Baseline | 95.0 | 7240 | 91.3 ± 0.03 | 7244 ± 115.3 | 93.1 ± 0.03 | 6985 ± 210.3 |
| ReBalance | 100.0 | 5230 | 98.8 ± 0.02 | 5120 ± 98.4 | 97.7 ± 0.02 | 4848 ± 190.2 |

**Table 14: Multi-sample evaluation on AMC23, AIME24, and AIME25.**

| Temperature | Confidence Distribution | | | | Baseline | | Rebalance | |
|---|---|---|---|---|---|---|---|---|
| | $q_{25}^c$ | $q_{75}^c$ | $q_{25}^v$ | $q_{75}^v$ | Pass@1 (%) | Token | Pass@1 (%) | Token |
| 0.2 | 1 | 1 | 0 | 0 | 68.4 | 6248 | 77.0 | 4335 |
| 0.4 | 0.9602 | 1 | 0 | 0.0006 | 72.6 | 5694 | 79.6 | 3906 |
| 0.6 | 0.8507 | 1 | 0 | 0.0039 | 79.4 | 4584 | 81.6 | 3689 |
| 0.8 | 0.7156 | 0.9845 | 0.0002 | 0.0098 | 81.0 | 4529 | 84.6 | 3485 |
| 1.0 | 0.5516 | 0.9449 | 0.0006 | 0.0164 | 82.4 | 4660 | 82.4 | 3488 |

**Table 15: Performance of Rebalance under different temperature settings on DeepSeek-R1-Distill-Qwen-1.5B.** Rebalance consistently provides higher accuracy and shorter reasoning traces from low- to high-temperature settings, demonstrating stable generalization under varying confidence distributions.

To examine the generalizability of Rebalance across distinct confidence regimes within the same model, we keep both the control surface and evaluation samples fixed while varying the model's decoding temperature. This setup allows us to assess whether Rebalance consistently yields performance gains under different confidence distributions. As shown in Tab. 15, Rebalance behaves adaptively across decoding temperatures. At low temperatures, the model exhibits persistently high confidence, which frequently triggers the underthinking detector; Rebalance then encourages more diverse reasoning trajectories. Conversely, at higher temperatures, the model produces higher variance and lower confidence, activating the overthinking detector; in this regime, Rebalance effectively narrows the search space and accelerates convergence. As illustrated in Fig. 13, the results provide a detailed view of how Rebalance adjusts the model's behavior across temperature regimes. At lower temperatures, Rebalance effectively suppresses overconfident predictions, thereby reducing underthinking and substantially improving the model's accuracy.

## C.5 SEMANTIC CHANGE AND CREATIVITY ANALYSIS

Even though Appendix A.6 examines the relationship between the keyword vocabulary and confidence, it remains necessary to analyze the semantics of the model's generated outputs. To this

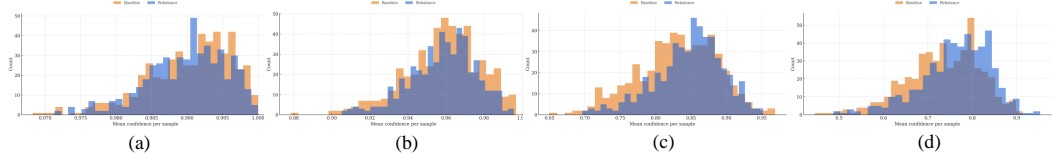

(a)      (b)      (c)      (d)

**Figure 13:** (a)–(d) show the mean confidence distributions of the Baseline and Rebalance models at temperatures 0.2, 0.4, 0.8, and 1.0, respectively. The trend reveals a clear temperature-dependent effect: at lower temperatures, Rebalance systematically reduces the model's overconfident predictions, whereas at higher temperatures, it shifts the distribution upward, counteracting the underconfidence introduced by increased sampling randomness.

| Method | Reflection | | | Transition | | | Performance Change | |
|---|---|---|---|---|---|---|---|---|
| | Word Count | TF | TF–IDF | Word Count | TF | TF–IDF | $\Delta$Pass@1 | $\Delta$Tokens |
| **DeepSeek-R1-Distill-Qwen-1.5B** | | | | | | | | |
| Baseline | 30.3 | 9.7 | 12.1 | 6.7 | 2.4 | 3.9 | – | – |
| NoThinking | 3.9 | 1.7 | 4.6 | 0.9 | 0.4 | 1.3 | -4.6 | -64.9% |
| NoWait | 1.6 | 1.7 | 3.5 | 0.1 | $1.9 \times 10^{-2}$ | $7.7 \times 10^{-2}$ | -1.6 | -41.4% |
| Rebalance | 10.5 | 6.4 | 9.3 | 1.7 | 1.0 | 2.3 | +3.4 | -23.1% |
| **DeepSeek-R1-Distill-Qwen-7B** | | | | | | | | |
| Baseline | 19.2 | 8.1 | 10.3 | 5.2 | 2.3 | 3.7 | – | – |
| NoThinking | 1.8 | 1.2 | 3.1 | 0.1 | $9.6 \times 10^{-2}$ | $4.5 \times 10^{-2}$ | -9.2 | -77.4% |
| NoWait | 1.2 | 1.8 | 3.9 | 0.03 | $2.8 \times 10^{-2}$ | $1.8 \times 10^{-2}$ | -3.0 | -32.9% |
| Rebalance | 12.8 | 6.7 | 9.2 | 2.7 | 1.4 | 2.8 | +2.8 | -21.5% |
| **Qwen3-14B** | | | | | | | | |
| Baseline | 24.0 | 9.2 | 10.8 | 8.7 | 3.3 | 4.4 | – | – |
| NoThinking | 10.4 | 5.5 | 9.4 | 2.8 | 1.0 | 2.4 | 0.0 | -40.5% |
| NoWait | 2.1 | 1.8 | 3.3 | 0.0 | $9.4 \times 10^{-3}$ | $4.1 \times 10^{-2}$ | -1.0 | -27.9% |
| Rebalance | 13.2 | 6.8 | 8.1 | 4.6 | 2.3 | 3.3 | +0.2 | -18.5% |
| **QwQ-32B** | | | | | | | | |
| Baseline | 26.7 | 11.2 | 13.3 | 10.1 | 3.8 | 5.1 | – | – |
| NoThinking | 26.0 | 11.5 | 13.7 | 9.5 | 3.8 | 5.1 | 0.0 | -13.7% |
| NoWait | 2.5 | 2.4 | 4.1 | 0.1 | $2.2 \times 10^{-2}$ | $8.1 \times 10^{-2}$ | -1.0 | -27.9% |
| Rebalance | 22.1 | 10.8 | 10.1 | 5.4 | 2.5 | 3.8 | +0.4 | -19.3% |

**Table 16: Semantic statistics of *Transition* and *Reflection* vocabularies across different methods and models, and corresponding performance changes.** For each method on each model, we report the total word count, term frequency (TF), and TF–IDF score for both vocabularies, as well as the change in accuracy and tokens relative to the Baseline of the same model.

end, we conduct a systematic semantic analysis of the DeepSeek-R1-Distill-Qwen-1.5B reasoning traces using the Transition and Reflection vocabularies introduced in SEAL Chen et al. (2025b). As shown in Tab. 16, we quantify the semantic changes of the model under NoWait, NoThinking, and our method before and after steering. For readability, TF and TF-IDF scores are scaled by a factor of 1000. We observe that both reflection and transition patterns decrease to varying degrees across all methods. However, the first two baselines suffer from noticeable accuracy drops. In contrast, our method preserves a non-negligible amount of these reasoning patterns, which substantially contributes to maintaining, and in some cases even improving, the model's accuracy. Retaining reflection and transition patterns appears to be a key factor for preserving reasoning correctness in large language models.

To evaluate whether applying Rebalance imposes any unintended drawbacks on creativity and the naturalness of the expressions, we further assess the models using the Creative Writing v3 benchmark Paech (2025). The experimental results are summarized in Tab. 17.

We evaluate four models using Claude-Sonnet-4.5 Anthropic (2024) as the judge model. For each model, we report three metrics: (1) Rubric Score, an aggregate quality score across multiple writing dimensions; (2) Elo Score, a relative writing-quality ranking calibrated with GPT-3.5-Turbo OpenAI (2023) and DeepSeek-R1-Distill-Qwen-1.5B as fixed anchors; and (3) Fine-grained Ability Scores,

| Method | Score | | Ability | | | | | | | | | | | | | | |
|---|---|---|---|---|---|---|---|---|---|---|---|---|---|---|---|---|---|
| | Rubric | Elo | Coh | Crt | Img | Pac | Ele | Inst | Voi | Dia | Flo | SDT | Ama | Emo | Pos | Pur | Ch |
| **DeepSeek-R1-Distill-Qwen-1.5B** | | | | | | | | | | | | | | | | | |
| Baseline | 15.0 | 30.0 | 1.3 | 1.8 | **2.8** | 1.9 | 5.4 | **1.5** | **2.9** | 2.9 | 4.1 | 2.5 | 3.2 | 1.2 | **11.5** | 9.2 | **1.8** |
| Rebalance | **15.2** | **57.5** | **1.5** | **1.9** | 2.7 | **2.2** | **5.6** | 1.4 | 2.8 | **3.0** | **4.3** | 2.5 | **3.3** | 1.2 | 11.2 | **9.4** | 1.7 |
| **DeepSeek-R1-Distill-Qwen-7B** | | | | | | | | | | | | | | | | | |
| Baseline | 22.9 | 308.9 | 4.0 | 4.4 | **4.8** | 4.8 | 6.6 | **3.1** | 5.0 | 4.2 | 6.6 | 4.0 | 5.4 | 2.7 | 9.9 | 9.6 | 3.4 |
| Rebalance | **23.4** | **349.7** | **4.3** | **4.5** | 4.7 | **5.1** | **6.7** | 3.0 | 5.0 | **4.6** | 6.6 | **4.1** | 5.4 | **2.9** | 9.9 | **9.9** | **3.5** |
| **Qwen3-14B** | | | | | | | | | | | | | | | | | |
| Baseline | 42.6 | 1295.2 | 8.8 | 7.1 | 10.3 | 7.7 | 8.2 | 9.2 | 11.2 | 6.4 | 8.1 | 5.5 | 6.6 | 8.1 | 10.8 | 8.4 | 8.8 |
| Rebalance | **50.5** | **1368.0** | **12.2** | **7.2** | **11.6** | **11.3** | **9.5** | **12.1** | **12.3** | **8.2** | **9.9** | **6.9** | **8.0** | **9.4** | **12.6** | **9.0** | **10.3** |
| **QwQ-32B** | | | | | | | | | | | | | | | | | |
| Baseline | 52.7 | 1438.2 | 12.6 | **8.1** | 12.0 | 11.7 | 9.8 | **13.6** | **12.6** | 8.5 | 9.8 | **8.1** | **8.3** | 9.8 | 11.4 | **9.6** | 10.6 |
| Rebalance | **52.8** | **1442.7** | **12.7** | 7.9 | **12.3** | **11.8** | **9.9** | 12.6 | 12.2 | **8.6** | 9.8 | 7.9 | 8.0 | 9.8 | **12.2** | 9.5 | **10.7** |
| **Qwen2.5-7B-Instruct** | | | | | | | | | | | | | | | | | |
| Baseline | 33.5 | 877.6 | 7.4 | 5.2 | 7.2 | 7.6 | 8.5 | 6.1 | 8.0 | 6.2 | 8.7 | 4.8 | 6.9 | 4.9 | 9.5 | 10.6 | 6.3 |
| **GPT3.5-Turbo** | | | | | | | | | | | | | | | | | |
| Baseline | 52.8 | 1500.0 | 13.2 | 7.1 | 11.5 | 12.2 | 10.5 | 12.7 | 13.3 | 8.1 | 11.0 | 7.2 | 8.9 | 9.1 | 11.5 | 10.4 | 10.8 |

**Table 17: Creative-writing performance on Creative Writing v3 benchmark.** Ability abbreviations: Coh = Coherent; Crt = Creativity; Img = Descriptive Imagery; Pac = Pacing; Ele = Elegant Prose; Inst = Instruction Following; Voi = Consistent Voice & Tone; Dia = Strong Dialogue; Flo = Sentence Flow; SDT = Show-Don't-Tell; Ama = Avoids Amateurish Prose; Emo = Emotional Depth; Pos = Avoids Positivity Bias; Pur = Avoids Purple Prose; Ch = Believable Characters.

covering fifteen creative-writing competencies: coherent (logical consistency and structural clarity), creativity (originality and non-templated expression), descriptive imagery (vivid, sensory-rich description), pacing (appropriate narrative flow), elegant prose (fluency and stylistic refinement), instruction following (accurate adherence to instructions), consistent voice & tone (stable narrative voice), strong dialogue (natural, character-appropriate dialogue), sentence flow (smooth transitions between sentences), show–don't–tell (conveying meaning through scene and action rather than exposition), avoids amateurish prose (avoidance of clichés or novice patterns), emotional depth (nuanced emotional expression), avoids positivity bias (avoidance of forced optimism), avoids purple prose (restraint from overly ornate language), believable characters (psychological plausibility and consistent character voices).

It can be observed that after applying ReBalance, the models generally maintain and even improve their performance in creativity and the naturalness of expressions. The percentage of metrics that are higher than or equal to those of the original models is notably high: 71% for DeepSeek-R1-Distill-Qwen-1.5B, 88% for DeepSeek-R1-Distill-Qwen-7B, 100% for Qwen3-14B, and 65% for QwQ-32B. Notably, Qwen3-14B achieves significant improvements across all metrics after applying ReBalance. We posit that this improvement stems from ReBalance's ability to continuously and gently guide the reasoning process, achieving a balance between overthinking and underthinking, thereby maintaining the model within effective reasoning boundaries. Consequently, the models exhibit measurable improvements in creative-writing performance, indicating that ReBalance accelerates convergence without compromising the model's capacity for innovative or divergent thinking.

We additionally evaluate Qwen2.5-7B-Instruct and find that its creativity scores are substantially higher than those of the distilled DeepSeek-R1-Distill-Qwen-7B, further supporting our observations. Our analysis reveals that distillation systematically reduces linguistic diversity, a property that is closely tied to creativity. Likewise, task-specific fine-tuning may limit creative expression by reinforcing existing patterns at the expense of novel exploration.

In contrast, Rebalance induces a form of cognitive restructuring in the model's internal reasoning process, which partially restores and enhances creative expressiveness. These findings highlight an important direction for future work: developing methods that improve reasoning stability without suppressing linguistic diversity or creative generation.

### C.6 CONFIDENCE CHARACTERISTICS OF OVERTHINKING AND UNDERTHINKING

Fig. 2(b) in the main text highlights a core contribution of our work: confidence serves as a continuous and reliable indicator for explicitly modeling overthinking and underthinking. In Fig. 2(b),

| Model | Normal (O-base) | | | Overthinking | | | Normal (U-base) | | | Underthinking | | |
|---|---|---|---|---|---|---|---|---|---|---|---|---|
| | Conf. | Var. | Len. | Conf. | Var. | Len. | Conf. | Var. | Len. | Conf. | Var. | Len. |
| DeepSeek-R1-Distill-Qwen-1.5B | 80.4 | 18.5 | 1357 | 78.6 | 21.3 | 2386 | 85.1 | 22.2 | 2909 | 89.7 | 16.7 | 1726 |
| DeepSeek-R1-Distill-Qwen-7B | 90.0 | 11.0 | 2809 | 81.8 | 23.0 | 5995 | 82.4 | 22.0 | 3259 | 91.2 | 12.0 | 2752 |
| Qwen3-14B | 92.6 | 8.7 | 5763 | 88.1 | 12.1 | 8819 | 85.3 | 11.3 | 6305 | 9.27 | 8.0 | 5743 |
| QwQ-32B | 84.0 | 16.1 | 3712 | 75.4 | 22.1 | 6377 | 76.2 | 18.5 | 4573 | 78.9 | 1.76 | 3080 |

**Table 18: Comparison of Confidence, Variance, and Output Length Across Thinking Modes.** For each model, we report statistics for Normal responses paired with Overthinking (Normal (O-base)) and Underthinking (Normal (U-base)), together with the corresponding Overthinking and Underthinking states.

we use DeepSeek-R1-Distill-Qwen-1.5B on MATH-500 for visualization. To demonstrate the generality of this observation, we extend the experimental setup to include four model sizes (1.5B to 32B) across three distinct model families. Tab. 18 presents a comprehensive quantitative analysis of stepwise confidence and confidence variance across these models. For clarity, confidence values are scaled by 100 and variance by 1000.

We can observe that, across all models shown in Tab. 18, the results consistently align with those of Fig. 2(b): Firstly, overthinking samples exhibit lower confidence and higher variance, suggesting hesitant and repeated switching between reasoning paths, often leading to redundant reasoning; Secondly, underthinking samples show higher confidence and lower variance, and this persistently high confidence often causes the model to prematurely commit to an incorrect reasoning path, hindering thorough exploration. This observation further confirms that confidence can serve as a general indicator, broadly applicable across various large reasoning models.

### C.7 PERFORMANCE COMPARISON WITH TRIMR AND FLASHTHINK

Since the official implementations of TrimR (Lin et al., 2025a) and FlashThink (Jiang et al., 2025) are not publicly available, we reproduce both methods for a fair comparison.

**Reproduction of TrimR.** We adopt the reflection tokens given in the paper to split reasoning into fixed-interval sub-thoughts and feed them into a verifier model for answer detection. We use a streaming inference pipeline that monitors generation through OpenAI-compatible endpoints. Since the step size is unspecified, we set it to 100 tokens, consistent with the original Fig. 8. Both the overthinking-compression module (via answer convergence) and the underthinking-compression module (via budget monitoring) are enabled accordingly. In the original study, TrimR was executed on Ascend NPUs with the NPU-native Pangu-7B model serving as the verifier. Since we do not have access to Ascend NPUs, our experiments are instead conducted on NVIDIA RTX PRO 6000 GPUs and adopt Qwen2.5-7B-Instruct (Yang et al., 2025a) as the verifier, following the other configuration outlined in their paper.

In TrimR, $R$ denotes the underthinking threshold defined in the original paper. Specifically, the reasoning process is terminated once it reaches $R\%$ of the maximum sequence length, with $R = 0.5$ used by default. Since TrimR is originally validated only on models of 32B parameters or larger, we conduct our comparison using QwQ-32B to ensure a fair assessment of its strengths. Despite TrimR's use of an additional 7B auxiliary model, an extra inference stage, and post-hoc optimization via repetition truncation, ReBalance still achieves significantly better performance. Moreover, our analysis of the $R$ parameter reveals that increasing $R$ leads to notable accuracy gains for TrimR on challenging benchmarks such as AIME24 and AIME25. This observation further supports the hypothesis that existing approaches designed to mitigate overthinking can inadvertently induce underthinking, highlighting a critical trade-off in current reasoning frameworks.

**Reproduction of FlashThink.** For FlashThink, we follow the paper's core procedure, i.e., segmenting the chain-of-thought using delimiter tokens and invoking a verifier model for early exit. Like TrimR, we employ a streaming inference pipeline that monitors generation via OpenAI-compatible endpoints. Each detected segment is forwarded to the verifier (Qwen2.5-7B-Instruct) using the original prompt template. The main results for TrimR and FlashThink are reported in Tab. 1, with more detailed comparisons provided in Tab. 19.

| QwQ-32B | MATH-500 | | AIME24 | | AIME25 | | GSM8K | | AMC23 | | GPQA-D | |
|---|---|---|---|---|---|---|---|---|---|---|---|---|
| | Pass@1↑ | #Tokens↓ | Pass@1↑ | #Tokens↓ | Pass@1↑ | #Tokens↓ | Pass@1↑ | #Tokens↓ | Pass@1↑ | #Tokens↓ | Pass@1↑ | #Tokens↓ |
| **TrimR** | | | | | | | | | | | | |
| threshold=0.5 | 93.8 | 3830 | 56.7 | 8345 | 43.3 | 8827 | 93.7 | 1319 | 90.0 | 6055 | 63.1 | 6380 |
| threshold=0.75 | 94.8 | 4048 | 70.0 | 10235 | 53.3 | 11397 | 96.7 | 1410 | 87.5 | 6778 | 69.2 | 7312 |
| threshold=1 | 93.8 | 4241 | 66.7 | 11007 | 56.7 | 11726 | 95.9 | 1432 | 90.0 | 6890 | 63.7 | 7012 |
| **Flashthink** | | | | | | | | | | | | |
| max tokens=16000 | 94.8 | 2854 | 56.7 | 8757 | 60.0 | 9613 | 96.3 | 1090 | 90.0 | 5200 | 64.7 | 5751 |
| max tokens=32000 | 94.6 | 2884 | 66.7 | 11390 | 63.3 | 11218 | 96.7 | 1098 | 95.0 | 5596 | 66.2 | 5982 |
| **ReBalance(ours)** | | | | | | | | | | | | |
| max tokens=16000 | 95.2 | 3662 | 70.0 | 10350 | 63.3 | 11575 | 96.8 | 1289 | 95.0 | 6064 | 67.2 | 6296 |

**Table 19: Performance comparison with TrimR and Flashthink on QwQ-32B.** All experiments are run with the same sampling parameters.

| Method | AMC23 | | AIME24 | | AIME25 | | Olympiad | |
|---|---|---|---|---|---|---|---|---|
| | Pass@1 | #Tokens | Pass@1 | #Tokens | Pass@1 | #Tokens | Pass@1 | #Tokens |
| **DeepSeek-R1-Distill-Qwen-1.5B** | | | | | | | | |
| Baseline | 55 | 8990 | 23.3 | 12596 | 16.7 | 14556 | 41.2 | 8785 |
| Dynamic Temperature | 75 | 7344 | 36.7 | 12054 | 23.3 | 11659 | **44.7** | 8538 |
| Dynamic Steering | **80** | **5216** | **36.7** | **9040** | **30.0** | **8140** | 43.9 | **7253** |
| **DeepSeek-R1-Distill-Qwen-7B** | | | | | | | | |
| Baseline | 75 | 6898 | 40.0 | 13994 | 26.7 | 13778 | 56.1 | 7590 |
| Dynamic Temperature | 82.5 | 5856 | 53.3 | 10593 | 36.7 | 10740 | **57.5** | 7498 |
| Dynamic Steering | **95.0** | **4767** | **56.7** | **9012** | **40.0** | **9227** | 57.0 | **6321** |
| **QwQ-32B** | | | | | | | | |
| Baseline | 87.5 | 7021 | 66.7 | 14342 | 46.7 | 13350 | 66.7 | 8219 |
| Dynamic Temperature | 92.5 | 6721 | 66.7 | 11202 | 53.3 | 12134 | 67.6 | 8160 |
| Dynamic Steering | **95.0** | **6064** | **70.0** | **10350** | **63.3** | **11575** | **68.6** | **7422** |

**Table 20:** Performance comparison across three model sizes under different inference-time control methods.

Overall, while FlashThink and TrimR effectively reduce token usage, they incur significant inference-time overhead. Frequent verification interrupts disrupt the decoding process, stall the pipeline, and degrade KV-cache efficiency; moreover, deploying a separate verifier increases system complexity. In contrast, ReBalance introduces minimal overhead, maintains uninterrupted decoding, and simultaneously reduces token consumption and latency, offering a more practical and self-contained solution.

## C.8 BALANCED THINKING WITH DYNAMIC TEMPERATURE

In this work, we use steering as an illustrative example to practically implement our idea of balancing overthinking and underthinking. Notably, the concept itself is generalizable and applicable to a variety of approaches aiming at promoting balanced thinking. To support this argument, inspired by Zhang et al. (2024) and Zhu et al. (2024), we replace the dynamic steering component in ReBalance with a much simpler approach: adjusting the temperature parameter. Specifically, when overthinking is detected during inference, we reduce the temperature to avoid excessively divergent exploration leading to redundant reasoning. Conversely, when underthinking is detected, we increase the temperature to broaden reasoning. Due to constraints in time and computational resources, we conduct only preliminary experiments using a discrete, binary hyperparameter setting without further tuning. Still using confidence as the indicator, we set the temperature to 1.2 upon detecting underthinking, and reduce it to 0.7 upon detecting overthinking.

The comparative results are summarized in Tab. 20. Even with such a simple configuration, our balanced thinking approach achieves significant performance improvements and reductions in reasoning length. Therefore, we believe the performance can be further enhanced by introducing a model behavior-based dynamic function fitting method similar to ReBalance, which naturally enables adaptive continuous regulation without manual hyperparameter tuning.

## C.9 ADDITIONAL PROTOTYPE CONSTRUCTION STRATEGIES

In Sec. 3.2 of the main text, we provide two definitions for overthinking and underthinking in Eq. (3) and Eq. (5), respectively, and select Eq. (5) as ReBalance's explicit modeling approach. To address potential concerns, we present a comparative analysis and experimental validation of these two definitions.

**Comparative analysis.** Although Eq. (3) appears more concise and intuitive, it serves only as a theoretical definition of overthinking and underthinking. Its purpose is to provide a conceptual distinction between the two phenomena through a formalized expression. This definition operates at the trajectory level, classifying an entire reasoning trajectory as overthinking or underthinking by comparing it against an idealized stability index that acts as a decision boundary. Consequently, if Eq. (3) were directly used as the indicator for steering vector extraction, it would inevitably lead to the following issues.

- **Mismatch in operational granularity.** Our method aims to adaptively adjust the model's behavior based on the real-time reasoning state at each step. This requires a step-level, fine-grained indicator to detect tendencies toward overthinking or underthinking. To avoid a mismatch in operational granularity, we therefore maintain the same step-level resolution during the steering vector extraction. However, Eq. (3) only supports trajectory-level classification of reasoning modes, and this granularity mismatch may lead to suboptimal performance.

- **Limited applicability.** The stability index requires access to ground truth, which introduces an additional dependency on labeled data. Although one could follow Lin et al. (2025a) using an external verifier to approximate ground truth by assessing the existence and equivalence of answers across consecutive sub-thoughts, this strategy still relies on the verifier's capability, prompt design, and hyperparameters for determining answer equivalence. Consequently, it may incur extra engineering overhead and introduce potential performance bottlenecks.

- **Difficulty in capturing complex reasoning dynamics.** Eq. (3) defines the stability index only when all subsequent reasoning steps yield identical answers that exactly match the ground truth. However, existing studies have shown that the accuracy of reasoning models is not positively correlated with reasoning length (Chen et al., 2024b); on the contrary, longer reasoning sequences may introduce more hallucinations (Liu et al., 2025a). Therefore, while Eq. (3) offers a convenient and intuitive way to conceptually distinguish overthinking from underthinking, its rigid requirement makes it unsuitable as a practical indicator in real-world scenarios, given the inherent complexity and variability of actual reasoning dynamics.

**Experimental validation.** To quantitatively validate the above analysis, we perform steering vector extraction using Equation 3 as follows. First, consistent with existing methods, we randomly select 500 seen samples from the MATH training set and feed them into DeepSeek-R1-Distill-Qwen-1.5B. We collect the generated output sequences and record the model's confidence at each reasoning step. To identify steps corresponding to overthinking and underthinking, following Lin et al. (2025a) and Jiang et al. (2025), we use Qwen2.5-7B-Instruct (Yang et al., 2025a) to determine whether each step contains the ground truth (see Appendix G for the prompt template). Once a step containing the ground truth is detected, it is designated as the stability index: all preceding steps are classified as underthinking, and all subsequent steps as overthinking. The labeling procedure produces 29,701 overthinking samples and 24,710 underthinking samples, showing that our partitioning strategy is reasonable and results in a well-balanced dataset.

To analyze the relative positional distribution of stability indices within the entire thinking sequence, we divide the range of relative positions (0–1) into 10 equal intervals (bins). We then count the number of occurrences of stability indices falling into each interval, as shown in Tab. 21. Notably, an interesting observation emerges here: the stability indices exhibit a bimodal distribution along the thinking sequence, with pronounced concentrations around Interval 1 and 10. These two Intervals correspond to the underthinking and overthinking behaviors of reasoning models, respectively, further providing strong empirical support for the necessity of balanced thinking.

The subsequent steps, including steering vector extraction, fitting of the dynamic control function, and dynamic steering during inference, are kept identical to those in ReBalance. The experimental results, along with the comparison of steering vector norms $\|S\|_2$, are shown in Tab. 22.

| Interval | 1 | 2 | 3 | 4 | 5 | 6 | 7 | 8 | 9 | 10 |
|---|---|---|---|---|---|---|---|---|---|---|
| Relative Position Range | [0.0, 0.1) | [0.1, 0.2) | [0.2, 0.3) | [0.3, 0.4) | [0.4, 0.5) | [0.5, 0.6) | [0.6, 0.7) | [0.7, 0.8) | [0.8, 0.9) | [0.9, 1.0] |
| Count of Stability Indices | **27.41** | 16.45 | 10.96 | 9.87 | 6.58 | 3.29 | 2.85 | 2.19 | 3.95 | **16.45** |

**Table 21:** Distribution of stability indices over relative position intervals.

| Methods | $\|S\|_2$ | MATH-500 (Pass@1) | MATH-500 (#Tokens) | AIME24 (Pass@1) | AIME24 (#Tokens) |
|---|---|---|---|---|---|
| Baseline | – | 79.6 | 4516 | 23.3 | 12596 |
| Vector Extraction w/ Stability Index | 11.4 | 81.8 | 4715 | 20.0 | 12563 |
| Vector Extraction w/ Confidence | **62.4** | **83.0** | **3474** | **36.7** | **9040** |

**Table 22:** Comparison of steering vector extraction variants.

We observe that the experimental results obtained using the steering vector extracted via Eq. (5) significantly outperform those of Eq. (3) in both accuracy and efficiency, strongly validating our analysis above. Moreover, the difference in $\|S\|_2$ reveals that when switching to stability index-based extraction, the norm of the steering vector becomes substantially smaller. This indicates a blurring of the boundary between overthinking and underthinking, further explaining the performance gap between the two methods.

## C.10 COMPARABILITY WITH NPU DEVICES

To further validate the generalization of REBALANCE, we conduct experiments on the Ascend 910B NPU platform using the NPU-native openPangu-Embedded-7B-V1.1 (Chen et al., 2025a) model in its slow-thinking mode. Our evaluation covers both simple (GSM8K) and challenging (AIME25) problems. Compared to the baseline, REBALANCE consistently improves reasoning accuracy and efficiency, as shown in Tab. 23. Method adaptation details are available in our GitHub repository.

| Method | AIME2025 | | GSM8K | |
|---|---|---|---|---|
| | Pass@1 | #Tokens | Pass@1 | #Tokens |
| Baseline | 73.3 | 14552 | 94.5 | 1933 |
| Ours | 76.7 | 9417 | 94.3 | 1363 |

**Table 23:** Evaluation on Ascend 910B (openPangu-Embedded-7B-V1.1 in slow-thinking mode): REBALANCE improves Pass@1 while reducing token usage.

## D DETAILS ON EXPERIMENTAL SETTINGS

**Benchmarks.** Evaluation is conducted on *mathematics reasoning* datasets: MATH-500 (Lightman et al., 2023b), AIME24 (AI-MO, 2024a), AIME25 (OpenCompass, 2025), AMC23 (AI-MO, 2024b), GSM8K (Cobbe et al., 2021), and OLYMPIADBENCH (He et al., 2024); *scientific reasoning* dataset, GPQA DIAMOND (Rein et al., 2024); *commonsense reasoning* dataset, STRATEGYQA (Geva et al., 2021); and *code reasoning* dataset, LIVECODEBENCH (Jain et al., 2024).

**Evaluation metrics.** We implement REBALANCE in both Hugging Face Transformers (Wolf et al., 2019) and vLLM (Kwon et al., 2023b). We evaluate using **Pass@1** ($\uparrow$) and the *average* number of generated tokens **Tok** ($\downarrow$). Unless otherwise specified, results are reported with the Transformers implementation.

**Backbone reasoning models.** We conduct experiments on 5 open-source large language models used as backbones: DEEPSEEK–R1–DISTILL–QWEN (1.5B and 7B) (Guo et al., 2025), QWEN3–14B (Yang et al., 2025a), QWQ–32B (Team, 2025), and OPENPANGU-EMBEDDED-7B-V1.1 (Chen et al., 2025a). Together, they span 4 model architectures and 4 parameter scales, enabling controlled comparisons across model families and sizes.

**Steering extraction and dynamic function fitting.** From 500 randomly sampled MATH (Hendrycks et al., 2021) problems, we estimate a *steering vector* and a *control surface* for each backbone once and hold them fixed across all benchmarks.

**Baseline methods.** We compare REBALANCE against representative training-free methods for efficient inference that do not rely on auxiliary models. (i) **PROMPT-BASED** methods: CoD (Xu et al., 2025a) and NoThinking (Ma et al., 2025b); (ii) **OUTPUT-BASED** methods: NoWAIT (Wang et al., 2025a); (iii) **DYNAMIC EARLY-EXIT** methods: DynaSoR–CoT (Fu et al., 2025), DEER (Yang et al., 2025b), FLASHThink Jiang et al. (2025), and TRIMR Lin et al. (2025a); (iv) **STEERING** methods: SEAL (Chen et al., 2025b) and MANIFOLD STEERING (Huang et al., 2025b). This set covers the major design paradigms (prompting, output-time control, early exiting, and latent steering) under a training-free setting. Further details on additional **PROMPT-BASED** baselines are provided in Appendix G.

**Hardware.** All experiments were conducted on a single server with **8× NVIDIA RTX PRO 6000 (Blackwell Server Edition)** GPUs.

**Decoding settings.** Unless otherwise noted, we use nucleus sampling with

$$\texttt{temperature} = 0.7, \quad \texttt{top\_p} = 0.95, \quad \texttt{max\_generated\_tokens} = 16000.$$

The same decoding configuration is applied across both the Transformers and vLLM backends.

## E  DETAILS ON BENCHMARKS

**MATH-500 (*moderate*; 500 problems).** Comprises 500 problems spanning arithmetic, algebra, geometry, and calculus, with varying difficulty levels. It evaluates a model's ability in complex mathematical formalism, equation solving, and structured reasoning. (Lightman et al., 2023b)

**AIME24 (*hard*; 30 problems).** An Olympiad-style set assessing logical deduction and advanced problem-solving skills; includes official AIME problems from the 2024 cycle. (AI-MO, 2024a)

**AIME25 (*hard*; 30 problems).** An updated set from the same AIME competition as AIME24, continuing to target high-level deductive and multi-step mathematical reasoning. (OpenCompass, 2025)

**GPQA DIAMOND (*hard*; 198 problems).** A challenging graduate-level subset with multiple-choice questions authored by domain experts in biology, physics, and chemistry. (Rein et al., 2024)

**AMC23 (*simple*; 40 problems).** An aggregated 40-problem set based on AMC 12 (2023 A/B). Items are multiple choice and span the standard high-school curriculum (calculus excluded), positioned below AIME-level difficulty. (AI-MO, 2024b)

**GSM8K (*simple*; 1319 problems).** Grade-school and middle-school word problems emphasizing short chain-of-thought arithmetic reasoning; commonly used split is ∼7.5k train / ∼1k test. (Cobbe et al., 2021)

**OLYMPIADBENCH (*hard*; 675 problems).** A bilingual math+physics Olympiad-style benchmark sourced from international/national olympiads and Gaokao, substantially more challenging than standard competition datasets. (He et al., 2024)

**STRATEGYQA (*simple*; 2,780 problems).** Yes/No questions requiring *implicit* multi-hop commonsense reasoning; each example provides a decomposition and supporting evidence passages. (Geva et al., 2021)

**LIVECODEBENCH (*hard*; 400 problems, v1).** A contamination-aware coding benchmark constructed from competitive-programming problems (e.g., LeetCode, AtCoder, Codeforces). Tasks require generating runnable programs that are judged by execution-based unit tests, emphasizing algorithmic reasoning, data-structure design, and implementation fidelity. We use version v1 with 400 problems. (Jain et al., 2024)

## F   DETAILS ON PROMPTS

**Math - (MATH-500, AIME 2024, AIME 2025, AMC23, GSM8K, Olympiad).**

```
<|System|> Please reason step by step, and place the final answer
    inside \boxed{}.
<|User|> [question]
```

**Science - GPQA.**

```
<|System|> Please reason step by step, and place the final answer
    inside \boxed{}.
<|User|> [question]

Answer with the choice letter only, in \boxed{}. Do not include
    option text.
```

**Commonsense - StrategyQA.**

```
<|System|> You answer binary commonsense questions. Think step by
    step, then output exactly one final line: \boxed{Yes} or \
    boxed{No}.
<|User|> [question]

Answer with \boxed{Yes} or \boxed{No} only.
```

**Code - LiveCodeBench.**

```
<|User|>
### Instruction: You will be given a question (problem
    specification) and will generate a correct Python program
    that matches the specification and passes all tests. You will
     NOT return anything except for the program.
Question:
[problem]
Ensure that when the python program runs, it reads the inputs,
    runs the algorithm and writes output to STDOUT.
python # YOUR CODE HERE
### Response:<|im_end|><|im_start|>assistant<think>
```

**Prompt Used for Steering Vector Extraction in Eq. (3).**

```
You are an Answer Detector for reasoning blocks of a large
    language model.
I will provide a gold answer and ONE sentence from the model's
    internal thinking process.

Your task: Determine whether THIS thinking sentence already
    clearly contains the correct answer.

Gold answer: {gold_answer}
Thinking sentence: {thinking_sentence}

Respond ONLY with a JSON object, in this exact format:

{found: <yes/no>}

Rules:
1. Output yes if this thinking sentence:
    - explicitly states the gold answer, OR
    - gives a mathematically/semantically equivalent expression,
     OR
    - computes a value that uniquely determines the gold answer.
2. Otherwise output no.
3. No explanations, no extra fields, no additional text.
4. Only judge THIS sentence. Ignore all context.
5. Ignore notation differences, equivalent arithmetic, and
    paraphrases.
```

## G  DETAILS ON PROMPT-BASED APPROACHES

From the main experimental table, we observe that prompt-based approaches can, to a certain extent, mitigate redundant reasoning. This represents one of the simplest methods for altering model behavior. The specific details of the currently popular "Magic Prompts" are summarized in the Table 24.

| Method | Prompt Modification |
|---|---|
| CoT (Wei et al., 2022) | *Please reason step by step.* |
| CoD (Xu et al., 2025a) | *Think step by step, but only keep a minimum draft for each thinking step, with 5 words at most.* |
| CCoT (Renze & Guven, 2024) | *Think step by step, Be concise.* |
| CCoT-2-45 (Nayab et al., 2024) | *Let's think a bit step by step and limit the answer length to 45 words.* |
| BTC (Ding et al., 2024) | *Rapidly evaluate and use the most effective reasoning shortcut to answer the question.* |
| NoThinking (Ma et al., 2025a) | *<think> Okay, I have finished thinking.</think>* |

Table 24: Prompt modifications used in different reasoning strategies.

Modifying prompts is not in conflict with our proposed method. In fact, we integrate our approach with prompt-based techniques in our experiments and observe that QwQ-32B achieves further performance improvements on several datasets. For instance, on the MATH-500 dataset, replacing our prompt reduced the number of generated tokens from 3662 to 3064 without compromising accuracy, demonstrating the complementary benefits of prompt refinement. Notably, existing research has shown that even with the same prompt content, different positioning strategies can significantly affect both the accuracy and efficiency of large language models (Cobbina & Zhou, 2025). However,

our main experimental results also indicate that prompt-based methods are not always effective, as the model does not consistently follow instructions. Regarding the broader issue of model controllability, a substantial body of research has already explored the use of reinforcement learning (RL) techniques to achieve more reliable control over large reasoning models (LRMs) (Aggarwal & Welleck, 2025; Yuan et al., 2024). Building upon these insights, exploring how to effectively combine multiple training-free strategies with advanced control methods remains an important direction for future work.

## H  DETAILED DISCUSSION OF RELATED WORKS

**Chain-of-Thought (CoT).**  CoT prompting elicits intermediate rationales and markedly improves multi-step reasoning (Wei et al., 2022); *self-consistency* further aggregates diverse chains (Wang et al., 2022). Beyond a single chain, search/verification variants under the Tree-of-Thought umbrella include *Tree-of-Thoughts* (Yao et al., 2023), *Stream-of-Search* (Gandhi et al., 2024), *Graph-of-Thoughts* (Besta et al., 2024), *Process Reward Models* (Lightman et al., 2023a), and *RL-based Self-Correction* (Kumar et al., 2024). A converging view is that judiciously increasing *test-time compute*—via multiple paths, search, or verification—can rival or surpass pure parameter scaling for reasoning (Snell et al., 2025).

**Latent Reasoning.**  Latent reasoning shifts the chain-of-thought from discrete tokens to *continuous* hidden representations, reducing tokenized traces and sampling while preserving intermediate signals—thus improving test-time efficiency. Representative approaches include soft thinking with gated hidden-state signals (Zhang et al., 2025d), training models to reason directly in a continuous latent space via internal activations (Hao et al., 2024), compressing long CoT into dense vectors (Cheng & Van Durme, 2024), and assistant-guided soft CoT that maintains a multi-step structure in latent form (Xu et al., 2025b).

**Post-Training Methods for Efficient Reasoning.**  Post-training approaches reduce test-time cost by shaping models' use of chain-of-thought (CoT) after pretraining. We group them into *SFT-based* and *reinforcement fine-tuning (RFT)-based*

*SFT-based.*  Supervised fine-tuning can induce *conditional brevity*: paired supervision under matched conditions (e.g., long vs. short) teaches when concise reasoning suffices (Kang et al., 2025). A complementary line supervises *compressed* rationales with an explicit compression control at inference; more principled compression schemes further improve faithfulness and controllability (Xia et al., 2025; Yuan et al., 2025).

*RFT-based.* Reinforcement fine-tuning directly optimizes the accuracy–efficiency trade-off via reward shaping and preference learning. Examples include difficulty-aware ranking to form preference data followed by SimPO optimization (Shen et al., 2025); fixed accuracy–length rewards with PPO (Arora & Zanette, 2025); self-adaptive CoT learning with GRPO (Yang et al., 2025c); and dynamically weighted rewards that balance accuracy and length during PPO training (Su & Cardie, 2025). Control-token policies decide *when* to think using GRPO, and explicit reinforcement of *how long* to reason enables length control (Aggarwal & Welleck, 2025).

Overall, these methods teach models to reason *when necessary* and remain concise otherwise, improving the accuracy–latency/token trade-off without increasing parameter count.

**Steering-Based Methods for Efficient Reasoning.**  Recent work explores *steering* mechanisms that intervene directly in a model's latent states to improve reasoning efficiency without retraining the backbone. Early work has already examined steering in the prompt space to elicit more controllable model outputs (Liu et al., 2023), and has also leveraged steering-based methods to mitigate hallucinations (Liu et al., 2024). Recent work has extensively explored using steering-based methods to enable more efficient reasoning. SEAL (Chen et al., 2025b) partitions model thoughts into *execution*, *reflection*, and *transition* phases, and uses a small amount of training data to bias the model toward the *execution* mode. MANIFOLD STEERING (Huang et al., 2025b) constructs *redundant* versus *concise* datasets based on response length and keyword density, derives a steering vector from them, and applies it to *all tokens across all layers*. REASONING STRENGTH PLANNING (Sheng et al., 2025) further introduces a *pre-allocated direction vector* injected into the activation corre-

sponding to the `<think>` token, whose magnitude encodes the desired reasoning strength in terms of the target number of reasoning tokens, with steering consistently applied at each layer for every generated token. Unlike the static steering methods above, CONTROLLING THINKING SPEED(Lin et al., 2025b) introduces a dynamic variant. They construct steering vectors by pairing long and short correct responses and extracting the hidden states of the last token in the first two reasoning steps at a chosen layer. A sliding-window controller then adjusts the steering strength based on token-level difficulty, measured via the Jensen–Shannon divergence between shallow- and deep-layer logit distributions, decreasing or increasing the magnitude according to a window-specific threshold.

However, the adjustment mechanism remains fundamentally one-directional in how steering magnitude is updated. Moreover, the steering-based methods above focus primarily on mitigating overthinking, yet overlook a complementary issue: alleviating overthinking often introduces or amplifies underthinking (Wang et al., 2025c). REBALANCE addresses this gap through a bidirectional dynamic steering mechanism that uses real-time confidence during reasoning as an indicator to assess the tendency toward overthinking or underthinking, dynamically adjusting both the direction and intensity of steering to achieve efficient reasoning with balanced thinking.

**Early-Exit Methods for Efficient Reasoning.** A prominent line of work toward efficient reasoning involves enabling models to exit early from their reasoning process once sufficient evidence for an answer has been gathered. Representative approaches such as TRIMR (Lin et al., 2025a) and FLASHTHINK (Jiang et al., 2025) employ an external instruction-following model to monitor the target model's chain-of-thought: when the monitor deems further reasoning unnecessary, it halts the process and triggers answer generation. Other methods, including DEER (Yang et al., 2025b) and DYNASOR-COT (Fu et al., 2025), instead rely on internal signals, such as the model's confidence or entropy over candidate answers, to determine an appropriate exit point. While these strategies effectively reduce token consumption by forcibly terminating redundant reasoning, they all share a fundamental limitation: the decision to terminate is binary and coarse-grained, typically applied at the level of the entire reasoning path (e.g., sub-thoughts in TRIMR or reasoning chunks in FLASH-THINK). This rigid binary selection risks discarding potentially valuable reasoning steps, thereby inducing additional underthinking (Wang et al., 2025c). Although TRIMR includes certain mechanisms addressing underthinking, it primarily opts for abandoning reasoning upon detecting underthinking, which leans towards engineering-driven token length optimization rather than genuinely addressing underthinking itself. Moreover, both TRIMR and FLASHTHINK depend on handcrafted keyword triggers to identify termination points, limiting their adaptability across models and tasks.

In contrast, REBALANCE departs fundamentally from this paradigm. Rather than merely mitigating overthinking through early exit, ReBalance is explicitly designed to simultaneously mitigate overthinking and prevent underthinking. It achieves this not by discarding reasoning paths, but by leveraging confidence as an indicator to dynamically detect the model's tendency toward overthinking or underthinking in real-time. Based on this detection, it adaptively adjusts the strength and direction of steering, thereby dynamically controlling the model's behavior to keep its reasoning state consistently within the reasoning boundary. This enables balanced thinking without requiring any additional verifiers or inference stages, making ReBalance an efficient, dynamic, and fine-grained reasoning acceleration approach.

**Acceleration for Efficient Reasoning.** Beyond training-time efficiency, a complementary line of work targets *inference-time* acceleration—reducing latency and improving throughput under fixed hardware budgets. One class of methods exploits *speculative decoding*, drafting tokens with a lightweight proposer and verifying them with the target model to amortize compute (Leviathan et al., 2023). A second class reduces memory and scheduling overhead via *paged* KV-cache management and continuous batching, enabling high-utilization serving at scale (Kwon et al., 2023a). A third line optimizes the *sampling process* itself: Monte Carlo tree or search–style strategies for data synthesis (Li et al., 2025), best-of-$n$ reasoning accelerated by speculative rejection (Sun et al., 2024), and early-decoding schemes that *self-estimate* the necessary $n$ to balance quality and cost (Wang et al., 2025b).

**Small Language Models (SLMs) for Efficient Reasoning.** A complementary line of work pursues efficient inference by *compressing* or *transferring* reasoning ability into smaller backbones. First, *quantization* can degrade multi-step reasoning if applied naively, calling for calibration-aware

| Method | TPS | TPR (s) | Additional GPU Memory (GB) | #Tokens |
|---|---|---|---|---|
| **DeepSeek-R1-Distill-Qwen-7B** | | | | |
| Baseline | 80.2 | 174.6 | – | 13994 |
| NoThinking | 80.1 | 55.2 | 0.0 | 4427 |
| CoD | 80.2 | 145.5 | 0.0 | 11663 |
| FlashThink | 73.8 | 135.9 | 18.3 | 10034 |
| TrimR | 48.8 | 147.7 | 15.4 | 7213 |
| SEAL | 78.8 | 128.3 | 0.0 | 10112 |
| ReBalance (Ours) | 78.5 | 114.8 | 0.0 | 9012 |
| **QwQ-32B** | | | | |
| Baseline | 20.5 | 698.6 | - | 14342 |
| NoThinking | 20.6 | 509.6 | 0.0 | 10507 |
| CoD | 20.7 | 552.0 | 0.0 | 11438 |
| FlashThink | 14.9 | 673.4 | 18.3 | 10034 |
| TrimR | 6.5 | 1289.8 | 15.7 | 8345 |
| SEAL | 20.3 | 508.8 | 0.0 | 10344 |
| ReBalance (Ours) | 20.5 | 504.4 | 0.0 | 10350 |

**Table 25:** Efficiency analysis on DeepSeek-R1-Distill-Qwen-7B and QwQ-32B.

schemes and mixed-precision designs (Liu et al., 2025b). Second, *distillation* transfers long-horizon reasoning into compact policies by (i) shortening and regularizing chain-of-thought traces and (ii) internalizing deliberate reasoning into fast feedforward behavior (Luo et al., 2025; Yu et al., 2024). Third, *pruning/compression* benchmarks sparsity and related compression knobs on complex reasoning tasks, revealing sensitivity to where and how compression is applied (Zhang et al., 2025b). Finally, a recent assessment reports the combined effects of distillation and pruning on SLMs, highlighting regimes where small models recover strong reasoning at a fraction of the compute (Srivastava et al., 2025).

## I  EFFICIENCY ANALYSIS

We conduct a comprehensive efficiency evaluation of ReBalance by comparing its inference overhead against both the baseline and other efficient reasoning methods. Specifically, we report four key metrics: tokens per second (TPS), time per request (TPR), and additional GPU memory consumption relative to the baseline.

Specifically, to ensure a thorough comparison, we include prompt-based methods (NoThinking, CoD), early-exit methods (FlashThink, TrimR), and latent steering methods (SEAL). Different from existing methods, ReBalance preserves an effective thinking process by promoting exploration when necessary to avoid underthinking, particularly when the model faces difficult reasoning problems. As a result, it is more likely to incur efficiency overheads on challenging datasets. In the following, we adopt AIME24 for the efficiency evaluation, as shown in Tab. 25.

- **Token generation efficiency:** By observing TPS, it can be seen that since the confidence utilized by ReBalance can be directly obtained from the log probability of each token's decoded output, and the dynamic function introduced in this method is very lightweight, the proposed dynamic adjustment logic has a negligible impact on the single-token generation time compared to the baseline.

- **Reasoning time acceleration:** As shown by TPS and #Tokens, ReBalance significantly shortens reasoning length without compromising token generation efficiency, yielding **1.5×** and **1.4×** TPR speedups over DeepSeek-R1-Distill-Qwen-7B and QwQ-32B, respectively.

- **Additional GPU memory usage:** Although ReBalance requires the use of extracted steering vectors during reasoning, their GPU memory footprint is minimal (e.g., the vector size for QwQ-32B is only 22 KB). In contrast, early-exit methods, such as FlashThink and TrimR, require additional verifiers (usually at least 7B), introducing extra GPU memory usage and communication load.

In conclusion, ReBalance achieves outstanding performance in tokens per second, time per request, additional GPU memory usage, and generated sequence length.

## J  THE USE OF LARGE LANGUAGE MODELS

In this work, large language models are used exclusively for polishing the writing and checking grammar. They are not involved in research ideation, experimental design, data analysis, or the formulation of conclusions. All substantive intellectual contributions are made by the authors.

## K  ETHICS STATEMENT

We confirm that this research adheres to the ICLR Code of Ethics. Our method is purely algorithmic and theoretical in nature, involving no human subjects, personal data, or real-world deployment that could lead to harmful, biased, or discriminatory outcomes. We have carefully considered potential ethical implications—including fairness, transparency, and societal impact—and conclude that the proposed approach poses no significant ethical concerns. All experiments were conducted on publicly available datasets.

## L  REPRODUCIBILITY STATEMENT

To facilitate replication, we fix the random seed to **42** for all sampling and evaluation. All experiments are run on the same hardware ($8\times$ NVIDIA RTX PRO 6000 GPUs) with unified decoding configurations (`temperature`=0.7, `top_p`=0.95, `max_generated_tokens`=16000). Steering vectors and control surfaces are extracted once from a random sample of 500 MATH problems and then reused across all benchmarks. Together with standardized evaluation metrics (Pass@1 and average token length), these settings ensure that our results can be reproduced reliably.

# M  CASE STUDY

| Question: | Answer:$(-\sqrt{3}, \sqrt{3})$ |
|---|---|
| For what real values of $x$ is $-4 < x^{4} + 4x^{2} < 21$ satisfied? Express your answer in interval notation. | |

Generated Responses:<think>\nAlright, so I have this inequality to solve: -4 < x⁴ + 4x² < 21. Hmm, okay. It looks like a quartic inequality because of the x⁴ term. I remember that to solve such inequalities, I can break it down into two separate inequalities and solve each part separately. That is, first solve -4 < x⁴ + 4x² and then solve x⁴ + 4x² < 21. After that, I can find the intersection of the solutions from both parts to get the final answer.\n\nLet me start by rewriting the inequality:\n\n-4 < x⁴ + 4x² < 21.\n\nSo, I'll split this into two parts:\n\n1. x⁴ + 4x² > -4\n2. x⁴ + 4x² < 21\n\nStarting with the first inequality: x⁴ + 4x² > -4.\n\nHmm, x⁴ is always non-negative since any real number raised to an even power is positive or zero. Similarly, 4x² is also non-negative for the same reason. So, adding them together, x⁴ + 4x² is always greater than or equal to zero. But here, the right side is -4, which is negative. So, is it possible for x⁴ + 4x² to be greater than -4?\n\nWell, since x⁴ + 4x² is always at least zero, and zero is greater than -4, that inequality x⁴ + 4x² > -4 is always true for all real numbers x. So, the first part doesn't impose any restrictions on x. That's a relief; I don't have to worry about excluding any x from the solution set because the left inequality is always satisfied.\n\nNow, moving on to the second inequality: x⁴ + 4x² < 21.\nThis one is more challenging. Let's try to simplify or rearrange it. Maybe I can set y = x² to make it easier to handle. Since x² is always non-negative, y ≥ 0. Substituting, the inequality becomes y² + 4y < 21.\n\nSo, let me write that out:\n\ny² + 4y - 21 < 0.\n\nNow, I can solve this quadratic inequality in terms of y. First, let's find the roots of the equation y² + 4y - 21 = 0.\n\nUsing the quadratic formula: y = [-b ± √(b² - 4ac)] / 2a.\n\nHere, a = 1, b = 4, c = -21.\n\nPlugging in, we get:\n\ny = [-4 ± √(16 - 4*1*(-21))] / 2\n\n= [-4 ± √(16 + 84)] / 2\n\n= [-4 ± √100] / 2\n\n= [-4 ± 10] / 2.\n\nSo, the roots are:\n\ny = (-4 + 10)/2 = 6/2 = 3,\n\ny = (-4 - 10)/2 = (-14)/2 = -7.\n\nBut since y = x² and x² can't be negative, y can't be -7. So, the only relevant root is y = 3.\n\nNow, to solve the inequality y² + 4y - 21 < 0, we can analyze the intervals determined by the roots. Since the quadratic opens upwards (because the coefficient of y² is positive), the quadratic will be below zero between its two roots. However, one of the roots is negative (y = -7), which we can ignore because y is non-negative. So, the relevant interval is from y = -7 to y = 3. But since y can't be negative, we only consider y from 0 to 3.\n\nThus, the inequality y² + 4y - 21 < 0 holds when y is between 0 and 3, i.e., 0 < y < 3. \n\nBut remember, y = x², so substituting back, we get:\n\n0 < x² < 3.\n\nThis inequality implies that x² is between 0 and 3. To find x, we can take square roots.\n\nFirst, since x² > 0, that means x ≠ 0. But we have to consider x² < 3 as well.\n\nSo, x² < 3 implies that |x| < √3, which means x is in the interval (-√3, √3).\n\nPutting it all together, the second inequality x⁴ + 4x² < 21 is satisfied when x is between -√3 and √3, excluding zero? Wait, hold on. No, actually, since x² > 0, the inequality x² < 3 holds for all x in (-√3, √3) except x = 0. But actually, when x = 0, x⁴ + 4x² = 0, which is less than 21. So, actually, x = 0 is included. \n\nWait, let's think again. When we transformed x⁴ + 4x² < 21 into y² + 4y - 21 < 0, we found that y must be between 0 and 3. Since y = x², that translates to x² between 0 and 3. So, x² < 3, which is equivalent to |x| < √3, meaning x is in (-√3, √3). So, actually, x can be zero here because when x is zero, x⁴ + 4x² = 0, which is indeed less than 21. So, the interval is from -√3 to √3, including zero. \n\nWait, but in the first inequality, x⁴ + 4x² > -4 is always true. So, combining both inequalities, the solution is all x such that x is in (-√3, √3). Is that right?\n\nWait, but when x is zero, x⁴ + 4x² is zero, which is less than 21, so it's included. So, the solution set is all real numbers x such that x is between -√3 and √3.\n\nBut let me double-check this. Let's test a value inside the interval, say x = 1. Then, x⁴ + 4x² = 1 + 4 = 5, which is between -4 and 21. So, it's valid.\n\nWhat about x = √3? Then, x⁴ + 4x² = (√3)^4 + 4*(√3)^2 = (9) + 4*(3) = 9 + 12 = 21, which is equal to the upper limit. So, x = √3 is not included because the inequality is strict (< 21). Similarly, x = -√3 is also not included.\n\nWhat about x = 2? Then, x⁴ + 4x² = 16 + 16 = 32, which is greater than 21, so it's excluded. So, indeed, the solution is x between -√3 and √3, not including the endpoints.\n\nWait, but let me check x = 0.5, which is inside the interval. x⁴ + 4x² = (0.5)^4 + 4*(0.5)^2 = 0.0625 + 4*(0.25) = 0.0625 + 1 = 1.0625, which is between -4 and 21, so it's valid.\n\nWhat about x = 1.5, which is still less than √3 (since √3 ≈ 1.732). x⁴ + 4x² = (1.5)^4 + 4*(1.5)^2 = 5.0625 + 4*(2.25) = 5.0625 + 9 = 14.0625, which is still less than 21, so it's valid.\n\nHow about x = √3 ≈ 1.732? As we saw earlier, x⁴ + 4x² = 21, so it's exactly 21, which isn't included. Similarly, x = -√3 gives 21 as well.\n\nSo, putting it all together, the solution is all real numbers x such that -√3 < x < √3.\n\nBut wait, let me think again about the first inequality. Since x⁴ + 4x² is always greater than or equal to zero, which is greater than -4, so the first inequality doesn't restrict x at all. So, the only restriction comes from the second inequality.\n\nTherefore, combining both inequalities, the solution is x ∈ (-√3, √3).\n\nBut just to make sure, let me consider if there are any points outside of this interval that satisfy both inequalities. For example, x = 2, which is outside, as we saw, gives x⁴ + 4x² = 32, which is greater than 21, so it's excluded. Similarly, x = -2 gives the same result. What about x = 0, as we saw, gives 21, which is excluded. x = 0, gives 0, which is included.\n\nSo, yeah, it seems that the only x that satisfy both inequalities are those in the interval (-√3, √3). Therefore, in interval notation, that would be (-√3, √3).\n\nWait, just to double-check, is there any way for x⁴ + 4x² to be less than or equal to -4? Since x⁴ + 4x² is always non-negative, it can't be less than -4. So, the only time when it's less than 21 is when it's between 0 and 21. But x⁴ + 4x² is 0 at x = 0, and as x increases or decreases, it grows. So, when does it reach 21? At x = ±√3, as we saw.\n\nTherefore, the solution is indeed x ∈ (-√3, √3). So, in interval notation, that's written as (-√3, √3).\n\nBut wait, just to make sure, let me consider the quadratic in y. So, we had y² + 4y - 21 < 0, with y = x² ≥ 0. We found that y is between 0 and 3, which corresponds to x between -√3 and √3.\n\nYes, that seems consistent.\n\nSo, I think I can confidently say that the solution is all real numbers x such that x is between -√3 and √3.\n\n**Final Answer**\nThe solution is \\boxed{(-\\sqrt{3}, \\sqrt{3})}.\n</think>......Therefore, the solution to the inequality \\(-4 < x^4 + 4x^2 < 21\\) is:\n\\[\n\\boxed{(-\\sqrt{3}, \\sqrt{3})}\n\\]\n\n

Correct Answer, But answer very long.(1659tokens)

**Figure 14:** A DeepSeek-R1-Distill-Qwen-1.5B inference case shows that, although the model arrives at the correct solution, its reasoning is rife with redundancy and repetition. Beyond the necessary boundary checks, it inspects numerous inconsequential points, incurring substantial token overhead.

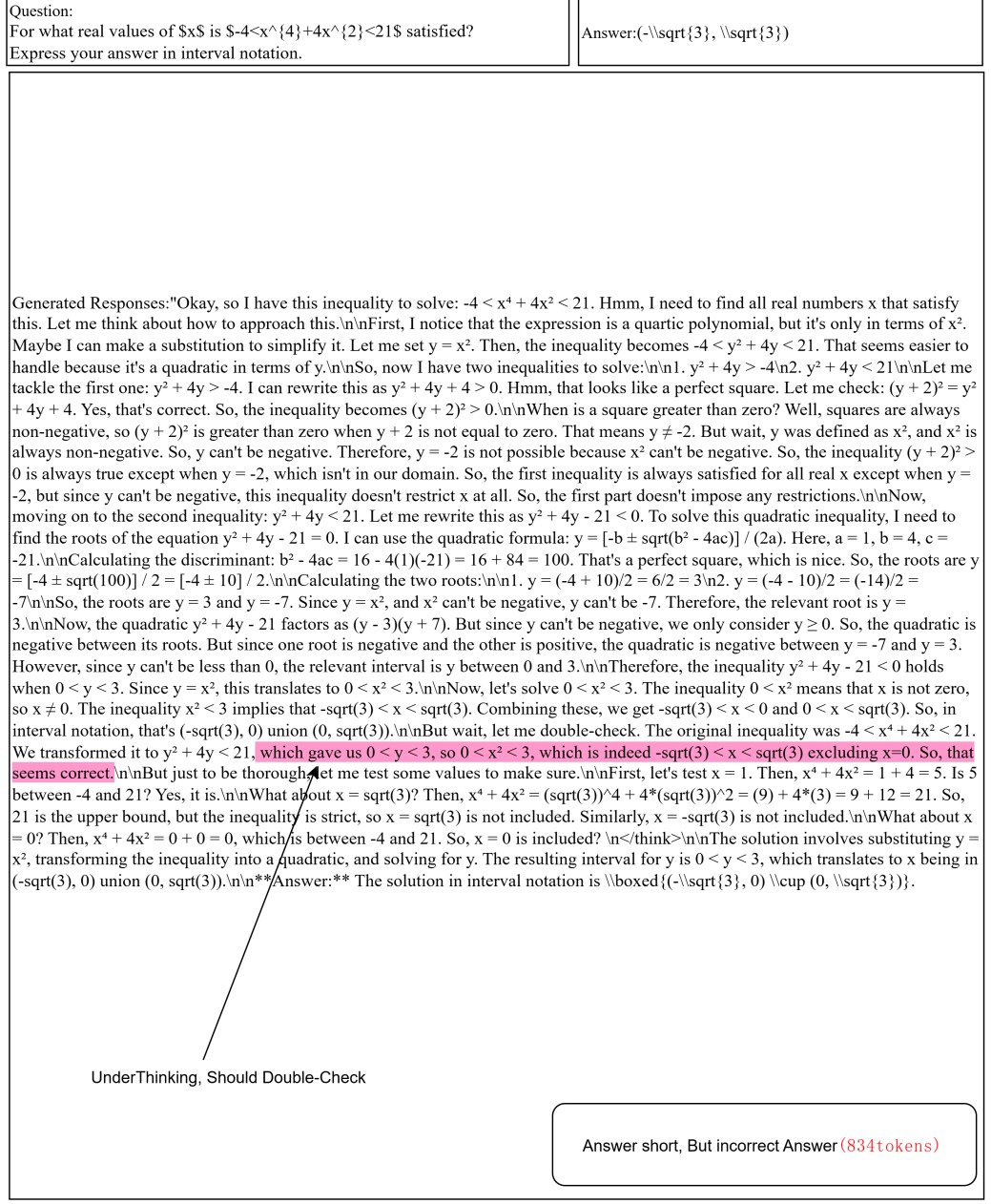

Question:
For what real values of $x$ is $-4<x^{4}+4x^{2}<21$ satisfied?
Express your answer in interval notation.

Answer:$(-\sqrt{3}, \sqrt{3})$

Generated Responses:"Okay, so I have this inequality to solve: $-4 < x^4 + 4x^2 < 21$. Hmm, I need to find all real numbers x that satisfy this. Let me think about how to approach this.\n\nFirst, I notice that the expression is a quartic polynomial, but it's only in terms of $x^2$. Maybe I can make a substitution to simplify it. Let me set $y = x^2$. Then, the inequality becomes $-4 < y^2 + 4y < 21$. That seems easier to handle because it's a quadratic in terms of y.\n\nSo, now I have two inequalities to solve:\n\n1. $y^2 + 4y > -4$\n2. $y^2 + 4y < 21$\n\nLet me tackle the first one: $y^2 + 4y > -4$. I can rewrite this as $y^2 + 4y + 4 > 0$. Hmm, that looks like a perfect square. Let me check: $(y + 2)^2 = y^2 + 4y + 4$. Yes, that's correct. So, the inequality becomes $(y + 2)^2 > 0$.\n\nWhen is a square greater than zero? Well, squares are always non-negative, so $(y + 2)^2$ is greater than zero when $y + 2$ is not equal to zero. That means $y \neq -2$. But wait, y was defined as $x^2$, and $x^2$ is always non-negative. So, y can't be negative. Therefore, $y = -2$ is not possible because $x^2$ can't be negative. So, the inequality $(y + 2)^2 > 0$ is always true except when $y = -2$, which isn't in our domain. So, the first inequality is always satisfied for all real x except when $y = -2$, but since y can't be negative, this inequality doesn't restrict x at all. So, the first part doesn't impose any restrictions.\n\nNow, moving on to the second inequality: $y^2 + 4y < 21$. Let me rewrite this as $y^2 + 4y - 21 < 0$. To solve this quadratic inequality, I need to find the roots of the equation $y^2 + 4y - 21 = 0$. I can use the quadratic formula: $y = [-b \pm sqrt(b^2 - 4ac)] / (2a)$. Here, $a = 1$, $b = 4$, $c = -21$.\n\nCalculating the discriminant: $b^2 - 4ac = 16 - 4(1)(-21) = 16 + 84 = 100$. That's a perfect square, which is nice. So, the roots are y = $[-4 \pm sqrt(100)] / 2 = [-4 \pm 10] / 2$.\n\nCalculating the two roots:\n\n1. $y = (-4 + 10)/2 = 6/2 = 3$\n2. $y = (-4 - 10)/2 = (-14)/2 = -7$\n\nSo, the roots are $y = 3$ and $y = -7$. Since $y = x^2$, and $x^2$ can't be negative, y can't be -7. Therefore, the relevant root is $y = 3$.\n\nNow, the quadratic $y^2 + 4y - 21$ factors as $(y - 3)(y + 7)$. But since y can't be negative, we only consider $y \geq 0$. So, the quadratic is negative between its roots. But since one root is negative and the other is positive, the quadratic is negative between $y = -7$ and $y = 3$. However, since y can't be less than 0, the relevant interval is y between 0 and 3.\n\nTherefore, the inequality $y^2 + 4y - 21 < 0$ holds when $0 < y < 3$. Since $y = x^2$, this translates to $0 < x^2 < 3$.\n\nNow, let's solve $0 < x^2 < 3$. The inequality $0 < x^2$ means that x is not zero, so $x \neq 0$. The inequality $x^2 < 3$ implies that $-sqrt(3) < x < sqrt(3)$. Combining these, we get $-sqrt(3) < x < 0$ and $0 < x < sqrt(3)$. So, in interval notation, that's $(-sqrt(3), 0)$ union $(0, sqrt(3))$.\n\nBut wait, let me double-check. The original inequality was $-4 < x^4 + 4x^2 < 21$. We transformed it to $y^2 + 4y < 21$, which gave us $0 < y < 3$, so $0 < x^2 < 3$, which is indeed $-sqrt(3) < x < sqrt(3)$ excluding x=0. So, that seems correct.\n\nBut just to be thorough, let me test some values to make sure.\n\nFirst, let's test $x = 1$. Then, $x^4 + 4x^2 = 1 + 4 = 5$. Is 5 between -4 and 21? Yes, it is.\n\nWhat about $x = sqrt(3)$? Then, $x^4 + 4x^2 = (sqrt(3))^4 + 4*(sqrt(3))^2 = (9) + 4*(3) = 9 + 12 = 21$. So, 21 is the upper bound, but the inequality is strict, so $x = sqrt(3)$ is not included. Similarly, $x = -sqrt(3)$ is not included.\n\nWhat about $x = 0$? Then, $x^4 + 4x^2 = 0 + 0 = 0$, which is between -4 and 21. So, $x = 0$ is included? \n</think>\n\nThe solution involves substituting $y = x^2$, transforming the inequality into a quadratic, and solving for y. The resulting interval for y is $0 < y < 3$, which translates to x being in $(-sqrt(3), 0)$ union $(0, sqrt(3))$.\n\n**Answer:** The solution in interval notation is $\boxed{(-\sqrt{3}, 0) \cup (0, \sqrt{3})}$.

UnderThinking, Should Double-Check

Answer short, But incorrect Answer(834 tokens)

**Figure 15:** In a DeepSeek-R1-Distill-Qwen-1.5B inference case, applying existing overthinking-mitigation techniques reduces token usage relative to the baseline; however, the absence of verification steps results in an incorrect answer.

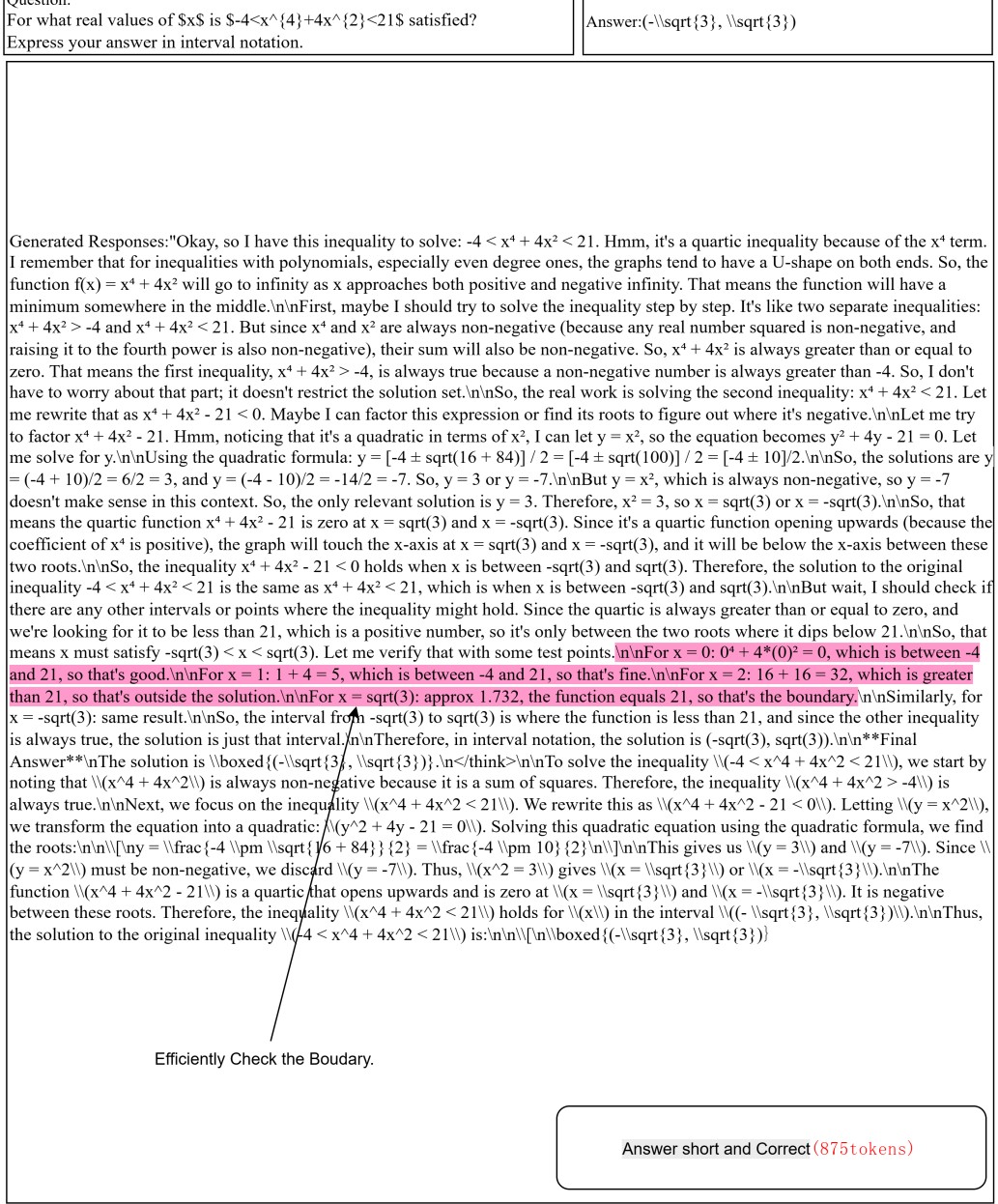

Question:
For what real values of $x$ is $-4<x^{4}+4x^{2}<21$ satisfied?
Express your answer in interval notation.

Answer:$(-\sqrt{3}, \sqrt{3})$

Generated Responses:"Okay, so I have this inequality to solve: -4 < x⁴ + 4x² < 21. Hmm, it's a quartic inequality because of the x⁴ term. I remember that for inequalities with polynomials, especially even degree ones, the graphs tend to have a U-shape on both ends. So, the function f(x) = x⁴ + 4x² will go to infinity as x approaches both positive and negative infinity. That means the function will have a minimum somewhere in the middle.\n\nFirst, maybe I should try to solve the inequality step by step. It's like two separate inequalities: x⁴ + 4x² > -4 and x⁴ + 4x² < 21. But since x⁴ and x² are always non-negative (because any real number squared is non-negative, and raising it to the fourth power is also non-negative), their sum will also be non-negative. So, x⁴ + 4x² is always greater than or equal to zero. That means the first inequality, x⁴ + 4x² > -4, is always true because a non-negative number is always greater than -4. So, I don't have to worry about that part; it doesn't restrict the solution set.\n\nSo, the real work is solving the second inequality: x⁴ + 4x² < 21. Let me rewrite that as x⁴ + 4x² - 21 < 0. Maybe I can factor this expression or find its roots to figure out where it's negative.\n\nLet me try to factor x⁴ + 4x² - 21. Hmm, noticing that it's a quadratic in terms of x², I can let y = x², so the equation becomes y² + 4y - 21 = 0. Let me solve for y.\n\nUsing the quadratic formula: y = [-4 ± sqrt(16 + 84)] / 2 = [-4 ± sqrt(100)] / 2 = [-4 ± 10]/2.\n\nSo, the solutions are y = (-4 + 10)/2 = 6/2 = 3, and y = (-4 - 10)/2 = -14/2 = -7. So, y = 3 or y = -7.\n\nBut y = x², which is always non-negative, so y = -7 doesn't make sense in this context. So, the only relevant solution is y = 3. Therefore, x² = 3, so x = sqrt(3) or x = -sqrt(3).\n\nSo, that means the quartic function x⁴ + 4x² - 21 is zero at x = sqrt(3) and x = -sqrt(3). Since it's a quartic function opening upwards (because the coefficient of x⁴ is positive), the graph will touch the x-axis at x = sqrt(3) and x = -sqrt(3), and it will be below the x-axis between these two roots.\n\nSo, the inequality x⁴ + 4x² - 21 < 0 holds when x is between -sqrt(3) and sqrt(3). Therefore, the solution to the original inequality -4 < x⁴ + 4x² < 21 is the same as x⁴ + 4x² < 21, which is when x is between -sqrt(3) and sqrt(3).\n\nBut wait, I should check if there are any other intervals or points where the inequality might hold. Since the quartic is always greater than or equal to zero, and we're looking for it to be less than 21, which is a positive number, so it's only between the two roots where it dips below 21.\n\nSo, that means x must satisfy -sqrt(3) < x < sqrt(3). Let me verify that with some test points.\n\n\n\nFor x = 0: 0⁴ + 4*(0)² = 0, which is between -4 and 21, so that's good.\n\nFor x = 1: 1 + 4 = 5, which is between -4 and 21, so that's fine.\n\nFor x = 2: 16 + 16 = 32, which is greater than 21, so that's outside the solution.\n\nFor x = sqrt(3): approx 1.732, the function equals 21, so that's the boundary.\n\nSimilarly, for x = -sqrt(3): same result.\n\nSo, the interval from -sqrt(3) to sqrt(3) is where the function is less than 21, and since the other inequality is always true, the solution is just that interval.\n\nTherefore, in interval notation, the solution is (-sqrt(3), sqrt(3)).\n\n**Final Answer**\nThe solution is \\boxed{(-\\sqrt{3}, \\sqrt{3})}.\n</think>\n\nTo solve the inequality \\(-4 < x^4 + 4x^2 < 21\\), we start by noting that \\(x^4 + 4x^2\\) is always non-negative because it is a sum of squares. Therefore, the inequality \\(x^4 + 4x^2 > -4\\) is always true.\n\nNext, we focus on the inequality \\(x^4 + 4x^2 < 21\\). We rewrite this as \\(x^4 + 4x^2 - 21 < 0\\). Letting \\(y = x^2\\), we transform the equation into a quadratic: \\(y^2 + 4y - 21 = 0\\). Solving this quadratic equation using the quadratic formula, we find the roots:\n\n\\[\ny = \\frac{-4 \\pm \\sqrt{16 + 84}}{2} = \\frac{-4 \\pm 10}{2}\n\\]\n\nThis gives us \\(y = 3\\) and \\(y = -7\\). Since \\(y = x^2\\) must be non-negative, we discard \\(y = -7\\). Thus, \\(x^2 = 3\\) gives \\(x = \\sqrt{3}\\) or \\(x = -\\sqrt{3}\\).\n\nThe function \\(x^4 + 4x^2 - 21\\) is a quartic that opens upwards and is zero at \\(x = \\sqrt{3}\\) and \\(x = -\\sqrt{3}\\). It is negative between these roots. Therefore, the inequality \\(x^4 + 4x^2 < 21\\) holds for \\(x\\) in the interval \\((- \\sqrt{3}, \\sqrt{3})\\).\n\nThus, the solution to the original inequality \\(-4 < x^4 + 4x^2 < 21\\) is:\n\n\\[\n\\boxed{(-\\sqrt{3}, \\sqrt{3})}
\]

Efficiently Check the Boudary.

Answer short and Correct (875 tokens)

**Figure 16:** In a DeepSeek-R1-Distill-Qwen-1.5B inference example, the incorporation of judicious verification steps yields a correct and succinct response.

