# OpenReview forum: "Efficient Reasoning with Balanced Thinking"
_ICLR.cc/2026/Conference — ICLR 2026 Poster_

### Official Review · Reviewer_4rRS · 2025-10-23

**Soundness:** 3
**Presentation:** 3
**Contribution:** 2
**Rating:** 6
**Confidence:** 5

**Summary:**

This paper proposes REBALANCE, a training-free framework that dynamically balances overthinking and underthinking in reasoning models by steering hidden representations based on confidence and variance signals. The method constructs a steering vector from a small calibration set (e.g., 500 samples) and adaptively adjusts the model’s internal states during inference. Experiments across multiple reasoning domains show improved efficiency (i.e., reducing the number of reasoning tokens) and good accuracy without additional parameter updates.

**Strengths:**

**[S1]** Using confidence to address overthinking and underthinking is interesting. Although the overall framework is complex, the method works surprisingly well in practice, which makes the idea conceptually appealing.

**[S2]** The fact that the approach operates without any parameter updates and can be instantiated with only a few hundred samples (e.g., 500) is impressive. It suggests that the proposed mechanism captures a general behavioral signal of the model rather than relying on large-scale retraining.

**[S3]** The method is applied not only to mathematical reasoning tasks but also to multiple domains, which demonstrates a degree of robustness and general applicability beyond a single task family.

**[S4]** The paper is overall well written and clearly structured, with extensive empirical validation and thoughtful analysis. The presentation is careful, and the experiments are relatively thorough.

**Weaknesses:**

**[W1]** While Figure 5 presents a reasonable ablation on the number of samples used for mean and variance estimation, the transferability of these statistics remains questionable. The current analysis is limited to sampling from MATH and evaluating within the similar domain. It would be more convincing to test more rigorously (than the current analysis in Figure 5) whether the mean/variance computed from easier datasets (e.g., GSM8K) transfer to harder domains (e.g., AIME), or whether math-derived statistics generalize to different modalities such as code reasoning. Without this, the generalization claim remains somewhat weak.

**[W2]** The method feels overly complex due to the number of hyperparameters involved. Decisions such as which layer to select, how strongly to apply steering, or what window size to use are not clearly motivated. It remains unclear whether there exists a single configuration that works consistently across datasets or domains, or whether each model and domain effectively requires its own tuning. This undermines the claim of universality and adds to the engineering-heavy impression.

**[W3]** The paper is somewhat engineering-driven rather than conceptually driven. Most design choices (confidence thresholds, gating, control surfaces) are empirically fitted, and it is not clear why or when the method should work beyond the tested settings. Thus, the approach lacks principled understanding of why such steering in hidden space improves reasoning. The steering direction is defined by empirical prototypes, but the underlying mechanism or geometry of reasoning states remains speculative. From my experience with steering-based approaches, this line of work is still at a very early exploratory stage. The paper demonstrates that something “can” work, but not really why it works. While the results are decent, the contribution feels somewhat ad-hoc and might not generalize to models or tasks with different internal confidence landscapes.

**[Overall]** I don’t find a strong reason to reject this paper. However, it also doesn’t give strong conceptual insights; it reads as an engineering effort that happens to work well rather than a principled step forward in understanding reasoning control. I would lean toward a weak accept, mostly for its empirical completeness and potential to inspire future, more principled research in steering-based reasoning.

**Questions:**

I think it is good to add the exact throughput analysis and memory usage analysis for the efficiency claim.

---

> ### Author Response · Authors · 2025-11-22
> **Response to Weakness 1 (1/2) (Difficulty and Domain Transferability)**
>
> >**Weakness 1**: While Figure 5 presents a reasonable ablation on the number of samples used for mean and variance estimation, the transferability of these statistics remains questionable. The current analysis is limited to sampling from MATH and evaluating within the similar domain. It would be more convincing to test more rigorously (than the current analysis in Figure 5) whether the mean/variance computed from easier datasets (e.g., GSM8K) transfer to harder domains (e.g., AIME), or whether math-derived statistics generalize to different modalities such as code reasoning. Without this, the generalization claim remains somewhat weak.
>
> Thank you for your valuable suggestion, which enables us to more comprehensively validate and clarify the broad applicability of ReBalance.
>
> **(1) Clarification of the cross-domain transfer setup in the paper**
>
> In our current experiments (Table 2), to evaluate the generalization capability of steering vectors and confidence statistics extracted from a single small-scale seen dataset across diverse domains, we have transferred the steering vector and confidence statistics derived from the mathematics domain (MATH) to science (GPQA-D), commonsense (StrategyQA), and coding (LiveCodeBench) tasks. Extensive validation has been conducted using 4 models across 3 model families (1.5B to 32B), covering the math-to-coding transfer setting requested in your review.
>
> Additionally, in Figure 5(d), we have further validated our approach by extracting the steering vector and confidence statistics from the science domain (GPQA-D) and applying them to mathematics (MATH-500), commonsense (StrategyQA), and coding (LiveCodeBench) tasks.
>
> Across all these cross-domain transfer settings, ReBalance consistently improves model accuracy (by up to 53.9%) while effectively reducing reasoning length (by up to 47.5%), demonstrating strong performance and robustness in the cross-domain scenarios.
>
> **Broader cross-domain transfer experiments.** To further address potential concerns, we introduce additional cross-domain evaluations on DeepSeek-R1-Distill-Qwen-1.5B. Specifically, we extract steering vectors and confidence statistics from coding (LiveCodeBench) and commonsense (StrategyQA) tasks and transfer them to the mathematics domain (MATH-500).
>
> We also report the quartiles of the confidence distribution (q25c, q75c) and variance distribution (q25v, q75v) derived from each extraction dataset. For readability, confidence quartiles are scaled by 100 and variance quartiles by 1,000.
>
> | Methods | Extraction Dataset | q25c | q75c | q25v | q75v | MATH-500 (Pass@1) | MATH-500 (#Tokens) |
> | --- | --- | --- | --- | --- | --- | --- | --- |
> | Baseline | - | - | - | - | - | 79.6 | 4516 |
> | ReBalance | MATH | 66.8 | 93.9 | 0.5 | 11.0 | 83.0 | **3474** |
> | ReBalance | LiveCodeBench | 62.4 | 91.2 | 0.5 | 6.2 | 82.0 | 3482 |
> | ReBalance | StrategyQA | 61.2 | 85.8 | 0.4 | 8.3 | **83.4** | 3667 |
>
> As observed, although the confidence statistics differ to some extent across domains, ReBalance achieves strong performance in all cases. Notably, extractions from StrategyQA can even achieve higher Pass@1 scores compared to those from MATH within the same domain.
>
> **(2) Cross-difficulty Transfer Experiments**
>
> In order to perform the analysis regarding cross-difficulty transferability, following your suggestion, we first conduct a transfer experiment from GSM8K to AIME on DeepSeek-R1-Distill-Qwen-1.5B. The quartile metrics for confidence statistics and their scaling factors remain consistent with those reported above.
>
> | Methods | Extraction Dataset | q25c | q75c | q25v | q75v | AIME24 (Pass@1) | AIME24 (#Tokens) | AIME25 (Pass@1) | AIME25 (#Tokens) |
> | --- | --- | --- | --- | --- | --- | --- | --- | --- | --- |
> | Baseline | - | - | - | - | - | 23.3 | 12596 | 16.7 | 14556 |
> | ReBalance | MATH | 66.8 | 93.9 | 0.5 | 11.0 | **36.7** | **9040** | **30.0** | **8140** |
> | ReBalance | GSM8K | 76.9 | 94.4 | 0.3 | 6.7 | 33.3 | 9548 | 26.7 | 8740 |
>
> We find that, although this transfer still significantly outperforms the baseline, the improvement is less pronounced compared to the original in-domain setting. To investigate the underlying reasons, we conduct a more comprehensive and fine-grained analysis.

---

> ### Author Response · Authors · 2025-11-22
> **Response to Weakness 1 (2/2) (Difficulty and Domain Transferability)**
>
> Specifically, we categorize commonly used mathematical reasoning datasets by difficulty level: *easy* (GSM8K, AMC23), *medium* (MATH/MATH-500), and *hard* (AIME24, AIME25, Olympiad). Using the medium-difficulty datasets as an intermediary, we examine two transfer directions, **easy-to-medium** and **hard-to-medium,** to explore how changes in task difficulty affect the transfer performance of ReBalance. The experimental results on DeepSeek-R1-Distill-Qwen-1.5B are shown below.
>
> | Methods | Extraction Dataset | q25c | q75c | q25v | q75v | MATH-500 (Pass@1) | MATH-500 (#Tokens) |
> | --- | --- | --- | --- | --- | --- | --- | --- |
> | Baseline | - | - | - | - | - | 79.6 | 4516 |
> | ReBalance | MATH | 66.8 | 93.9 | 0.5 | 11.0 | 83.0 | 3474 |
> | ReBalance | GSM8K | 76.9 | 94.4 | 0.3 | 6.7 | 80.6 | **3221** |
> | ReBalance | AMC23 | 61.8 | 94.2 | 0.4 | 10.4 | 82.8 | 3277 |
> | ReBalance | AIME24 | 63.2 | 92.1 | 0.5 | 9.8 | 83.2 | 3568 |
> | ReBalance | AIME25 | 58.9 | 90.2 | 0.4 | 9.3 | **84.0** | 3796 |
> | ReBalance | Olympiad | 58.1 | 84.2 | 0.5 | 9.5 | 83.6 | 3862 |
>
> However, data distributions vary across different datasets, which may introduce confounding factors beyond just difficulty. Moreover, it is challenging to precisely quantify the difficulty gaps between datasets, posing significant challenges to the analysis. Therefore, we leverage the ground-truth difficulty grading within the MATH dataset and conduct an analysis based on QwQ-32B, performing an in-distribution, fine-grained difficulty classification from Level 1 to Level 5 within the same dataset.
>
> | Methods | Extraction Dataset | q25c | q75c | q25v | q75v | MATH-500 (Pass@1) | MATH-500 (#Tokens) |
> | --- | --- | --- | --- | --- | --- | --- | --- |
> | Baseline | - | - | - | - | - | 94.8 | 4535 |
> | ReBalance | MATH | 70.6 | 92.0 | 0.4 | 7.3 | 95.2 | 3662 |
> | ReBalance | MATH (Level-1) | 74.8 | 92.8 | 0.3 | 5.2 | 94.8 | **3447** |
> | ReBalance | MATH (Level-2) | 72.4 | 92.7 | 0.3 | 6.9 | 94.8 | 3623 |
> | ReBalance | MATH (Level-3) | 71.0 | 92.1 | 0.3 | 6.9 | 95.2 | 3678 |
> | ReBalance | MATH (Level-4) | 69.6 | 92.0 | 0.4 | 7.4 | 95.0 | 3696 |
> | ReBalance | MATH (Level-5) | 69.2 | 91.1 | 0.4 | 7.7 | **95.4** | 3720 |
>
> Based on the experiments presented in the two tables above, we have the following observations:
>
> - **The higher the difficulty of the extraction dataset, the lower the confidence and the higher the variance.** This reflects the model's broader and more frequent exploration of reasoning paths, aligning with the decision rule we proposed in Formula 5, where confidence serves as an indicator.
> - **Extraction datasets of easy difficulty prioritize token reduction, while those of hard difficulty prioritize accuracy improvement.** We believe this occurs because easy extraction datasets exhibit higher confidence and lower variance; thus, according to Formula 5, the range classified as overthinking is wider, and the underthinking range is narrower. This results in more frequent triggering of the overthinking criterion, causing the dynamic control function to preferentially suppress redundant reasoning. The opposite occurs for harder datasets.
>
> Therefore, we suggest using extraction datasets of medium difficulty (e.g., MATH) in practical applications to achieve an optimal accuracy-efficiency trade-off. The above experimental results and analysis have been added to Appendix C.3.

---

> ### Author Response · Authors · 2025-11-22
> **Response to Weakness 2 (Generalization of Hyperparameters)**
>
> >**Weakness 2**: The method feels overly complex due to the number of hyperparameters involved. Decisions such as which layer to select, how strongly to apply steering, or what window size to use are not clearly motivated. It remains unclear whether there exists a single configuration that works consistently across datasets or domains, or whether each model and domain effectively requires its own tuning. This undermines the claim of universality and adds to the engineering-heavy impression.
>
> Below, we list all parameters in ReBalance that might raise concerns regarding universality and engineering intensity, and clarify each point individually.
>
> **(1) Steering Layer Depth**
>
> As described in Section 3.3 and detailed in Appendix A.5, our method automatically selects the layer whose hidden states exhibit the strongest correlation with confidence (based on the $R^2$ score) during the steering vector extraction process. This selected steering layer is then applied uniformly across all datasets and domains, without requiring additional inference or manual tuning.
>
> **(2) Steering Strength**
>
> As stated in Section 3.4 and detailed in Appendix B.3, both the strength and direction of steering in our method are adaptively controlled during reasoning by the proposed dynamic control function, also eliminating the need for manual tuning. Specifically, this dynamic function:
>
> - Tracks reasoning states in real time through confidence indicators during inference;
> - Dynamically adjusts the steering strength at each reasoning step, proportional to the detected tendency toward overthinking or underthinking;
> - Dynamically modulates the steering direction at each reasoning step based on the detected tendency type: negative for overthinking, and positive for underthinking.
>
> **(3) Dynamic Control Function Parameters**
>
> The complete set of parameters in the dynamic control function, along with their respective setting methods, is summarized and compared in the table below. Among them, only one hyperparameter $B_u$ requires manual tuning. However, we maintain the same value of $B_u$ across all models and datasets, demonstrating the robustness. As described in Section 3.4 and detailed in Appendix B.3, all other parameters are *directly adaptively derived from the model’s behavior during steering vector extraction*, without the need for additional inference steps or manual tuning.
>
> | Dynamic Control Function Parameters | Underthinking Boundary $B_u$ | Overthinking Boundary $B_o$ | Moderate Boundary $B_m$ | Lower Confidence Threshold $\tau_c^{\mathrm{L}}$ |  Higher Confidence Threshold $\tau_c^{\mathrm{H}}$ | Lower Variance Threshold $\tau_v^{\mathrm{L}}$ |  Higher Variance Threshold $\tau_v^{\mathrm{H}}$ |
> | --- | --- | --- | --- | --- | --- | --- | --- |
> | Is Hyperparameter | Yes | No | No | No | No | No | No |
> | Setting Method | **Fixed identical values** across all models and benchmarks | **Automatically** computed based on aggressive hidden state displacement from overthinking to underthinking space | **Automatically** computed based on moderate hidden state displacement from overthinking/underthinking to underthinking/overthinking space | **Automatically** elected based on quartiles of model confidence distribution | **Automatically** elected based on quartiles of model confidence distribution | **Automatically** elected based on quartiles of model confidence distribution | **Automatically** elected based on quartiles of model confidence distribution |
>
> **(4) Sliding Window Size**
>
> As stated in Section 4.2, we demonstrate in Appendix A.3 that the confidence trajectory during inference satisfies a Markovian continuity assumption. Consequently, a small window size ($|\mathcal{W}_s| = 2$) is sufficiently expressive and more sensitive to local reasoning patterns. Therefore, we *fix the window size to 2 across all experiments*, eliminating the need for manual tuning.
>
> ***Does there exist a single configuration that works consistently across models, datasets, and domains?***
>
> **Yes.** As shown in the above analysis, the parameters involved in our method fall into two categories:
>
> - **Parameters that can be automatically fitted.** These are fitted concurrently with steering vector extraction, requiring no additional inference phase. The fitting logic is unified across all models and needs to be performed only once per model. Once fitted, these parameters generalize seamlessly to all datasets and domains.
> - **Hyperparameters that cannot be automatically fitted.** These values are kept fixed across all models, datasets, and domains. Therefore, the same configuration adapts to diverse scenarios without requiring additional manual tuning per setting.
>
> Consequently, our method requires minimal manual intervention and, after an automatic initialization on a lightweight seen dataset (e.g., 500 samples) for each model, can be broadly applied across diverse datasets and domains.

---

> ### Author Response · Authors · 2025-11-22
> **Response to Weakness 3 (1/2) (Generalization and Conceptual Clarity)**
>
> >**Weakness 3**: The paper is somewhat engineering-driven rather than conceptually driven. Most design choices (confidence thresholds, gating, control surfaces) are empirically fitted, and it is not clear why or when the method should work beyond the tested settings. Thus, the approach lacks principled understanding of why such steering in hidden space improves reasoning. The steering direction is defined by empirical prototypes, but the underlying mechanism or geometry of reasoning states remains speculative. From my experience with steering-based approaches, this line of work is still at a very early exploratory stage. The paper demonstrates that something “can” work, but not really why it works. While the results are decent, the contribution feels somewhat ad-hoc and might not generalize to models or tasks with different internal confidence landscapes.
>
> Thank you for your insightful review. Below is our detailed response to your concerns.
>
> **(1) Concern regarding whether our approach is engineering-driven or conceptually-driven**
>
> As stated in our response to Weakness 2, and further validated by the experiments in our response to Weakness 1, our method achieves strong performance across various models, datasets, and domains without requiring manual tuning. Such broad and robust generalization is more likely to be achieved by a conceptually-driven approach rather than an engineering-driven one, to ensure the pipeline is agnostic to different scenarios.
>
> Additionally, since the gating function mentioned in this weakness was not discussed in our earlier analysis, we provide a detailed clarification as follows. Specifically, as stated in Section 3.4 and detailed in Appendix B.3, the gating function is used to produce a smoother fitted surface and avoid abrupt changes in steering strength. We chose the sigmoid function simply because it is commonly used, not due to any special engineering tuning. To validate this claim, we replace the gating function with several other alternative functions and conduct experiments across various models, difficulty levels, and domains, as detailed in the table below.
>
> *DeepSeek-R1-Distill-Qwen-1.5B*
>
> | Gating Functions | MATH-500 (Pass@1) | MATH-500 (#Tokens) | GSM8K (Pass@1) | GSM8K (#Tokens) | Olympiad (Pass@1) | Olympiad (#Tokens) | GPQA-D (Pass@1) | GPQA-D (#Tokens) |
> | --- | --- | --- | --- | --- | --- | --- | --- | --- |
> | Baseline | 79.6 | 4516 | 76.0 | 1018 | 41.2 | 8785 | 17.1 | 8727 |
> | Sigmoid | 83.0 | 3474 | 78.3 | **765** | 43.9 | 7235 | **21.7** | 6902 |
> | Linear | 83.6 | 3600 | 79.2 | 800 | 42.5 | **6794** | 17.7 | 7019 |
> | HardStep | 81.2 | 3830 | 79.7 | 820 | 41.8 | 7528 | 17.2 | 6537 |
> | Polynomial | 82.8 | 3508 | 79.0 | 798 | **44.2** | 7059 | 20.7 | **6161** |
> | ReLU-Like | **84.0** | **3277** | **80.1** | 818 | 43.0 | 6875 | 19.7 | 6966 |
>
> *DeepSeek-R1-Distill-Qwen-7B*
>
> | Gating Functions | MATH-500 (Pass@1) | MATH-500 (#Tokens) | GSM8K (Pass@1) | GSM8K (#Tokens) | Olympiad (Pass@1) | Olympiad (#Tokens) | GPQA-D (Pass@1) | GPQA-D (#Tokens) |
> | --- | --- | --- | --- | --- | --- | --- | --- | --- |
> | Baseline | 89.8 | 3699 | 89.2 | 1098 | 56.1 | 7590 | 33.8 | 7392 |
> | Sigmoid | **92.6** | **2903** | 91.6 | 912 | 57.0 | 6321 | 39.4 | **5180** |
> | Linear | 91.2 | 3113 | 90.0 | 936 | 56.2 | 6303 | **43.0** | 5589 |
> | HardStep | 92.0 | 3115 | 90.0 | **909** | **57.8** | 6319 | 40.4 | 5448 |
> | Polynomial | 91.6 | 2932 | 91.3 | 913 | 56.0 | **6206** | 38.9 | 5768 |
> | ReLU-Like | 91.4 | 3054 | **91.7** | 928 | 57.0 | 6362 | 39.9 | 5758 |
>
> It can be observed that, though the sigmoid function yields significant performance gain, it might not be the optimal choice. With additional engineering efforts, the method’s performance could be further improved. However, as the specific design is not the focus of this work, we did not carefully tune our method and adopt the sigmoid function as the default setting.

---

> ### Author Response · Authors · 2025-11-22
> **Response to Weakness 3 (2/2) (Generalization and Conceptual Clarity)**
>
> **(2) Concern regarding the Lack of principled understanding of the reasoning state’s underlying mechanism**
>
> Although ReBalance demonstrates solid performance across various scenarios, the core is to propose the concept of balanced thinking that is driven by the model’s inherent behavioral dynamics. To clarify the underlying mechanism of this reasoning state, we provide detailed discussions and theoretical analyses in both the main paper and the appendix. Below, we briefly summarize these key insights:
>
> - **Confidence serves as an indicator of overthinking/underthinking.** Through quantitative analysis in Figure 2(b), we have found that samples exhibiting overthinking or underthinking display significantly different distributions in confidence and variance compared to normal samples.
> - **Correlation between confidence and reasoning length.** As shown in Appendix A.2 and Appendix Table 4, Spearman correlation coefficients reveal a significant negative correlation between reasoning length and the minimum stepwise confidence, and a significant positive correlation between reasoning length and confidence variance.
> - **Markov persistence of confidence states.** In Appendix A.3, we demonstrate that the confidence trajectory during inference satisfies a Markovian continuity assumption (Appendix Table 5). Consequently, a small window size ($|\mathcal{W}_s| = 2$) is sufficiently expressive and more sensitive to local reasoning patterns.
> - **Distributional heterogeneity in model confidence.** In Appendix A.4, we visualize the confidence distributions of different models (Appendix Figure 7) and observe substantial inter-model variation, highlighting the necessity of a reasoning control method that can adaptively adjust according to the behavioral characteristics of each specific model.
> - **Encoding of confidence signals in latent representations.** As shown in Appendix A.5, our probing experiments reveal that the hidden representations of reasoning models contain linearly decodable signals of confidence, with the strongest encoding observed in middle-to-late layers (Appendix Figure 9). Furthermore, t-SNE visualizations demonstrate that confidence patterns are clearly distinguishable in the latent space (Appendix Figure 8).
> - **Confidence as evidence under vocabulary coverage gaps.** As shown in Appendix A.6, we aggregate keyword vocabularies from current efficient reasoning methods to uncover the mechanisms behind their effectiveness. Analysis of the correlation between keyword frequency and confidence (Appendix Table 8) reveals that vocabulary-based methods are essentially incomplete variants of confidence-based approaches, partially capturing low-confidence patterns.
>
> We hope that these analyses of the reasoning state’s underlying mechanisms could provide valuable insights for inspiring future research.
>
> **(3) Concern regarding the generalization across different internal confidence landscapes**
>
> To provide a thorough evaluation of generalization, we decompose the potential sources of varying confidence distributions into three dimensions:
>
> - Transfer to different models;
> - Transfer to datasets of varying difficulty or domain;
> - Decoding the same model under different temperature settings.
>
> The first two dimensions have been validated in Table 1 of the main paper and in our response to Weakness 2, respectively. To evaluate the third dimension, we extract steering vectors and confidence statistics under a temperature setting of 0.7, and then conduct experiments on DeepSeek-R1-Distill-Qwen-1.5B over MATH-500 using various decoding temperatures. We report the quartiles of the confidence distribution (q25c, q75c) and variance distribution (q25v, q75v) as representations of the confidence landscape.
>
> | Temperature | q25c | q75c | q25v | q75v | Baseline (Pass@1) | Baseline (#Tokens) | ReBalance (Pass@1) | ReBalance (#Tokens) |
> | --- | --- | --- | --- | --- | --- | --- | --- | --- |
> | 0.2 | 100.0 | 100.0 | 0.0 | 0.0 | 68.4 | 6248 | **77.0** | **4335** |
> | 0.4 | 96.0 | 100.0 | 0.0 | 0.6 | 72.6 | 5694 | **79.6** | **3906** |
> | 0.6 | 85.1 | 100.0 | 0.0 | 3.9 | 79.4 | 4584 | **81.6** | **3689** |
> | 0.8 | 71.6 | 98.5 | 0.2 | 9.8 | 81.0 | 4529 | **84.6** | **3485** |
> | 1.0 | 55.2 | 94.5 | 0.6 | 16.4 | 82.4 | 4660 | **82.4** | **3488** |
>
> It can be observed that ReBalance consistently delivers significant improvements in both accuracy and efficiency across different confidence distributions induced by varying decoding temperatures.
>
> Thank you for your insightful feedback. The above experimental results and analysis have been added to Appendix C.5.

---

> ### Author Response · Authors · 2025-11-22
> **Response to Question 1 (Efficiency Analysis)**
>
> >**Question 1**: I think it is good to add the exact throughput analysis and memory usage analysis for the efficiency claim.
>
> The efficiency analysis on AIME24 using DeepSeek-R1-Distill-Qwen-7B is shown in the table below. To ensure a thorough comparison, we include prompt-based methods (NoThinking, CoD), early-exit methods (FlashThink, TrimR), and latent steering methods (Seal). Evaluation metrics include tokens per second (TPS), time per request (TPR), and additional GPU memory usage relative to the baseline.
>
> | Methods | TPS | TPR(s) | Additional GPU Memory(GB) | Pass@1 | #Tokens |
> | --- | --- | --- | --- | --- | --- |
> | Baseline | 80.15 | 174.6 | - | 40.0 | 13994 |
> | NoThinking | 80.14 | 55.2 | 0.0 | 26.7 | 4427 |
> | CoD | 80.16 | 145.5 | 0.0 | 46.7 | 11663 |
> | FlashThink | 73.84 | 135.9 | 18.3 | 46.7 | 10034 |
> | TrimR | 48.84 | 147.7 | 15.4 | 36.7 | 7213 |
> | SEAL | 78.82 | 128.3 | 0.0 | 43.3 | 10112 |
> | ReBalance (Ours) | 78.53 | 114.8 | 0.0 | 56.7 | 9012 |
> - **Token generation efficiency:** By observing TPS, it can be seen that since the confidence utilized by ReBalance can be directly obtained from the log probability of each token's decoded output, and the dynamic function introduced in this method is very lightweight, the dynamic adjustment logic we proposed has a negligible impact on the single-token generation time compared to the baseline.
> - **Reasoning time acceleration:** As shown by TPS and #Tokens, ReBalance significantly shortens reasoning length without compromising token generation efficiency, yielding **1.5×** TPR speedups.
> - **Additional GPU memory usage:** Although ReBalance requires the use of extracted steering vectors during reasoning, their GPU memory footprint is minimal (e.g., the vector size for QwQ-32B is only 22kb). In contrast, early-exit methods, i.e., FlashThink and TrimR, require additional verifiers (usually at least 7B), introducing extra GPU memory usage and communication load.
>
> Thank you for your suggestion, and we have included detailed efficiency analysis and experimental results in Appendix I.

---

> ### Author Response · Authors · 2025-11-27
> **Looking Forward to Your Valuable Feedback**
>
> Dear Reviewer 4rRS,
>
> I hope this message finds you well. We have carefully addressed the questions and concerns you previously raised, and we would greatly appreciate any further feedback you might be able to provide. Your insights are highly valuable to us, and we remain fully available to clarify or refine any remaining points.
>
> As the discussion deadline is approaching, if possible, we would be grateful for your response at your earliest convenience, as it would greatly help us move the discussion forward.
>
> Thank you sincerely for your time and effort in reviewing our paper.
>
> Best,
>
> The Authors

---

### Official Review · Reviewer_4rhm · 2025-10-29

**Soundness:** 3
**Presentation:** 3
**Contribution:** 3
**Rating:** 8
**Confidence:** 3

**Summary:**

This paper proposes a training-free solution that balances overthinking and underthinking in reasoning models.
Based on the key observation that confidence values and their variances help distinguish these two reasoning modes, the authors introduce a representation steering method that dynamically adjusts the model’s reasoning behavior.
Experimental results demonstrate that the proposed method improves performance while reducing reasoning length.

**Strengths:**

- The motivation is clear, and the addressed problem is crucial in the reasoning field.
- The paper is well-written and easy to follow.
- The experimental results, from initial observations to ablation studies, strongly support the proposed method.
- The method is simple, easy to implement, and can be plug-and-play across various reasoning models.

**Weaknesses:**

No major weaknesses are observed, but clarification on the following questions would strengthen the paper and may affect my final score.

**Questions:**

**[Q1] Extraction of the Steering Vector**
As I understand, the authors extract the steering vector using confidence-defined sets in Eq. (5). However, directly using the definition in Eq. (3) appears simpler. Could the authors report the performance gap between these two methods?

**[Q2] Observation Across Models**
What type of model is used in Figure 2 (b)? I am curious whether confidence values and variances serve as general indicators across different models.

**[Q3] Inference Overheads** Appendix I provides only a brief latency analysis. Could the authors include a quantitative throughput evaluation (e.g., tokens per second) comparing their method with the original model?

**[Q4] Number of Samplings** How many samplings are used per test sample? Since challenge datasets such as AIME and AMC contain relatively few examples (< 100), multiple samplings may be required for more reliable evaluation.

---

> ### Author Response · Authors · 2025-11-22
> **Response to Question 1 (1/2) (Performance Gap between Steering Vector Extraction Methods)**
>
> >**Question 1**: **Extraction of the Steering Vector**  As I understand, the authors extract the steering vector using confidence-defined sets in Eq. (5). However, directly using the definition in Eq. (3) appears simpler. Could the authors report the performance gap between these two methods?
>
> Thank you for your constructive questions, which allow us to elaborate on the differences between the two implementations and the resulting performance gap.
>
> **(1) Why do we adopt Equation 5 instead of Equation 3**
>
> Although Equation 3 appears more concise and intuitive, it serves only as a *theoretical definition* of overthinking and underthinking. Its purpose is to provide a *conceptual distinction* between the two phenomena through a formalized expression. This definition operates at the *trajectory level*, classifying an entire reasoning trajectory as overthinking or underthinking by comparing it against an idealized *stability index* that acts as a decision boundary. Consequently, if Equation 3 were directly used as the indicator for steering vector extraction, it would inevitably lead to the following issues:
>
> - **Mismatch in operational granularity.** Our method aims to adaptively adjust the model’s behavior based on the real-time reasoning state at each step. This requires a *step-level*, fine-grained indicator to detect tendencies toward overthinking or underthinking. To avoid a mismatch in operational granularity, we therefore maintain the same step-level resolution during the steering vector extraction. However, Equation 3 only supports *trajectory-level* classification of reasoning modes, and this granularity mismatch may lead to suboptimal performance.
> - **Limited applicability.** The stability index requires access to ground truth, which introduces an *additional dependency on labeled data*. Although one could follow [1] using an external verifier to approximate ground truth by assessing the existence and equivalence of answers across consecutive sub-thoughts, this strategy still relies on the verifier’s capability, prompt design, and hyperparameters for determining answer equivalence. Consequently, it may incur extra engineering overhead and introduce potential performance bottlenecks.
> - **Difficulty in capturing complex reasoning dynamics.** Equation 3 defines the stability index only when all subsequent reasoning steps yield identical answers that exactly match the ground truth. However, existing studies have shown that the *accuracy of reasoning models is not positively correlated with reasoning length* [2]; on the contrary, *longer reasoning sequences may introduce more hallucinations* [3]. Therefore, while Equation 3 offers a convenient and intuitive way to conceptually distinguish overthinking from underthinking, its rigid requirement makes it unsuitable as a practical indicator in real-world scenarios, given the inherent complexity and variability of actual reasoning dynamics.
>
> **(2) Performance Analysis Based on Equation 3**
>
> To quantitatively validate the above analysis, we perform steering vector extraction using Equation 3 as follows.
>
> First, consistent with existing methods, we randomly select 500 seen samples from the MATH training set and feed them into DeepSeek-R1-Distill-Qwen-1.5B. We collect the generated output sequences and record the model’s confidence at each reasoning step.
>
> To identify steps corresponding to overthinking and underthinking, we follow [1][4] and use Qwen2.5-7B-Instruct to determine whether each step contains the ground truth (see Appendix C.10 for the prompt template). Once a step containing the ground truth is detected, it is designated as the **stability index**: all preceding steps are classified as *underthinking*, and all subsequent steps as *overthinking*. The labeling procedure produces 29,701 overthinking samples and 24,710 underthinking samples, showing that our partitioning strategy is reasonable and results in a well-balanced dataset.
>
> [1] TrimR: Verifier-based Training-Free Thinking Compression for Efficient Test-Time Scaling
>
> [2] Do not think that much for 2+3=? on the overthinking of O1-like LLMs
>
> [3] More Thinking, Less Seeing? Assessing Amplified Hallucination in Multimodal Reasoning Models
>
> [4] Flashthink: An early exit method for efficient reasoning

---

> ### Author Response · Authors · 2025-11-22
> **Response to Question 1 (2/2) (Performance Gap between Steering Vector Extraction Methods)**
>
> To analyze the relative positional distribution of stability indices within the entire thinking sequence, we divide the range of relative positions (0–1) into 10 equal intervals (bins). We then count the number of occurrences of stability indices falling into each interval.
>
> | Interval | 1 | 2 | 3 | 4 | 5 | 6 | 7 | 8 | 9 | 10 |
> | --- | --- | --- | --- | --- | --- | --- | --- | --- | --- | --- |
> | Relative Position Range | [0.0, 0.1) | [0.1, 0.2) | [0.2, 0.3) | [0.3, 0.4) | [0.4, 0.5) | [0.5, 0.6) | [0.6, 0.7) | [0.7, 0.8) | [0.8, 0.9) | [0.9, 1.0] |
> | Count of Stability Indices | **27.41** | 16.45 | 10.96 | 9.87 | 6.58 | 3.29 | 2.85 | 2.19 | 3.95 | **16.45** |
>
> Notably, an interesting observation emerges here: the stability indices exhibit a *bimodal* distribution along the thinking sequence, with pronounced concentrations around Interval 1 and 10. These two Intervals correspond to the underthinking and overthinking behaviors of reasoning models, respectively, further providing strong empirical support for the necessity of balanced thinking.
>
> The subsequent steps, including steering vector extraction, fitting of the dynamic control function, and dynamic steering during inference, are kept identical to those in ReBalance. The experimental results, along with the comparison of steering vector norms $\lVert S \rVert_2$, are shown below:
>
> | Methods | $\lVert S \rVert_2$ | MATH-500 (Pass@1) | MATH-500 (#Tokens) | AIME24 (Pass@1) | AIME24 (#Tokens) |
> | --- | --- | --- | --- | --- | --- |
> | Baseline | - | 79.6 | 4516 | 23.3 | 12596 |
> | Vector Extraction w/ Stability Index | 11.4 | 81.8 | 4715 | 20.0 | 12563 |
> | Vector Extraction w/ Confidence | **62.4** | **83.0** | **3474** | **36.7** | **9040** |
>
> We observe that the **experimental results obtained using the steering vector extracted via Equation 5 significantly outperform those of Equation 3 in both accuracy and efficiency**, strongly validating our analysis above. Moreover, the difference in $\lVert S \rVert_2$ reveals that when switching to stability index-based extraction, the norm of the steering vector becomes substantially smaller. This indicates a blurring of the boundary between overthinking and underthinking, further explaining the performance gap between the two methods. The above experimental results and analysis have been included in Appendix C.10.

---

> ### Author Response · Authors · 2025-11-22
> **Response to Question 2 (Observation Across Models)**
>
> >**Question 2**: **Observation Across Models**  What type of model is used in Figure 2 (b)? I am curious whether confidence values and variances serve as general indicators across different models.
>
> Thank you for your question. Figure 2b in the paper highlights a core contribution of our work: confidence serves as a continuous and reliable indicator for explicitly modeling overthinking and underthinking. We appreciate your comments and agree that this finding warrants a more detailed discussion.
>
> In Figure 2b, we use DeepSeek-R1-Distill-Qwen-1.5B on MATH-500 for visualization. To demonstrate the generality of this observation, we extend the experimental setup to include four model sizes (1.5B to 32B) across three distinct model families. The table below presents a comprehensive quantitative analysis of stepwise confidence and confidence variance across these models. For clarity, confidence values are scaled by 100 and variance by 1,000.
>
> *Overthinking Analysis*
>
> | Models | Confidence (Normal) | Confidence (Overthinking) | Variance (Normal) | Variance (Overthinking) | Length (Normal) | Length (Overthinking) |
> | --- | --- | --- | --- | --- | --- | --- |
> | DeepSeek-R1-Distill-Qwen-1.5B | 80.4 | 78.6 | 18.5 | 21.3 | 1357 | 2386 |
> | DeepSeek-R1-Distill-Qwen-7B | 90.0 | 81.8 | 11.0 | 23.0 | 2809 | 5995 |
> | QwQ-32B | 84.0 | 75.4 | 16.1 | 22.1 | 3712 | 6377 |
> | Qwen3-14B | 94.1 | 88.6 | 8.7 | 12.1 | 3319 | 4027 |
>
> *Underthinking Analysis*
>
> | Models | Confidence (Normal) | Confidence (Underthinking) | Variance (Normal) | Variance (Underthinking) | Length (Normal) | Length (Underthinking) |
> | --- | --- | --- | --- | --- | --- | --- |
> | DeepSeek-R1-Distill-Qwen-1.5B | 85.1 | 89.7 | 22.2 | 16.7 | 2909 | 1726 |
> | DeepSeek-R1-Distill-Qwen-7B | 82.4 | 91.2 | 22.0 | 12.0 | 3259 | 2752 |
> | QwQ-32B | 76.2 | 78.9 | 18.5 | 17.6 | 4573 | 3080 |
> | Qwen3-14B | 82.0 | 85.3 | 11.3 | 8.0 | 8146 | 4002 |
>
> We can observe that, across all models shown in the table, the results consistently align with those of  Figure 2b:
>
> - **Overthinking** samples exhibit **lower confidence** and **higher variance**, suggesting hesitant and repeated switching between reasoning paths, often leading to redundant reasoning
> - **Underthinking** samples show **higher confidence** and **lower variance**, and this persistently high confidence often causes the model to prematurely commit to an incorrect reasoning path, hindering thorough exploration.
>
> This observation further confirms that confidence can serve as a general indicator, broadly applicable across various large reasoning models. The above experimental results and analysis have been included in Appendix C.7.

---

> ### Author Response · Authors · 2025-11-22
> **Response to Question 3 (Inference Overheads)**
>
> >**Question 3**: **Inference Overheads** Appendix I provides only a brief latency analysis. Could the authors include a quantitative throughput evaluation (e.g., tokens per second) comparing their method with the original model?
>
> Thank you for your suggestions. Given that ReBalance introduces a dynamic balanced thinking mechanism, we fully agree that a comprehensive efficiency analysis is necessary. In addition to comparing tokens per second (TPS) with the baseline as requested, we also include comparisons of inference overheads with other efficient reasoning methods, and add time per request (TPR) and additional GPU memory usage relative to the baseline to more comprehensively verify the efficiency of ReBalance.
>
> Specifically, as shown in the table below, to ensure a thorough comparison, we include prompt-based methods (NoThinking, CoD), early-exit methods (FlashThink, TrimR), and latent steering methods (Seal). Different from existing methods, ReBalance preserves an effective thinking process by promoting exploration when necessary to avoid underthinking, particularly when the model faces difficult reasoning problems. As a result, it is more likely to incur efficiency overheads on challenging datasets. In the following, we adopt AIME24 for the efficiency evaluation. The results on DeepSeek-R1-Distill-Qwen-7B are as follows:
>
> | Methods | TPS | TPR(s) | Additional GPU Memory(GB) | Pass@1 | #Tokens |
> | --- | --- | --- | --- | --- | --- |
> | Baseline | 80.15 | 174.6 | - | 40.0 | 13994 |
> | NoThinking | 80.14 | 55.2 | 0.0 | 26.7 | 4427 |
> | CoD | 80.16 | 145.5 | 0.0 | 46.7 | 11663 |
> | FlashThink | 73.84 | 135.9 | 18.3 | 46.7 | 10034 |
> | TrimR | 48.84 | 147.7 | 15.4 | 36.7 | 7213 |
> | SEAL | 78.82 | 128.3 | 0.0 | 43.3 | 10112 |
> | ReBalance (Ours) | 78.53 | 114.8 | 0.0 | 56.7 | 9012 |
> - **Token generation efficiency:** By observing TPS, it can be seen that since the confidence utilized by ReBalance can be directly obtained from the log probability of each token's decoded output, and the dynamic function introduced in this method is very lightweight, the dynamic adjustment logic we proposed has a negligible impact on the single-token generation time compared to the baseline.
> - **Reasoning time acceleration:** As shown by TPS and #Tokens, ReBalance significantly shortens reasoning length without compromising token generation efficiency, yielding **1.5×** TPR speedups over DeepSeek-R1-Distill-Qwen-7B.
> - **Additional GPU memory usage:** Although ReBalance requires the use of extracted steering vectors during reasoning, their GPU memory footprint is minimal (e.g., the vector size for QwQ-32B is only 22kb). In contrast, early-exit methods, i.e., FlashThink and TrimR, require additional verifiers (usually at least 7B), introducing extra GPU memory usage and communication load.
>
> In conclusion, ReBalance achieves outstanding performance in tokens per second, time per request, additional GPU memory usage, and generated sequence length. Thank you for your suggestion, and we have included detailed efficiency analysis results and experimental details in Appendix I.

---

> ### Author Response · Authors · 2025-11-22
> **Response to Question 4 (Number of Sampling)**
>
> >**Question 4**: **Number of Samplings** How many samplings are used per test sample? Since challenge datasets such as AIME and AMC contain relatively few examples (< 100), multiple samplings may be required for more reliable evaluation.
>
> **(1) Clarification of the Experimental Setup**
>
> As shown in Table 1, all experiments in our paper are conducted using a single sample per problem, with **Pass@1** as the accuracy metric. This design is motivated by two key considerations:
>
> - Pass@1 is widely adopted in efficient reasoning research [5–7];
> - Our work targets improvements in the quality and efficiency of *individual* reasoning trajectories, for which Pass@1 offers a clearer and more direct evaluation than multi-sample metrics.
>
> To minimize variability from single-sample evaluation, we fix the random seed across all experiments, as detailed in Appendix L, ensuring full reproducibility of our results.
>
> **(2) Exploration of ReBalance’s Effectiveness in Multi-Sample Scenarios**
>
> Thank you for raising this important point. We fully agree that the effectiveness of ReBalance under a larger number of samples is indeed a topic worthy of in-depth investigation. Below is our analysis and validation.
>
> First, the core issue addressed by ReBalance is *balanced thinking*, i.e., mitigating overthinking while simultaneously preventing underthinking. Here, "underthinking" refers to cases where the model is inherently capable of solving a problem but produces an incorrect answer due to insufficient reasoning. According to the definition of Pass@20, this metric effectively grants the model multiple reasoning attempts and expanded exploration space, counting a question as correct if any one of the 20 sampled solutions is successful. Therefore, theoretically, for problems that meet the underthinking criterion, which are those that the model is truly capable of solving, the likelihood of obtaining a correct answer approaches certainty as the number of samples increases, eventually converging toward the model's capability ceiling [8]. At this point, the impact of underthinking on accuracy becomes negligible, and ReBalance primarily serves to alleviate overthinking.
>
> To validate the above analysis, as requested, we evaluate Pass@20 over 20 samples and measure the average token length of generated sequences on the few-example datasets AMC23, AIME24, and AIME25. The results are shown in the table below.
>
> *Evaluation on AMC23*
>
> | Methods | Pass@1 | #Tokens (Pass@1) | Pass@20 | #Tokens (Pass@20) |
> | --- | --- | --- | --- | --- |
> | DeepSeek-R1-Distill-Qwen-1.5B | 55.0 | 8990 | 97.5 | 7430 |
> | w/ ReBalance | **62.5** | **6812** | **97.5** | **4832** |
> | DeepSeek-R1-Distill-Qwen-7B | 75.0 | 6898 | 100.0 | 6021 |
> | w/ ReBalance | **95.0** | **4767** | **100.0** | **5264** |
> | Qwen3-14B | 95.0 | 7240 | 100.0 | 7331 |
> | w/ ReBalance | **100.0** | **5230** | **100.0** | **5124** |
> | QwQ-32B | 87.5 | 7021 | 100.0 | 7034 |
> | w/ ReBalance | **95.0** | **6064** | **100.0** | **5651** |
>
> *Evaluation on AIME24*
>
> | Methods | Pass@1 | #Tokens (Pass@1) | Pass@20 | #Tokens (Pass@20) |
> | --- | --- | --- | --- | --- |
> | DeepSeek-R1-Distill-Qwen-1.5B | 23.3 | 12596 | 63.3 | 10645 |
> | w/ ReBalance | **36.7** | **9040** | **63.3** | **9243** |
> | DeepSeek-R1-Distill-Qwen-7B | 40.0 | 13994 | 76.7 | 11249 |
> | w/ ReBalance | **56.7** | **9012** | **76.7** | **8563** |
> | Qwen3-14B | 66.7 | 10888 | 90.0 | 11367 |
> | w/ ReBalance | **73.3** | **9464** | **90.0** | **9247** |
> | QwQ-32B | 66.7 | 14342 | 86.7 | 14121 |
> | w/ ReBalance | **70.0** | **10350** | **86.7** | **9853** |
>
> *Evaluation on AIME25*
>
> | Methods | Pass@1 | #Tokens (Pass@1) | Pass@20 | #Tokens (Pass@20) |
> | --- | --- | --- | --- | --- |
> | DeepSeek-R1-Distill-Qwen-1.5B | 16.7 | 14556 | 43.3 | 10447 |
> | w/ ReBalance | **30.0** | **8140** | **46.7** | **8652** |
> | DeepSeek-R1-Distill-Qwen-7B | 26.7 | 13778 | 66.7 | 11379 |
> | w/ ReBalance | **40.0** | **9227** | **66.7** | **9019** |
> | Qwen3-14B | 56.7 | 13125 | 80.0 | 12717 |
> |w/ ReBalance|**56.7**|**11057**|**80.0**|**10381**|
> |QwQ-32B|46.7|13350|83.3|13386|
> |w/ ReBalance|**63.3**|**11575**|**83.3**|**11586**|
>
> As can be seen, consistent with our analysis, **ReBalance significantly reduces reasoning length without any degradation in model accuracy**. This demonstrates that our proposed confidence indicator accurately characterizes overthinking and underthinking. Thanks to this precise characterization, ReBalance prunes only genuinely redundant reasoning steps rather than essential or effective ones, enabling lossless and efficient sequence compression even when the model operates near its capability ceiling. We have included the experimental results and detailed analysis in Appendix C.4.
>
> [5] Seal: Steerable reasoning calibration of large language models for free
>
> [6] Dynamic early exit in reasoning models
>
> [7] Mitigating Overthinking in Large Reasoning Models via Manifold Steering
>
> [8] Reasoning with sampling: Your base model is smarter than you think

---

> > ### Comment · Reviewer_4rhm · 2025-11-24
> >
> > Thank you for the detailed responses. However, I still find it unusual that pass@1 is measured using only a single sample. In the reasoning literature, this is not standard practice for ensuring accurate measurement. As shown in recent works [1,2], reasoning models exhibit high variance across samples, and pass@1 is therefore typically estimated using multiple stochastic generations (e.g., 4–64 samples, depending on the test set size) rather than a single inference.
> >
> > I strongly recommend that the authors re-evaluate pass@1 metrics using multi-sample estimation, following the methodology adopted in the above literature [1,2], to ensure a reliable assessment of the proposed method's effectiveness.
> >
> > References
> > [1] DeepSeek-R1: Incentivizing Reasoning Capability in LLMs via Reinforcement Learning, Nature, 2025.
> > [2] OpenAI o1 System Card, arXiv, 2024.

---

> > > ### Author Response · Authors · 2025-11-26
> > > **Further Responses**
> > >
> > > Thank you for your constructive comments. Following your suggestion, we have re-evaluated the performance with the Avg@k metric.
> > >
> > > Specifically, we focus on a large-sample dataset (MATH-500) and the few-sample datasets mentioned in Question 4 (AIME24, AIME25, and AMC23). To balance the need for reduced randomness and computational feasibility, we evaluate performance using Avg@1, Avg@4, and Avg@16, where Avg@k represents the average accuracy of k independent Pass@1 scores computed following [1]. Additionally, we measure Tok@k, the average token count of k independent output sequences. These allow us to assess the relative performance gains of ReBalance over the baseline under multi-sample estimation.
> > >
> > > First, for MATH-500, randomness-induced variance is supposed to be smaller due to the sufficient number of samples, as shown by Std (Avg@k) and Std (Tok@k) in the table below. Thus, we adopt Avg@4 to re-evaluate Pass@1. For readability, Std (Avg@k) is scaled by 1000.
> > >
> > > *Evaluation on MATH-500*
> > >
> > > |Models|Avg@1|Tok@1|Avg@4|Tok@4|Std (Avg@4)|Std (Tok@4)|
> > > |---|---|---|---|---|---|---|
> > > |DeepSeek-R1-Distill-Qwen-1.5B|79.6|4516|79.6|4620|3.7|100.9|
> > > |w/ ReBalance|**83.0**|**3474**|**83.3**|**3553**|6.7|55.5|
> > > |DeepSeek-R1-Distill-Qwen-7B|89.8|3699|89.4|3703|7.4|56.4|
> > > |w/ ReBalance|**92.6**|**2903**|**92.6**|**2894**|1.4|38.1|
> > > |Qwen3-14B|93.8|4470|93.8|4550|1.4|38.8|
> > > |w/ ReBalance|**94.0**|**3641**|**94.1**|**3674**|1.0|44.6|
> > >
> > > Second, for AIME24, AIME25, and AMC23 with fewer samples, we compare the results and standard deviations of Avg@1, Avg@4, and Avg@16, where the choice of Avg@16 follows the experimental setup in Figure 2 of [1]. The results are reported below.
> > >
> > > *Evaluation on AIME24*
> > >
> > > |Models|Avg@1|#Tok@1|Avg@4|#Tok@4|Avg@16|#Tok@16|Std (Avg@4)|Std (Tok@4)|Std (Avg@16)|Std (Tok@16)|
> > > |---|---|---|---|---|---|---|---|---|---|---|
> > > |DeepSeek-R1-Distill-Qwen-1.5B|23.3|12596|21.7|12897|19.6|12931|16.7|830.68|33.1|830.7|
> > > |w/ ReBalance|**36.7**|**9040**|**34.2**|**9179**|**35.6**|**9179**|43.3|718.40|49.8|602.6|
> > > |DeepSeek-R1-Distill-Qwen-7B|40.0|13994|41.7|13636|41.3|13764|28.9|434.19|42.3|418.2|
> > > |w/ ReBalance|**56.7**|**9012**|**55.0**|**8664**|**57.9**|**9167**|16.7|896.54|51.1|620.9|
> > > |Qwen3-14B|66.7|10888|70.0|11488|67.1|11174|23.6|188.05|61.1|307.9|
> > > |w/ ReBalance|**73.3**|**9464**|**71.7**|**9627**|**73.1**|**9613**|16.5|115.31|18.5|112.1|
> > >
> > > *Evaluation on AIME25*
> > >
> > > |Models|Avg@1|#Tok@1|Avg@4|#Tok@4|Avg@16|#Tok@16|Std (Avg@4)|Std (Tok@4)|Std (Avg@16)|Std (Tok@16)|
> > > |---|---|---|---|---|---|---|---|---|---|---|
> > > |DeepSeek-R1-Distill-Qwen-1.5B|16.7|14556|15.0|14107|15.4|14589|16.7|354.3|16.1|416.3|
> > > |w/ ReBalance|**30.0**|**8140**|**27.5**|**8447**|**27.5**|**8723**|27.6|447.0|25.0|626.9|
> > > |DeepSeek-R1-Distill-Qwen-7B|26.7|13778|28.3|12340|27.3|12192|37.3|308.3|41.2|308.3|
> > > |w/ ReBalance|**40.0**|**9227**|**40.0**|**8813**|**40.0**|**9241**|23.6|609.3|20.4|463.2|
> > > |Qwen3-14B|56.7|13125|55.0|12900|54.4|12457|50.0|241.7|62.0|298.7|
> > > |w/ ReBalance|56.7|**11057**|**57.5**|**11023**|**57.3**|**11013**|59.5|213.1|68.5|224.3|
> > >
> > > *Evaluation on AMC23*
> > >
> > > |Models|Avg@1|#Tok@1|Avg@4|#Tok@4|Avg@16|#Tok@16|Std (Avg@4)|Std (Tok@4)|Std (Avg@16)|Std (Tok@16)|
> > > |---|---|---|---|---|---|---|---|---|---|---|
> > > |DeepSeek-R1-Distill-Qwen-1.5B|55.0|8990|53.8|8616|54.1|8637|12.5|302.6|19.5|538.8|
> > > |w/ ReBalance|**80.0**|**5216**|**80.6**|**4730**|**80.0**|**4729**|44.6|255.3|42.4|433.5|
> > > |DeepSeek-R1-Distill-Qwen-7B|75.0|6898|74.9|6297|74.8|6297|18.6|520.2|19.7|335.1|
> > > |w/ ReBalance|**95.0**|**4767**|**93.8**|**4115**|**92.2**|**4226**|21.7|423.4|39.4|394.4|
> > > |Qwen3-14B|95.0|7240|91.3|7244|93.1|6985|28.0|115.3|27.2|210.3|
> > > |w/ ReBalance|**100.0**|**5230**|**98.8**|**5120**|**97.7**|**4848**|21.7|98.4|18.7|190.2|
> > >
> > > From the experimental results in the tables above, we have the following observations:
> > >
> > > - **Inter-dataset variability in randomness**: As reflected by Std (Avg@k) and Std (Tok@k), the datasets with few samples exhibit higher standard deviations (up to 10× greater than those of large-sample datasets, e.g., MATH-500) in both accuracy and generated sequence length.
> > > - **Performance stability on large-sample datasets**: Multi-sample estimation has minimal impact on the relative performance of ReBalance v.s. the baseline, which remains largely consistent across sampling counts.
> > > - **Performance stability on few-shot datasets**: Although accuracy and sequence length vary with sampling count, the relative improvement of ReBalance over the baseline remains stable. Notably, under 16-sample evaluation, ReBalance demonstrates even greater performance gains compared to the original single-sample configuration across multiple datasets and models, further validating its strong effectiveness and robustness.
> > >
> > > Thank you for your constructive feedback. These results will be added to the final version.
> > >
> > > [1] DeepSeek-R1: Incentivizing Reasoning Capability in LLMs via Reinforcement Learning, Nature, 2025.

---

> ### Comment · Reviewer_4rhm · 2025-11-27
>
> Thank you for the additional experiments and clarifications.
> Most of my concerns have been fully addressed. I am going to increase both my presentation score and my confidence score.

---

> > ### Author Response · Authors · 2025-11-28
> >
> > Thank you sincerely for your positive feedback and insightful suggestions. We truly appreciate your time and effort in reviewing our paper.

---

### Official Review · Reviewer_HMxB · 2025-11-01

**Soundness:** 4
**Presentation:** 4
**Contribution:** 3
**Rating:** 10
**Confidence:** 5

**Summary:**

This paper addresses the issues of "Overthinking" (redundant computational steps) and "Underthinking" (insufficient reasoning paths) in Large Reasoning Models. The authors propose REBALANCE, a training-free framework that aims to achieve efficient and "balanced thinking." The core idea is to use model confidence and confidence variance as real-time indicators to dynamically steer the model's hidden states. The framework uses a pre-computed "steering vector," derived from reasoning mode prototypes extracted in an offline pass, to guide the model's reasoning trajectory. A dynamic control function modulates this vector's strength and direction, pruning redundancy during overthinking and promoting exploration during underthinking. Experimental results show that the proposed method outperforms existing baselines and achieve better efficiency.

**Strengths:**

1. Clear Optimization Objective: The paper defines Overthinking and Underthinking based on numerical features of internal states (stepwise confidence and confidence variance) derived from empirical observations. This semantic-agnostic definition, independent of specific keywords, provides a very clear and quantifiable optimization objective.

2. Extensive Experimentation: The paper provides comprehensive experiments and ablation studies. The REBALANCE framework is validated across multiple models of varying scales and tested on diverse domains beyond mathematics, demonstrating strong generalization. Furthermore, thorough ablation studies are conducted for key design choices, offering justifications for the method's components.

3. Demonstrated Effectiveness: The experimental results demonstrate that the proposed approach can successfully balance the trade-off between Overthinking and Underthinking. It outperforms previous baselines in terms of task performance and token efficiency.

**Weaknesses:**

1. Missing Citation to Key Related Work: The paper focuses significantly on the problem of "Underthinking". However, it fails to cite an important and highly relevant recent study on this specific phenomenon [1]. This omission is a significant gap in the related work section.
Reference:
[1] Wang Y, Liu Q, Xu J, et al. Thoughts are all over the place: On the underthinking of o1-like llms[J]. arXiv preprint arXiv:2501.18585, 2025.

2. Practical Application Burden: Although the method is presented as "training-free," it remains highly dependent on hyperparameter tuning. Furthermore, it requires an additional "seen dataset" and a model-specific calibration step (to extract steering vectors and fit the control function). This introduces an extra implementation and tuning burden for each new backbone model, which may complicate its practical application.

**Questions:**

1. The intervention method is specifically developed to target the paper's definitions of Overthinking (low-confidence/high-variance) and Underthinking (high-confidence/low-variance). However, could this direct manipulation of internal reasoning states have other unintended side effects on reasoning, beyond just mitigating these two defined issues? For instance, could it negatively impact creativity or the naturalness of the expressions?

2. While the quantitative results are strong, have the authors analyzed the semantic changes in the model's reasoning process post-intervention? For example, did the REBALANCE framework lead to a measurable change in the frequency of keywords associated with reflection or hesitation (e.g., "wait," "alternatively," "let me check")?

---

> ### Author Response · Authors · 2025-11-22
> **Response to Weakness 1 (Citation to Key Related Work)**
>
> >**Weakness 1**: Missing Citation to Key Related Work: The paper focuses significantly on the problem of "Underthinking". However, it fails to cite an important and highly relevant recent study on this specific phenomenon [1]. This omission is a significant gap in the related work section. Reference: [1] Wang Y, Liu Q, Xu J, et al. Thoughts are all over the place: On the underthinking of o1-like llms[J]. arXiv preprint arXiv:2501.18585, 2025.
>
> Thank you for your valuable suggestion. We have added citations to [1] where the concept of underthinking is introduced in the Introduction and discussed in Related Work (Appendix H).
>
> [1] Thoughts are all over the place: On the underthinking of o1-like llms

---

> ### Author Response · Authors · 2025-11-22
> **Response to Weakness 2 (1/2) (Practical Application Burden)**
>
> >**Weakness 2**: Practical Application Burden: Although the method is presented as "training-free," it remains highly dependent on hyperparameter tuning. Furthermore, it requires an additional "seen dataset" and a model-specific calibration step (to extract steering vectors and fit the control function). This introduces an extra implementation and tuning burden for each new backbone model, which may complicate its practical application.
>
> Thank you for raising this issue. Below, we respond to each of your concerns in detail.
>
> **(1) Dependent on Hyperparameter Tuning**
>
> In this study, we observe that methods with higher degrees of freedom typically require more parameters, which imposes practical burdens and compromises generalization, thereby limiting their flexibility. This trade-off is especially pronounced in training-free approaches.
>
> Our work aims to adaptively modulate model behavior during inference based on sample-specific characteristics and real-time reasoning dynamics. Specifically, we fit a three-dimensional surface using stepwise confidence and its variance as inputs. While greater surface flexibility demands more parameters, our analysis in Appendix A.4 reveals significant heterogeneity in confidence distributions across models, rendering fixed hyperparameter settings suboptimal for generalization.
>
> To overcome this challenge, we introduce a *model behavior-based dynamic control function fitting* method in Section 3.4, and details are in Appendix B.3. Crucially, our approach **only requires tuning a single hyperparameter**, $B_u$; **all other parameters are derived directly from the model’s behavior during steering vector extraction**, eliminating the need for extra inference steps or manual tuning. The full set of parameters in the dynamic function is summarized and compared in the table below.
>
> | Dynamic Control Function Parameters | Underthinking Boundary $B_u$ | Overthinking Boundary $B_o$ | Moderate Boundary $B_m$ | Lower Confidence Threshold $\tau_c^{\mathrm{L}}$ |  Higher Confidence Threshold $\tau_c^{\mathrm{H}}$ | Lower Variance Threshold $\tau_v^{\mathrm{L}}$ |  Higher Variance Threshold $\tau_v^{\mathrm{H}}$ |
> | --- | --- | --- | --- | --- | --- | --- | --- |
> | Is Hyperparameter | Yes | No | No | No | No | No | No |
> | Setting Method | **Fixed identical values** across all models and benchmarks | **Automatically** computed based on aggressive hidden state displacement from overthinking to underthinking space | **Automatically** computed based on moderate hidden state displacement from overthinking/underthinking to underthinking/overthinking space | **Automatically** elected based on quartiles of model confidence distribution | **Automatically** elected based on quartiles of model confidence distribution | **Automatically** elected based on quartiles of model confidence distribution | **Automatically** elected based on quartiles of model confidence distribution |
>
> **(2) The Control Function Fitting**
>
> As discussed above, one of our core contributions is a model behavior–based dynamic function fitting method that enhances control flexibility without sacrificing generalizability. Crucially, this fitting procedure can be seamlessly integrated into steering vector extraction, leveraging hidden states and confidence scores already computed during that step. Consequently, it incurs *no additional inference overhead*, requires *no extra manual hyperparameter tuning*, and imposes *no extra practical burden*.
>
> **(3) Steering Vector Extraction on an Additional Seen Dataset**
>
> In our method, when the overthinking or underthinking tendencies are identified, an intervention mechanism is needed to modulate the model’s behavior accordingly. We adopt steering as a representative approach because it offers an intuitive way to directly adjust hidden states, enabling smooth transitions between the overthinking and underthinking boundaries. However, we would like to clarify that the core idea of *balanced thinking* is method-agnostic.
>
> Notably, steering inherently requires two components, i.e., steering vector extraction and a seen dataset, but these are prerequisites of steering-based methods, not assumptions introduced by our work. To demonstrate the generality of our framework, inspired by [2-3], we also explore an alternative implementation: **Balanced Thinking with Dynamic Temperatures**.
>
> [2] EDT: Improving Large Language Models' Generation by Entropy-based Dynamic Temperature Sampling
>
> [3] Hot or cold? adaptive temperature sampling for code generation with large language models

---

> ### Author Response · Authors · 2025-11-22
> **Response to Weakness 2 (2/2) (Practical Application Burden)**
>
> Specifically, we replace dynamic steering with dynamic temperature scaling. In this context, if *overthinking* is observed during inference, we lower the temperature to suppress excessive exploration; if *underthinking* is detected, we raise it to encourage broader reasoning.  From the table presented below, despite the simplicity, we can observe that the Dynamic Temperature yields significant performance gains and reduces reasoning length compared to the baseline, achieving balanced reasoning *without relying on steering vectors or access to a seen dataset*, thereby validating the effectiveness of the proposed balanced thinking strategy.
>
> *DeepSeek-R1-Distill-Qwen-1.5B*
>
> | Methods | AMC23 (Pass@1) | AMC23 (#Tokens) | AIME24 (Pass@1) | AIME24 (#Tokens) | AIME25 (Pass@1) | AIME25 (#Tokens) | Olympiad (Pass@1) | Olympiad (#Tokens) |
> | --- | --- | --- | --- | --- | --- | --- | --- | --- |
> | Baseline | 55 | 8990 | 23.3 | 12596 | 16.7 | 14556 | 41.2 | 8785 |
> | Dynamic Temperature | 75 | 7344 | 36.7 | 12054 | 23.3 | 11659 | **44.7** | 8538 |
> | Dynamic Steering | **80** | **5216** | **36.7** | **9040** | **30** | **8140** | 43.9 | **7253** |
>
> *DeepSeek-R1-Distill-Qwen-7B*
>
> | Methods | AMC23 (Pass@1) | AMC23 (#Tokens) | AIME24 (Pass@1) | AIME24 (#Tokens) | AIME25 (Pass@1) | AIME25 (#Tokens) | Olympiad (Pass@1) | Olympiad (#Tokens) |
> | --- | --- | --- | --- | --- | --- | --- | --- | --- |
> | Baseline | 75 | 6898 | 40 | 13994 | 26.7 | 13778 | 56.1 | 7590 |
> | Dynamic Temperature | 82.5 | 5856 | 53.3 | 10593 | 36.7 | 10740 | **57.5** | 7498 |
> | Dynamic Steering | **95** | **4767** | **56.7** | **9012** | **40** | **9227** | 57 | **6321** |
>
> *QwQ-32B*
>
> | Methods | AMC23 (Pass@1) | AMC23 (#Tokens) | AIME24 (Pass@1) | AIME24 (#Tokens) | AIME25 (Pass@1) | AIME25 (#Tokens) | Olympiad (Pass@1) | Olympiad (#Tokens) |
> | --- | --- | --- | --- | --- | --- | --- | --- | --- |
> | Baseline | 87.5 | 7021 | 66.7 | 14342 | 46.7 | 13350 | 66.7 | 8219 |
> | Dynamic Temperature | 92.5 | 6721 | 66.7 | 11202 | 53.3 | 12134 | 67.6 | 8160 |
> | Dynamic Steering | **95** | **6064** | **70** | **10350** | **63.3** | **11575** | **68.6** | **7422** |
>
> The above experimental results and analysis have been included in Appendix C.9.

---

> ### Author Response · Authors · 2025-11-22
> **Response to Question 1 (Side Effects on Reasoning)**
>
> >**Question 1**: The intervention method is specifically developed to target the paper's definitions of Overthinking (low-confidence/high-variance) and Underthinking (high-confidence/low-variance). However, could this direct manipulation of internal reasoning states have other unintended side effects on reasoning, beyond just mitigating these two defined issues? For instance, could it negatively impact creativity or the naturalness of the expressions?
>
> Following your suggestion and in line with [4], we select *Creative Writing v3* for testing, as its rich and multidimensional metrics can provide ReBalance with a comprehensive and convincing evaluation. The results are shown in the table below.
>
> Specifically, we use Claude-Sonnet-4.5 as the judge model to evaluate three core metrics across four models:
>
> - **Rubric Score,** an aggregate quality score across multiple writing dimensions;
> - **Elo Score**, a relative writing-quality ranking calibrated with GPT-3.5-Turbo and DeepSeek-R1-Distill-Qwen-1.5B as fixed anchors;
> - **Fine-grained Ability Scores**, covering 15 creative-writing competencies: Coh (Coherent), Crt (Creativity), Img (Descriptive Imagery), Pac (Pacing), Ele (Elegant Prose), Inst (Instruction Following), Voi (Consistent Voice & Tone), Dia (Strong Dialogue), Flo (Sentence Flow), SDT (Show-Don’t-Tell), Ama (Avoids Amateurish Prose), Emo (Emotional Depth), Pos (Avoids Positivity Bias), Pur (Avoids Purple Prose), Ch (Believable Characters).
>
> | Models | Rubric | Elo | Coh | Crt | Img | Pac | Ele | Inst | Voi | Dia | Flo | SDT | Ama | Emo | Pos | Pur | Ch |
> | --- | --- | --- | --- | --- | --- | --- | --- | --- | --- | --- | --- | --- | --- | --- | --- | --- | --- |
> | DeepSeek-R1-Distill-Qwen-1.5B | 15.0 | 30.0 | 1.3 | 1.8 | **2.8** | 1.9 | 5.4 | **1.5** | **2.9** | 2.9 | 4.1 | 2.5 | 3.3 | 1.2 | **11.5** | 9.2 | **1.8** |
> | w/ ReBalance | **15.2** | **57.5** | **1.5** | **1.9** | 2.7 | **2.2** | **5.6** | 1.4 | 2.8 | **3.0** | **4.3** | 2.5 | 3.3 | 1.2 | 11.2 | **9.4** | 1.7 |
> | DeepSeek-R1-Distill-Qwen-7B | 22.9 | 308.9 | 4.0 | 4.4 | **4.8** | 4.8 | 6.6 | **3.1** | 5.0 | 4.2 | 6.6 | 4.0 | 5.4 | 2.7 | 9.9 | 9.6 | 3.4 |
> | w/ ReBalance | **23.4** | **349.7** | **4.3** | **4.5** | 4.7 | **5.1** | **6.7** | 3.0 | 5.0 | **4.6** | 6.6 | **4.1** | 5.4 | **2.9** | 9.9 | **9.9** | **3.5** |
> | Qwen3-14B | 42.6 | 1295.2 | 8.8 | 7.1 | 10.3 | 7.7 | 8.2 | 9.2 | 11.2 | 6.4 | 8.1 | 5.5 | 6.6 | 8.1 | 10.8 | 8.4 | 8.8 |
> | w/ ReBalance | **50.5** | **1368.0** | **12.2** | **7.2** | **11.6** | **11.3** | **9.5** | **12.1** | **12.3** | **8.2** | **9.9** | **6.9** | **8.0** | **9.4** | **12.6** | **9.0** | **10.3** |
> | QwQ-32B | 52.7 | 1438.2 | 12.6 | **8.1** | 12.0 | 11.7 | 9.8 | **13.6** | **12.6** | 8.5 | 9.8 | **8.1** | **8.3** | 9.8 | 11.4 | **9.6** | 10.6 |
> | w/ ReBalance | **52.8** | **1442.7** | **12.7** | 7.9 | **12.3** | **11.8** | **9.9** | 12.6 | 12.2 | **8.6** | 9.8 | 7.9 | 8.0 | 9.8 | **12.2** | 9.5 | **10.7** |
>
> It can be observed that **after applying ReBalance, the models generally maintain and even improve their performance in creativity and the naturalness of expressions**. The percentage of metrics that are higher than or equal to those of the original models is notably high: 71% for DeepSeek-R1-Distill-Qwen-1.5B, 88% for DeepSeek-R1-Distill-Qwen-7B, 100% for Qwen3-14B, and 65% for QwQ-32B. Notably, **Qwen3-14B achieves significant improvements across all metrics after applying ReBalance.**
> We posit that this improvement stems from ReBalance’s ability to continuously and gently guide the reasoning process, achieving a balance between overthinking and underthinking, thereby maintaining the model within effective reasoning boundaries. Consequently, the models exhibit measurable improvements in creative-writing performance, indicating that ReBalance accelerates convergence without compromising the model’s capacity for innovative or divergent thinking. The above experimental results and analysis have been included in Appendix C.9.
>
> [4] Qwen3 technical report

---

> ### Author Response · Authors · 2025-11-22
> **Response to Question 2 (1/2) (Semantic Changes after Intervention)**
>
> >**Question 2**: While the quantitative results are strong, have the authors analyzed the semantic changes in the model's reasoning process post-intervention? For example, did the REBALANCE framework lead to a measurable change in the frequency of keywords associated with reflection or hesitation (e.g., "wait," "alternatively," "let me check")?
>
> Thank you for your suggestion. Quantifying the frequency changes of reflection and hesitation keywords indeed offers an effective way to enhance the interpretability of our method.
>
> Building upon a similar idea, we have explored the relationship between keyword lists and model confidence patterns in Appendix A.6. Our findings indicate that reflection and hesitation keywords essentially represent the model's low-confidence behavior, although they are not as comprehensive as directly using confidence as indicators. This observation naturally motivates our hypothesis: ReBalance, by dynamically reshaping the model’s confidence distribution in response to real-time tendencies toward overthinking or underthinking, should effectively suppress low-confidence behavior and consequently reduce the frequency of such keywords in the model’s outputs.
>
> **(1) Experimental Setup for Semantic Analysis**
>
> To validate this hypothesis, we conduct a semantic analysis of reasoning traces on MATH-500, utilizing the Reflection and Transition keyword vocabularies introduced in [5].
>
> Specifically, we report the metrics averaged over all responses, comparing baseline and post-intervention results. These metrics include word count, term frequency (TF), and term frequency-inverse document frequency (TF-IDF) scores for both vocabularies, as well as changes in accuracy and token count relative to the baseline. For readability, TF and TF-IDF scores are scaled by a factor of 1,000.
>
> *DeepSeek-R1-Distill-Qwen-1.5B*
>
> | Methods | Reflection (Word Count) | Reflection (TF) | Reflection (TF-IDF) | Transition (Word Count) | Transition  (TF) | Transition  (TF-IDF) | Relative Pass@1 | Relative #Tokens |
> | --- | --- | --- | --- | --- | --- | --- | --- | --- |
> | Baseline | 30.3 | 9.7 | 12.1 | 6.7 | 2.4 | 3.9 | - | - |
> | NoThinking | 3.9 | 1.7 | 4.6 | 0.9 | 0.4 | 1.3 | -4.6 | -64.9% |
> | NoWait | 1.6 | 1.7 | 3.5 | 0.1 | 1.9$ \times 10^{-2}$ | 7.7$ \times 10^{-2}$ | -1.6 | -41.4% |
> | ReBalance (Ours) | 10.5 | 6.4 | 9.3 | 1.7 | 1.0 | 2.3 | **+3.4** | -23.1% |
>
> *DeepSeek-R1-Distill-Qwen-7B*
>
> | Methods | Reflection (Word Count) | Reflection (TF) | Reflection (TF-IDF) | Transition (Word Count) | Transition  (TF) | Transition  (TF-IDF) | Relative Pass@1 | Relative #Tokens |
> | --- | --- | --- | --- | --- | --- | --- | --- | --- |
> | Baseline | 19.2 | 8.1 | 10.3 | 5.2 | 2.3 | 3.7 | - | - |
> | NoThinking | 1.8 | 1.2 | 3.1 | 0.1 | 9.6$ \times 10^{-2}$ | 4.5$ \times 10^{-2}$ | -9.2 | -77.4% |
> | NoWait | 1.2 | 1.8 | 3.9 | 0.0 | 2.8$ \times 10^{-2}$ | 1.8$ \times 10^{-2}$ | -3.0 | -32.9% |
> | ReBalance (Ours) | 12.8 | 6.7 | 9.2 | 2.7 | 1.4 | 2.8 | **+2.8** | -21.5% |
>
> *Qwen3-14B*
>
> | Methods | Reflection (Word Count) | Reflection (TF) | Reflection (TF-IDF) | Transition (Word Count) | Transition  (TF) | Transition  (TF-IDF) | Relative Pass@1 | Relative #Tokens |
> | --- | --- | --- | --- | --- | --- | --- | --- | --- |
> | Baseline | 24.0 | 9.2 | 10.8 | 8.7 | 3.3 | 4.4 | - | - |
> | NoThinking | 10.4 | 5.5 | 9.4 | 2.8 | 1.0 | 2.4 | 0.0 | -40.5% |
> | NoWait | 2.1 | 1.8 | 3.3 | 0.0 | 9.4$ \times 10^{-2}$ | 4.1$ \times 10^{-2}$ | -1.0 | -27.9% |
> | ReBalance (Ours) | 13.2 | 6.8 | 8.1 | 4.6 | 2.3 | 3.3 | **+0.2** | -18.5% |
>
> *QwQ-32B*
>
> | Methods | Reflection (Word Count) | Reflection (TF) | Reflection (TF-IDF) | Transition (Word Count) | Transition  (TF) | Transition  (TF-IDF) | Relative Pass@1 | Relative #Tokens |
> | --- | --- | --- | --- | --- | --- | --- | --- | --- |
> | Baseline | 26.7 | 11.2 | 13.3 | 10.1 | 3.8 | 5.1 | - | - |
> | NoThinking | 26.0 | 11.5 | 13.7 | 9.5 | 3.8 | 5.1 | 0.0 | -13.7% |
> | NoWait | 2.5 | 2.4 | 4.1 | 0.1 | 2.2$ \times 10^{-2}$ | 8.1$ \times 10^{-2}$ | -1.0 | -27.9% |
> | ReBalance (Ours) | 22.1 | 10.8 | 10.1 | 5.4 | 2.5 | 3.8 | **+0.4**  | -19.3% |
>
> **(2) Limitations of Uniform Keyword Suppression**
>
> As shown in the table above, NoThinking and NoWait suppress all keywords indicative of reflection or transition uniformly and indiscriminately. Although this leads to a clear reduction in output length, it also causes a sharp decline in the frequency of such keywords, thereby inhibiting effective reasoning and resulting in significant performance degradation.
>
> [5] Seal: Steerable reasoning calibration of large language models for free.

---

> ### Author Response · Authors · 2025-11-22
> **Response to Question 2 (2/2) (Semantic Changes after Intervention)**
>
> **(3) ReBalance Achieves Selective Reduction with Preserved Reasoning Quality**
>
> In contrast, Word Count, TF, and TF-IDF scores show that our method does not explicitly suppress reflection or transition keywords at the semantic level, yet still achieves a notable reduction in their generation frequency. This further validates our analysis in Appendix A.6: *current keyword-based approaches are essentially variants of confidence-based methods like ReBalance, but they fall short of capturing the model’s reasoning patterns comprehensively*. This observation can help explain the superior performance of ReBalance.
>
> Moreover, the opposing trends of declining TF and rising relative accuracy indicate that *while ReBalance reduces keyword frequency, it also preserves a meaningful portion of these patterns*. This aligns with our proposed concept of balanced thinking, i.e., simultaneously pruning redundant reasoning while maintaining or even encouraging productive exploration. Together, these findings underscore the importance of retaining essential reflection and transition patterns for sustained model performance, positioning the proposed *balanced thinking* as a promising direction for future research. The experimental results and analysis above have been included in Appendix C.6.

---

> > ### Comment · Reviewer_HMxB · 2025-11-24
> > **Thank you for your work on rebuttle**
> >
> > Thank you for your detailed reply and for your work on rebuttle. This has resolved my doubts, and I have no further questions.

---

> > > ### Author Response · Authors · 2025-11-24
> > >
> > > We sincerely appreciate your positive feedback and constructive review. Thank you again for your time and thoughtful consideration.

---

### Official Review · Reviewer_RNpb · 2025-11-02

**Soundness:** 3
**Presentation:** 2
**Contribution:** 3
**Rating:** 4
**Confidence:** 3

**Summary:**

This paper addresses a key challenge in LRMs: the trade-off between overthinking and underthinking. The authors propose ReBalance, a training-free and plug-and-play framework that dynamically controls reasoning behavior via internal confidence signals. Experiments on 4 models and 9 reasoning benchmarks show that ReBalance reduces reasoning token length significantly while improving or maintaining accuracy.

**Strengths:**

1. An important problem that is of strong interest to the LRM community.

2. The paper uses stepwise confidence and confidence variance to detect reasoning modes, with a motivation that is well-explained and intuitively convincing.

3. The method achieves superior results across a wide range of reasoning benchmarks.

**Weaknesses:**

1. The main text lacks a concise introduction to the baselines, which makes the paper more difficult for readers to follow.

2. Novelty of this paper is limited. Steering vector has been used before. How does ReBalance differ from SEAL? Both of them leverage the steering vector. I want to see an apple-to-apple comparison.

3. ReBalance is similar in spirit to adaptive halting or early-exit reasoning models, such as TrimR [1] and FlashThink [2]. I noticed that the authors cited these two papers, but neither compared them as baselines nor discussed them in the Related Work section of the appendix or other places.

[1] Weizhe Lin et al. TrimR: Verifier-based training-free thinking compression for efficient test-time scaling. arXiv preprint arXiv:2505.17155, 2025.

[2] Guochao Jiang et al. FlashThink: An early exit method for efficient reasoning. arXiv preprint arXiv:2505.13949, 2025. (edited)

**Questions:**

see the weakness above

---

> ### Author Response · Authors · 2025-11-22
> **Response to Weakness 1 (Introduction to Baselines)**
>
> >**Weakness 1**: The main text lacks a concise introduction to the baselines, which makes the paper more difficult for readers to follow.
>
> Thank you for your suggestion. We agree that presenting baseline methods in the main text allows readers to quickly understand the compared approaches. Previously, due to space limitations, baseline descriptions were placed in Appendix D. **We have now added a clear explanation of this content in Section 4 of the main text.**

---

> ### Author Response · Authors · 2025-11-22
> **Response to Weakness 2 (1/2) (Distinction from Existing Steering-Based Methods)**
>
> >**Weakness 2**: Novelty of this paper is limited. Steering vector has been used before. How does ReBalance differ from SEAL? Both of them leverage the steering vector. I want to see an apple-to-apple comparison.
>
> We appreciate your feedback, as it offers us an opportunity to further clarify our motivation and underscore the key contributions of our work.
>
> Firstly, as requested, we present an **apple-to-apple comparison** between ReBalance and SEAL. We analyze both methods through a comprehensive assessment covering six dimensions: *steering layer selection*, *steering vector extraction*, *steering strength*, *steering direction*, *steering mode*, and *resulting effects*. Further details and an expanded discussion differentiating ReBalance from other steering-based methods for efficient reasoning have been included in Appendix H.
>
> | Methods | Steering Layer Selection | Steering Vector Extraction | Steering Strength | Steering Direction | Steering Mode | Resulting Effects |
> | --- | --- | --- | --- | --- | --- | --- |
> | SEAL | Requires multiple full experiments with manual tuning per model | Splits reasoning steps into execution and reflection/transition via empirically defined keywords | Manually sets fixed, uniform strength across models, samples, and reasoning steps | Manually selects a fixed direction, continuously suppressing reflection/transition across all steps | Uniformly employs steering weights, consistently suppressing reflection/transition in a single direction | Indiscriminately suppresses reflection/transition, inevitably causing underthinking |
> | ReBalance | **Automatically** selects optimal steering layers during vector extraction, eliminating extra inference and manual tuning | Uses **confidence** as a reliable continuous indicator to identify steps prone to overthinking or underthinking | **Dynamically** adjusts steering strength per reasoning step, proportional to the detected tendency toward over- or underthinking | **Dynamically** modulates steering direction per reasoning step, based on the detected tendency type: negative for overthinking, positive for underthinking | **Dynamically** tracks reasoning states through confidence indicators, adjusting strength and direction in real-time | Smoothly maintains model reasoning within the boundaries of over- and underthinking, achieving **balanced thinking** |
>
> SEAL is a valuable work and represents a good application of the steering concept in efficient reasoning, providing us with considerable inspiration. Nevertheless, this comprehensive analysis clearly demonstrates that ReBalance substantially differs from SEAL across each dimension. Furthermore, **ReBalance significantly outperforms SEAL in generalization across different models, adaptability to various samples and real-time inference dynamics, and achieving a balanced reasoning state**. These advantages have been extensively validated in Table 1 of our paper across 4 parameter scales, 3 model families, and 6 datasets with varying difficulties.
>
> However, despite ReBalance's notable advantages over existing steering-based methods, we emphasize that **the primary contribution of our paper is not proposing a new steering technique**. Rather, our goal is to share an insight with the community: *current methods for mitigating overthinking typically neglect the delicate balance between overthinking and underthinking*, inevitably introducing underthinking while alleviating overthinking. Thus, there is an urgent need for methods that achieve balanced thinking.
>
> Although overthinking and underthinking have become widely acknowledged in the efficient reasoning community as two problematic yet abstract reasoning states, existing studies typically remain at a *conceptual level* without *explicit modeling*. In contrast, we demonstrate through experiments that *model confidence, when utilized effectively, serves as a continuous and reliable indicator of overthinking and underthinking*. This approach allows accurate, real-time detection of when models exhibit either state during inference, enabling immediate and balanced adjustments. Considering that model behavior is represented by hidden states, steering-based approaches provide an intuitive and interpretable means to directly modify hidden states and facilitate transitions between the two extremes (overthinking and underthinking). Thus, we **use steering as an illustrative example to practically implement our idea of balancing overthinking and underthinking**. Importantly, *the concept itself is generalizable and applicable to a variety of approaches aiming at promoting balanced thinking*.

---

> ### Author Response · Authors · 2025-11-22
> **Response to Weakness 2 (2/2) (Distinction from Existing Steering-Based Methods)**
>
> **Balanced Thinking with Dynamic Temperature.** To further support our arguments above, inspired by [1-2], we replace the dynamic steering component in ReBalance with a much simpler approach: adjusting the temperature parameter. Specifically, when *overthinking* is detected during inference, we reduce the temperature to avoid excessively divergent exploration leading to redundant reasoning. Conversely, when *underthinking* is detected, we increase the temperature to broaden reasoning. Due to constraints in time and computational resources, we conduct only preliminary experiments using a discrete, binary hyperparameter setting without further tuning. Still using confidence as the indicator, we set the temperature to 1.2 upon detecting underthinking, and reduce it to 0.7 upon detecting overthinking.
>
> *DeepSeek-R1-Distill-Qwen-1.5B*
>
> | Methods | AMC23 (Pass@1) | AMC23 (#Tokens) | AIME24 (Pass@1) | AIME24 (#Tokens) | AIME25 (Pass@1) | AIME25 (#Tokens) | Olympiad (Pass@1) | Olympiad (#Tokens) |
> | --- | --- | --- | --- | --- | --- | --- | --- | --- |
> | Baseline | 55 | 8990 | 23.3 | 12596 | 16.7 | 14556 | 41.2 | 8785
> | Dynamic Temperature | 75 | 7344 | 36.7 | 12054 | 23.3 | 11659 | **44.7** | 8538 |
> | Dynamic Steering | **80** | **5216** | **36.7** | **9040** | **30** | **8140** | 43.9 | **7253** |
>
> *DeepSeek-R1-Distill-Qwen-7B*
>
> | Methods | AMC23 (Pass@1) | AMC23 (#Tokens) | AIME24 (Pass@1) | AIME24 (#Tokens) | AIME25 (Pass@1) | AIME25 (#Tokens) | Olympiad (Pass@1) | Olympiad (#Tokens) |
> | --- | --- | --- | --- | --- | --- | --- | --- | --- |
> | Baseline | 75 | 6898 | 40 | 13994 | 26.7 | 13778 | 56.1  | 7590 |
> | Dynamic Temperature | 82.5 | 5856 | 53.3 | 10593 | 36.7 | 10740 | **57.5** | 7498 |
> | Dynamic Steering | **95** | **4767** | **56.7** | **9012** | **40** | **9227** | 57 | **6321** |
>
> *QwQ-32B*
>
> | Methods | AMC23 (Pass@1) | AMC23 (#Tokens) | AIME24 (Pass@1) | AIME24 (#Tokens) | AIME25 (Pass@1) | AIME25 (#Tokens) | Olympiad (Pass@1) | Olympiad (#Tokens) |
> | --- | --- | --- | --- | --- | --- | --- | --- | --- |
> | Baseline | 87.5 | 7021 | 66.7 | 14342 | 46.7 | 13350 | 66.7 | 8219 |
> | Dynamic Temperature | 92.5 | 6721 | 66.7 | 11202 | 53.3 | 12134 | 67.6 | 8160 |
> | Dynamic Steering | **95** | **6064** | **70** | **10350** | **63.3** | **11575** | **68.6** | **7422** |
>
> Even with such a simple configuration, our balanced thinking approach achieves significant performance improvements and reductions in reasoning length. Therefore, we believe the performance can be further enhanced by introducing a model behavior-based dynamic function fitting method similar to ReBalance, which naturally enables adaptive continuous regulation without manual hyperparameter tuning. The above experimental results and analysis have been included in Appendix C.9.
>
> [1] EDT: Improving Large Language Models' Generation by Entropy-based Dynamic Temperature Sampling
>
> [2] Hot or cold? adaptive temperature sampling for code generation with large language models

---

> ### Author Response · Authors · 2025-11-22
> **Response to Weakness 3 (1/2) (Distinction from Existing Early-Exit Methods)**
>
> >**Weakness 3**: ReBalance is similar in spirit to adaptive halting or early-exit reasoning models, such as TrimR [1] and FlashThink [2]. I noticed that the authors cited these two papers, but neither compared them as baselines nor discussed them in the Related Work section of the appendix or other places.
>
> Thank you for your valuable comments. We agree that early-exit methods are representative strategies for mitigating overthinking. However, we would like to clarify the differences between ReBalance and these early-exit methods.
>
> **Differences between Early-Exit Methods and ReBalance.** For baseline comparisons, as detailed in Appendix D, we have selected two recent open-source early-exit methods (Dynasor-CoT and DEER). We did not directly compare TrimR and FlashThink as their implementations are *not publicly available*, making the comparison challenging. Additionally, TrimR is implemented on Ascend NPUs, which limits its reproducibility in standard experimental settings with NVIDIA GPUs.
>
> Besides, detailed discussions of these methods have been presented in several places in the main paper. First, in the key observation in Section 1, we have discussed that although current early-exit methods can reduce reasoning sequences using manually designed metrics, they typically retain or discard entire reasoning paths (sub-thoughts in TrimR, reasoning chunks in FlashThink). This rigid binary selection risks discarding potentially valuable reasoning steps, thereby inducing additional underthinking. Second, in Appendix A.6, as TrimR and FlashThink both rely on empirically set keywords to trigger early exits, we have discussed them as representative keyword-based methods and conducted exploratory experiments on the underlying mechanisms enabling keyword effectiveness. Also, as newly included in revised Appendix H, we have further discussed the distinctions between ReBalance and existing early-exit methods to facilitate understanding.
>
> Moreover, early-exit methods such as TrimR and FlashThink *exclusively focus on alleviating model overthinking*, overlooking the potential underthinking issues. Although TrimR includes certain mechanisms addressing underthinking, it primarily opts for abandoning reasoning paths upon detecting underthinking, which leans towards engineering-driven token length optimization rather than genuinely addressing underthinking itself. Differently, ReBalance specifically targets this issue by *concurrently mitigating overthinking and preventing the introduction of underthinking*, thereby achieving efficient reasoning with balanced thinking.
>
> **Reproduced Results of TrimR and FlashThink**.
>
> Although the official implementations of TrimR and FlashThink are not publicly available, we have carefully reproduced their results by strictly adhering to the experimental configurations described in their respective papers, enabling a fair quantitative comparison. The details are as follows.
>
> For **TrimR**, we first replicate the experiments using QwQ-32B under the exact settings reported in the original paper to validate the correctness of our implementation. In the original study, TrimR was executed on Ascend NPUs with the NPU-native Pangu-7B model serving as the verifier. Since we do not have access to Ascend NPUs, our experiments are instead conducted on NVIDIA RTX PRO 6000 GPUs and adopt Qwen2.5-7B-Instruct as the verifier, following the other configuration outlined in the paper. To this end, our reproduced results closely match the relative improvements over the baseline reported in the original work, and notably, we even observe greater relative gains on AIME24, AIME25, and GPQA-D.  This alignment provides evidence for the reliability of our reproduction.
>
> | Methods | MATH-500 | AIME24 | AIME25 | GPQA-D |
> | --- | --- | --- | --- | --- |
> | Baseline (Paper) | 95.6 | 76.7 | 60.0 | 66.0 |
> | TrimR (Paper) | 96.8 | 76.7 | 60.8 | 65.2 |
> | Baseline (Reproduced) | 94.8 | 66.7 | 46.7 | 63.1 |
> | TrimR (Reproduced) | 94.8 | 70.0 | 53.3 | 69.2 |

---

> ### Author Response · Authors · 2025-11-22
> **Response to Weakness 3 (2/2) (Distinction from Existing Early-Exit Methods)**
>
> Next, we evaluate TrimR under the same experimental setting as ReBalance. In TrimR, R denotes the underthinking threshold defined in the original paper. Specifically, the reasoning process is terminated once it reaches R% of the maximum sequence length, with R = 0.5 used by default. Since TrimR is originally validated only on models of 32B parameters or larger, we conduct our comparison using QwQ-32B to ensure a fair assessment of its strengths. Despite TrimR’s use of an additional 7B auxiliary model, an extra inference stage, and post-hoc optimization via repetition truncation, ReBalance still achieves significantly better performance. Moreover, our analysis of the R parameter reveals that increasing R leads to notable accuracy gains for TrimR on challenging benchmarks such as AIME24 and AIME25. This observation further supports the hypothesis that *existing approaches designed to mitigate overthinking can inadvertently induce underthinking*, highlighting a critical trade-off in current reasoning frameworks.
>
> | Methods | MATH-500 (Pass@1) | MATH-500 (#Tokens) | AIME24 (Pass@1) | AIME24 (#Tokens) | AIME25 (Pass@1) | AIME25 (#Tokens) | GSM8K (Pass@1) | GSM8K (#Tokens) | AMC23 (Pass@1) | AMC23 (#Tokens) |
> | --- | --- | --- | --- | --- | --- | --- | --- | --- | --- | --- |
> | Baseline | 94.8 | 4535 | 66.7 | 14342 | 46.7 | 13350 | 96.3 | 1506 | 87.5 | 7021 |
> | TrimR (R=0.5) | 93.8 | 3830 | 56.7 | 8345 | 43.3 | 8827 | 93.7 | 1319 | 90.0 | 6055 |
> | TrimR (R=0.75) | 94.8 | 4048 | 70.0 | 10235 | 53.3 | 11397 | 96.7 | 1410 | 87.5 | 6778 |
> | TrimR (R=1.0) | 93.8 | 4241 | 66.7 | 11007 | 56.7 | 11726 | 95.9 | 1432 | 90.0 | 6890 |
> | ReBalance | **95.2** | 3662 | **70.0** | 10350 | **63.3** | 11575 | **96.8** | 1289 | **95.0** | 6064 |
>
> Regarding **FlashThink**, we first conduct experiments on QwQ-32B using the exact experimental settings and original dataset from the paper, selecting Qwen2.5-7B-Instruct as the verifier to confirm our reproduction’s correctness. As shown below, our reproduced results substantially exceed the original results. We attribute this improvement potentially to the fact that the original delimiter token set is not open-sourced, prompting us to establish a more scientifically robust set.
>
> | Methods | GSM8K | GPQA-D |
> | --- | --- | --- |
> | Baseline (Paper) | 92.7 | 58.6 |
> | FlashThink (Paper) | 92.7 | 58.1 |
> | Baseline (Reproduced) | 96.3 | 63.1 |
> | FlashThink (Reproduced) | 96.5 | 63.6 |
>
> Next, we evaluate FlashThink under the same experimental setup as ReBalance. Although our reimplementation of FlashThink surpasses the original results reported in its paper, ReBalance still demonstrates clear superiority over FlashThink.
>
> | Methods | MATH-500 (Pass@1) | MATH-500 (#Tokens) | AIME24 (Pass@1) | AIME24 (#Tokens) | AIME25 (Pass@1) | AIME25 (#Tokens) | GSM8K (Pass@1) | GSM8K (#Tokens) | AMC23 (Pass@1) | AMC23 (#Tokens) |
> | --- | --- | --- | --- | --- | --- | --- | --- | --- | --- | --- |
> | Baseline | 94.8 | 4535 | 66.7 | 14342 | 46.7 | 13350 | 96.3 | 1506 | 87.5 | 7021 |
> | FlashThink | 93.2 | 3144 | 60.0 | 10034 | 40.0 | 11861 | 96.5 | 910 | 92.5 | 6702 |
> | ReBalance | **95.2** | 3662 | **70.0** | 10350 | **63.3** | 11575 | **96.8** | 1289 | **95.0** | 6064 |
>
> The above experimental results, along with the reproduction details for TrimR and FlashThink, have been included in Appendix C.8.

---

> ### Author Response · Authors · 2025-11-27
> **Looking Forward to Your Valuable Feedback**
>
> Dear Reviewer RNpb,
>
> I hope this message finds you well. We have carefully addressed the questions and concerns you previously raised, and we would greatly appreciate any further feedback you might be able to provide. Your insights are highly valuable to us, and we remain fully available to clarify or refine any remaining points.
>
> As the discussion deadline is approaching, if possible, we would be grateful for your response at your earliest convenience, as it would greatly help us move the discussion forward.
>
> Thank you sincerely for your time and effort in reviewing our paper.
>
> Best,
>
> The Authors

---

### Author Response · Authors · 2025-12-03
**Summary of Rebuttal**

Dear Area Chairs (ACs),

We sincerely appreciate your time and effort in reviewing our paper.

During the discussion period, we have carefully addressed each comment and believe most concerns are resolved. So far, we've received positive feedback from Reviewer HMxB and 4rhm indicating that our rebuttal has addressed their concerns.

| Reviewer | Initial Rating | Confidence | Post-rebuttal Feedback |
| --- | --- | --- | --- |
| RNpb | 4 | 3 | - |
| HMxB | 10 | 5 | **Concerns are addressed.** |
| 4rhm | 8 | 3 | **Concerns are addressed, committed to raising confidence and the presentation score.** |
| 4rRS | 6 | 5 | - |

Best regards,

The authors

---

> ### Author Response · Authors · 2025-12-03
> **Summary of Rebuttal (Part 1/5): Reviewer RNpb (Initial Rating: 4, Confidence: 3, Feedback: None)**
>
> Reviewer RNpb highlights that our work addresses an **important problem of strong interest** to the LRM community, is **well-motivated** with an **intuitively compelling rationale**, and achieves **superior results** across a **wide range** of reasoning benchmarks.
>
> > **Weakness 1:** Require baseline introduction in the main text.
> >
> - **Solution:** Add related statements in Section 4.
>
> > **Weakness 2:** Require apple-to-apple comparison with SEAL.
> >
> - **Solution:** (1) Compare point-by-point across six key dimensions; (2) Discuss with existing steering-based methods in detail (Appendix H); (3) Highlight ReBalance's core concept is balanced thinking, with steering serving only as an illustrative example, and quantitatively validate using temperature tuning instead (Appendix C.9).
> - **Conclusion:** ReBalance significantly outperforms SEAL in terms of model generalization, adaptability to real-time inference dynamics, and maintaining a balanced reasoning state. Our core concept of balanced thinking is method-agnostic.
>
> > **Weakness 3:** Require comparison and discussion with FlashThink and TrimR.
> >
> - **Solution:** (1) Clarify compared early-exit methods, justify the exclusion of FlashThink and TrimR, and reference the relevant discussions from the original paper; (2) Discuss with existing early-exit methods in detail (Appendix H); (3) Reproduce closed-source FlashThink and TrimR and compare with ReBalance (Appendix C.8).
> - **Conclusion:** ReBalance’s balanced thinking mechanism is fundamentally different from early-exit methods, which exclusively focus on overthinking. And ReBalance notably surpasses FlashThink and TrimR in both accuracy and efficiency.
>
> Overall, Reviewer RNpb’s main concerns focus on distinguishing ReBalance from existing methods. We provide apple-to-apple comparisons and reproduce the requested closed-source methods, underscoring ReBalance’s unique advantages and addressing the reviewer’s concerns.

---

> ### Author Response · Authors · 2025-12-03
> **Summary of Rebuttal (Part 2/5): Reviewer HMxB (Initial Rating: 10, Confidence: 5, Feedback: Positive)**
>
> Reviewer HMxB commends our work for its **clear optimization objective**, **extensive experimentation** with **thorough ablations**, and **demonstrated effectiveness** in balancing overthinking and underthinking.
>
> > **Weakness 1:** Require citation to related work.
> >
> - **Solution:** Add citations in Section 1 and Appendix H.
>
> > **Weakness 2:** Require clarification on the practical application burden introduced by hyperparameter tuning, control function fitting, seen datasets, and steering vector extraction.
> >
> - **Solution:** (1) List all function parameters, noting that only one requires tuning and is set to a fixed, identical value; (2) Emphasize that control function fitting entails no extra inference phase or manual intervention; (3) Clarify that seen datasets and steering vector extraction stem from the chosen steering method, not from ReBalance, whose balanced thinking concept is method-agnostic, as confirmed by dynamic temperature experiments (Appendix C.9).
> - **Conclusion:** ReBalance incurs no additional inference overhead, requires no extra manual hyperparameter tuning, and imposes no extra practical application burden.
>
> > **Question 1:** Require analysis of the side effects on creativity and naturalness.
> >
> - **Solution:** Evaluate ReBalance on 17 related metrics from Creative Writing v3 (Appendix C.6).
> - **Conclusion:** ReBalance not only prevents negative effects on creativity and naturalness but also enhances these qualities.
>
> > **Question 2:** Require interpretability analysis of the semantic change.
> >
> - **Solution:** Statistically compare the output frequencies of reflection/transition-related keywords before and after applying ReBalance (Appendix C.6).
> - **Conclusion:** ReBalance dynamically suppresses low-confidence behaviors, effectively reducing the output frequency of such keywords and thereby shortening reasoning length.
>
> Reviewer HMxB explicitly states that our rebuttal has resolved his/her concerns, and we have updated the paper accordingly.

---

> ### Author Response · Authors · 2025-12-03
> **Summary of Rebuttal (Part 3/5): Reviewer 4rhm (Initial Rating: 8, Confidence: 3, Feedback: Positive)**
>
> Reviewer 4rhm praises the paper for its **clear motivation** and focus on a **crucial reasoning problem**, its **well-written** and **easy to follow**, **strong experimental validation**, and its **simplicity** and **broad applicability**.
>
> > **Question 1:** Require comparison of steering vector extraction using different definitions of over-/underthinking
> >
> - **Solution:** Adopt Eq. (3) to model over-/underthinking and extract steering vectors accordingly for comparison (Appendix C.10).
> - **Conclusion:** Using confidence as the indicator is essential for achieving strong performance.
>
> > **Question 2:** Require validation of the observations from Figure 2(b) on a broader range of models.
> >
> - **Solution:** Repeat the experiment of Figure 2(b) on 4 models across 3 model families with parameter sizes ranging from 1.5B to 32B (Appendix C.7).
> - **Conclusion:** Confidence proves a broadly applicable indicator of over-/underthinking across various models.
>
> > **Question 3:** Require quantitative throughput evaluation.
> >
> - **Solution:** Compare ReBalance’s efficiency with baseline methods regarding TPS, TPR, GPU memory usage, and output token count (Appendix I).
> - **Conclusion:** ReBalance achieves a superior accuracy/efficiency trade-off compared to existing methods.
>
> > **Question 4:** Require evaluation on multi-sample scenarios
> >
> - **Solution:** Evaluate ReBalance under Pass@k and Avg@k metrics across multiple sampling counts from 4 to 20 (Appendix C.4).
> - **Conclusion:** ReBalance remains robust in multi-sample scenarios and demonstrates significant improvements in both accuracy and efficiency over the baseline.
>
> Reviewer 4rhm explicitly states that most of his concerns have been fully addressed, and commits to increasing confidence and the presentation score. We have also updated the paper accordingly.

---

> ### Author Response · Authors · 2025-12-03
> **Summary of Rebuttal (Part 4/5): Reviewer 4rRS (Initial Rating: 6, Confidence: 5, Feedback: None)**
>
> Reviewer 4rRS praises our work for its **conceptual appeal**, **ease of instantiation**, **robustness**, **broad applicability**, as well as the paper's **well-written**, **clearly structured**, **extensive empirical validation**, and **thoughtful analysis**.
>
> > **Weakness 1:** Require validation on cross-domain and cross-difficulty transferability.
> >
> - **Solution:** (1) Clarify that the original paper already includes the required cross-domain setup and perform broader cross-domain validation (Appendix C.3); (2) Conduct extensive transfer evaluations across datasets of varying difficulty (Appendix C.3).
> - **Conclusion:** ReBalance shows robust performance across mathematics, coding, commonsense, and science domains, as well as 5 difficulty levels, with varying priorities between accuracy and efficiency gains depending on problem difficulty.
>
> > **Weakness 2:** Require clarification on the generalization of hyperparameters.
> >
> - **Solution:** List and explain all ReBalance parameters, clarifying that most are automatically fitted using a unified logic applied once per model, while the remaining parameters are kept fixed across models, datasets, and domains.
> - **Conclusion:** ReBalance requires no manual hyperparameter tuning, and a single configuration works consistently across models, datasets, and domains.
>
> > **Weakness 3:** Require clarification on the method’s conceptual motivation, principled foundation, and generalization across diverse confidence distributions.
> >
> - **Solution:** (1) Clarify our conceptual motivation based on existing experiments with ablations on gate functions for further illustration (Appendix C.2); (2) Highlight sections in the original paper containing principled insights; (3) Validate generalization across 3 settings that induce confidence distribution shifts (Appendix C.5).
> - **Conclusion:** ReBalance is conceptually driven, grounded in extensive principled understandings, and exhibits strong generalization across different confidence distributions.
>
> > **Question 1:** Require throughput analysis and memory usage analysis
> >
> - **Solution:** Compare ReBalance’s efficiency with baseline methods regarding TPS, TPR, GPU memory usage, and output token count (Appendix I).
> - **Conclusion:** ReBalance achieves a superior accuracy/efficiency trade-off compared to existing methods.
>
> Overall, most concerns raised by Reviewer 4rRS were addressed in the original manuscript. We have provided further clarifications and conducted additional experiments where necessary, addressing the reviewer’s left concerns.

---

> ### Author Response · Authors · 2025-12-03
> **Summary of Rebuttal (Part 5/5): Revised Paper**
>
> During the rebuttal and discussion phases, we have carefully addressed each comment and believe most concerns are resolved. Below, we summarize the additional experiments during rebuttal, which have been incorporated into the revised paper and marked in blue.
>
> ---
>
> **Broader Applicability**
>
> 1. **Cross-Domain Transferability (`Appendix C.3`)**: ReBalance demonstrates robust performance across mathematics, coding, commonsense, and science domains.
> 2. **Cross-Difficulty Transferability (`Appendix C.3`)**: Vector extractions across 6 datasets and 5 difficulty levels show that easy problems prioritize token reduction, hard ones prioritize accuracy, and medium problems balance both.
> 3. **Multi-Sample Estimation (`Appendix C.4`)**: ReBalance consistently achieves strong results under Pass@k and Avg@k metrics across 4 sampling counts.
> 4. **Different Confidence Distributions (`Appendix C.5`)**: ReBalance excels across models, datasets/domains, and sampling temperatures, 3 settings that induce confidence distribution shifts.
> 5. **Observation Across Models (`Appendix C.7`)**: Confidence proves a broadly applicable indicator of over-/underthinking across 4 models spanning 1.5B to 32B
>
> **In-Depth Ablations**
>
> 1. **Gating Functions (`Appendix C.2`)**: ReBalance’s performance remains robust across alternative gating functions and may further enhance with additional engineering.
> 2. **Model Behavior Controlling Methods (`Appendix C.9`)**: Substituting ReBalance’s steering-based approach with temperature tuning maintains strong results, highlighting balanced thinking as the primary factor of effectiveness.
> 3. **Steering Vector Extraction Methods (`Appendix C.10`)**: Adopting Eq. (3) for over-/underthinking modeling quantitatively validates confidence as a necessary indicator.
>
> **Side Effects and Interpretability**
>
> 1. **Creativity and Naturalness Analysis (`Appendix C.6`)**: Evaluations on 17 creativity and naturalness metrics indicate that ReBalance not only prevents negative effects but also enhances these qualities.
> 2. **Semantic Change (`Appendix C.6`)**: Keyword frequency analysis shows ReBalance effectively reduces redundant expressions by reshaping confidence distributions, thus shortening reasoning length.
>
> **Efficiency and Comparative Analysis**
>
> 1. **Efficiency Analysis (`Appendix I`)**: Comparative evaluation assesses ReBalance’s efficiency regarding TPS, TPR, GPU memory usage, and output token count.
> 2. **Comparison with More Early-Exit Methods (`Appendix C.8`)**: Reproduce closed-source early-exit methods and compare with ReBalance.

---

### Meta-Review · Area_Chair_ou8P · 2026-01-11

**Summary:**

The paper proposes a stepwise confidence-based dynamic control method to mitigate both overthinking and underthinking problems. Given a small training set, the proposed ReBalance method extracts a steering vector from the hidden states of over/underthinking instances, and fits a control function of confidence and confidence variance to determine the steering direction and strengths. The inference keeps the reasoning hidden state away from the over/underthinking boundaries by dynamically adjusting the steering direction and strengths according to the control function. The paper demonstrates the effectiveness of ReBalance on reducing token counts and obtaining promising reasoning performance across multiple benchmarks and models in several different settings.

**Reviewer Concerns:**

- Complex engineering efforts with less emphasis on the interpretation of the success of the steering approach.
- Too many hyperparameters and the requirement of a small seen dataset make the "training-free" claim questionable.
- Lack of experiments on different domains and difficulty levels.
- Lack of throughput efficiency comparison.
- Lack of comparisons with a few baselines on efficient reasoning.
- Lack of comparisons in a multi-sample setting.

**Reviewer Scores:**

- Two reviewers who gave positive ratings at the beginning (10 and 8) confirmed that their concerns have been addressed thoroughly, so I think they will keep their positive ratings.
- The other two reviewers who gave 4 and 6 have main concerns about experiments and the engineering complexity of this method. I carefully checked the author's response. The concerns about experiments have been mostly addressed by additional experiments. However, the concerns regarding the complexity and hyperparameters still hold (despite the authors' claims that all hyperparameters can be automatically determined except $B_u$), as I am also confused about how the quantiles and thresholds in Eq. 4 can be automatically determined.

---

### Decision · Program_Chairs · 2026-01-26

Accept (Poster)